# GraphMaster: Automated Graph Synthesis via LLM Agents in Data-Limited Environments

**Enjun Du[1], Xunkai Li[1], Tian Jin[2], Zhihan Zhang[1], Rong-Hua Li[1],[*] Guoren Wang[1]**
[1]Beijing Institute of Technology
[2]The Hong Kong University of Science and Technology (Guangzhou)

## Abstract

The era of foundation models has revolutionized AI research, yet Graph Foundation Models (GFMs) remain constrained by the scarcity of large-scale graph corpora. Traditional graph data synthesis techniques primarily focus on simplistic structural operations, lacking the capacity to generate semantically rich nodes with meaningful textual attributes—a critical limitation for real-world applications. While large language models (LLMs) demonstrate exceptional text generation capabilities, their direct application to graph synthesis is impeded by context window limitations, hallucination phenomena, and structural consistency challenges. To address these issues, we introduce **GraphMaster**—the first multi-agent framework specifically designed for graph data synthesis in data-limited environments. GraphMaster orchestrates four specialized LLM agents (Manager, Perception, Enhancement, and Evaluation) that collaboratively optimize the synthesis process through iterative refinement, ensuring both semantic coherence and structural integrity. To rigorously evaluate our approach, we create new data-limited "Sub" variants of six standard graph benchmarks, specifically designed to test synthesis capabilities under realistic constraints. Additionally, we develop a novel interpretability assessment framework that combines human evaluation with a principled Grassmannian manifold-based analysis, providing both qualitative and quantitative measures of semantic coherence. Experimental results demonstrate that GraphMaster significantly outperforms traditional synthesis methods across multiple datasets, establishing a strong foundation for advancing GFMs in data-scarce environments.[2]

## 1 Introduction

In the era of foundation models, unprecedented advances in natural language processing [43, 59] and computer vision [14, 48, 15, 26] have been enabled by massive training corpora [57, 55, 58, 56, 66, 52, 9]. Graph Foundation Models (GFMs) [31, 28, 43, 59, 8] represent a promising frontier for AI in graph-structured data, yet their development faces a critical bottleneck: the scarcity of large-scale, diverse graph datasets.Unlike text and image domains where data collection is relatively straightforward, gathering and annotating graph data often requires specialized expertise and significant resources. This data quantity constraint has become the primary challenge for training robust GFMs, particularly as model size increases and demands exponentially more training examples for optimal performance.

Graph data synthesis offers a strategic solution to this fundamental constraint by automatically generating new graph samples that maintain both semantic richness and structural validity. Existing synthesis approaches, however, face substantial limitations. Edge-level operations [63, 65] manipulate

---

[*] Corresponding author

[2]Code is available on `https://github.com/EnjunDu/GraphMaster`.

39th Conference on Neural Information Processing Systems (NeurIPS 2025).

existing connections but cannot create novel nodes or patterns. Node-level mixing techniques like GraphMixup [53] generate synthetic nodes by interpolating features but often produce semantically inconsistent attributes, particularly with textual features. Graph-level synthesis methods such as G-Mixup [18] create entirely new graphs but struggle to balance global structure with local semantic coherence. The core limitation across these traditional methods is their inability to simultaneously preserve meaningful semantics while generating structurally valid expansions—a deficiency particularly pronounced when handling text-attributed graphs (TAGs) where both connectivity patterns and textual node features must remain coherent.

Large language models have demonstrated remarkable capabilities in understanding and generating text [19, 11, 61, 24, 2, 45], suggesting potential for synthesizing text-attributed graphs. However, directly applying LLMs encounters several critical challenges: standard context windows cannot process entire graphs with numerous textual nodes [3]; LLMs excel at semantic understanding but struggle to maintain structural consistency [12]; and without proper coordination, they tend to produce inconsistent or hallucinated content that fails to capture the intricate balance between topology and semantics [33]. Furthermore, in realistic scenarios with limited available data, LLMs have insufficient examples to learn complex graph patterns [27, 44, 25].

To address these challenges, we propose **GraphMaster**, a novel multi-agent framework specifically designed for graph synthesis in data-limited environments. GraphMaster decomposes the complex synthesis task into specialized sub-tasks handled by four collaborative LLM-powered agents, each targeting specific challenges: (1) The Manager Agent coordinates the overall process and determines optimal synthesis strategies based on current graph characteristics, orchestrating the complex synthesis workflow; (2) The Perception Agent analyzes graph structure and employs advanced sampling to identify representative subgraphs processable within LLM context constraints, directly addressing the context window limitations; (3) The Enhancement Agent generates new nodes and edges with consistent semantics and structure, mitigating hallucination by maintaining coherence with existing graph elements; and (4) The Evaluation Agent assesses quality based on both semantic coherence and structural integrity, providing feedback for iterative improvement to ensure structural and semantic consistency. This decomposition enables targeted solutions for each challenge that a single-pass LLM approach cannot address.

Through this collaborative, iterative process, these specialized agents overcome the limitations of both traditional methods and direct LLM applications. The multi-agent architecture enables GraphMaster to effectively balance semantic richness with structural validity—producing high-quality synthetic graph data even with limited training examples. By introducing modular reasoning (through task decomposition), semantic control (via specialized agent expertise), and iterative optimization (through feedback cycles), GraphMaster achieves synthesis capabilities beyond what single-pass approaches can deliver.

Our contributions can be summerized as follows:

- **New perspective for LLM-based TAG Synthesis**: we first propose a novel multi-agent framework from the RAG perspective to synthesize TAG under data-limited environment. By integrating context retrieval with iterative feedback, this new perspective enables both semantic richness and structural fidelity.

- **Groundbreaking Benchmark**: We introduce a standardized "Data-limited" variant testbed for text-attributed graph synthesis and develop a dual-perspective interpretability assessment—combining expert human evaluation with Grassmann manifold-based analysis—to provide reproducible comparisons and deep semantic-structural insights.

- **State-of-the-Art Performance**: Extensive experiments on multiple datasets and GNN architectures demonstrate that our method consistently outperforms existing baselines, setting a new benchmark for data-limited TAG synthesis.

## 2 Background Methods

### 2.1 Classic Graph Data Synthesis Methods

Traditional graph data synthesis methods [7] address data scarcity through various approaches. Edge-level operations [63] modify topology by adding or deleting connections. Node-level techniques like

GraphSMOTE [62] generate new nodes through minority class interpolation. Graph-level methods such as G-Mixup [18] create entirely new graph instances via graphon interpolation. Interpolation-based approaches [40, 39] combine hidden representations to enhance model robustness. Despite their diversity, these methods primarily focus on structural manipulations without generating semantically meaningful textual attributes.

## 2.2 LLM-based Multi-Agent Systems for Data Generation

Recent LLM-powered multi-agent frameworks demonstrate capabilities for complex data generation tasks. General collaboration systems like Self-Instruct [41] and distributed simulation platforms [34] establish architectures for coordinated AI systems. In graph contexts, approaches like GoG [47] and LLM-based social simulations [20] leverage semantic understanding for graph-related tasks. However, specific applications for text-attributed graph synthesis in data-limited environments remain largely unexplored.

## 2.3 Problem Formulation: Graph Data Synthesis

**Text-Attributed Graphs.** We formally define a text-attributed graph (TAG) as $\mathcal{G} = (\mathcal{V}, \mathcal{E}, \mathcal{X}, \mathcal{Y})$, where $\mathcal{V} = \{v_1, v_2, \ldots, v_N\}$ is a set of $N$ nodes, $\mathcal{E} \subseteq \mathcal{V} \times \mathcal{V}$ is the set of edges with corresponding adjacency matrix $\mathcal{A} \in \{0,1\}^{N \times N}$, $\mathcal{X} = \{x_1, x_2, \ldots, x_N\}$ is the set of textual features with each $x_i$ corresponding to node $v_i \in \mathcal{V}$, and $\mathcal{Y} = \{y_1, y_2, \ldots, y_N\}$ represents the set of node labels where $y_i \in \{1, 2, \ldots, C\}$ for $C$ distinct classes.

**Knowledge Extraction.** Given the context length constraints of LLMs, we define a knowledge extraction function $\Phi : \mathcal{G} \to \mathcal{K}$ that samples a representative subgraph as:

$$\mathcal{K} = \Phi(\mathcal{G}) = (\mathcal{V}_k, \mathcal{E}_k, \mathcal{X}_k, \mathcal{Y}_k), \tag{1}$$

where $\mathcal{V}_k \subset \mathcal{V}$, $\mathcal{E}_k = \{(v_i, v_j) \in \mathcal{E} \mid v_i, v_j \in \mathcal{V}_k\}$, and $\mathcal{X}_k, \mathcal{Y}_k$ are the corresponding text attributes and labels. The extraction function $\Phi$ employs specialized sampling strategies to ensure $\mathcal{K}$ captures both structural and semantic characteristics of $\mathcal{G}$ while remaining within LLM context limits.

**Graph Synthesis Process.** The graph synthesis process is formalized as a function $\Psi : \mathcal{K} \to \mathcal{G}_s$ that generates new graph elements based on the extracted knowledge:

$$\mathcal{G}_s = \Psi(\mathcal{K}) = (\mathcal{V}_s, \mathcal{E}_s, \mathcal{X}_s, \mathcal{Y}_s), \tag{2}$$

where $\mathcal{G}_s$ represents the synthesized graph components. Function $\Psi$ is implemented through our framework that encompasses both semantic understanding and structural pattern recognition.

**Graph Synthesis.** The final enhanced graph merges the original and synthesized components:

$$\mathcal{G}_{\text{new}} = \mathcal{G} \oplus \mathcal{G}_s = (\mathcal{V} \cup \mathcal{V}_s, \mathcal{E} \cup \mathcal{E}_s \cup \mathcal{E}_c, \mathcal{X} \cup \mathcal{X}_s, \mathcal{Y} \cup \mathcal{Y}_s), \tag{3}$$

where $\mathcal{E}_c = \{(v_i, v_j) \mid v_i \in \mathcal{V}, v_j \in \mathcal{V}_s\}$ represents newly created edges connecting original and synthetic nodes. The quality of $\mathcal{G}_{\text{new}}$ is evaluated based on both semantic coherence (how well $\mathcal{X}_s$ aligns with original textual patterns) and structural fidelity (how well $\mathcal{E}_s$ and $\mathcal{E}_c$ preserve the topological characteristics of $\mathcal{G}$).

## 3 The Proposed Method

We present GraphMaster, a multi-agent framework conceptualized through the lens of Retrieval-Augmented Generation (RAG) [50, 54] to address the challenges of graph synthesis in data-constrained environments. As illustrated in Figure 1, GraphMaster implements a hierarchical RAG paradigm wherein four specialized LLM-powered agents operate collaboratively in a recursive optimization loop to generate semantically rich and structurally coherent graph extensions.

## 3.1 Framework Overview: RAG-Based Multi-Agent Architecture

GraphMaster formalizes graph synthesis as an iterative RAG process, operating through specialized agents in a closed-loop optimization system. While a single LLM might possess the theoretical

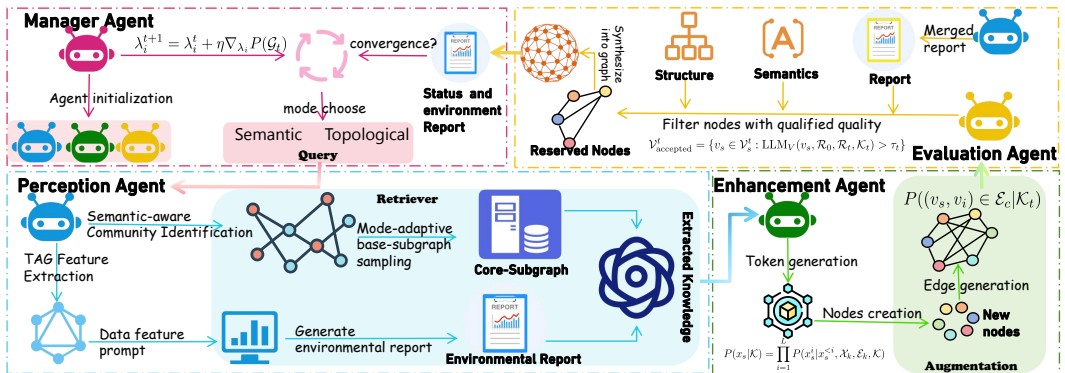

Figure 1: GraphMaster: A hierarchical multi-agent framework for text-attributed graph synthesis.

capability to understand graph structures, the inherent complexity of generating coherent graph data necessitates a specialized multi-agent approach for three critical reasons. First, real-world graphs substantially exceed typical LLM context windows, requiring strategic sampling and knowledge extraction. Second, maintaining structural consistency across generated elements demands focused attention on connectivity patterns that single-pass generation cannot guarantee. Third, controlling hallucination requires continuous evaluation and refinement through iterative feedback. A collaborative multi-agent architecture effectively addresses these challenges through specialization and integration:

$$\mathcal{G}_{\text{new}} = \Psi_{\text{RAG}}(\mathcal{G}, \mathcal{Q}, \mathcal{R}, \mathcal{A}_{\text{retrieve}}, \mathcal{A}_{\text{generate}}, \mathcal{A}_{\text{evaluate}}) \tag{4}$$

where $\mathcal{G}$ is the original graph, $\mathcal{Q}$ represents the query formulation (enhancement mode), $\mathcal{R}$ denotes the retrieval strategy, and $\mathcal{A}_{\text{retrieve}}$, $\mathcal{A}_{\text{generate}}$, and $\mathcal{A}_{\text{evaluate}}$ correspond to the agent-specific functions for retrieval, generation, and evaluation, respectively. In each iteration, the Manager Agent formulates the query to guide the synthesis process, the Perception Agent retrieves relevant context to overcome context window limitations, the Enhancement Agent generates new content while maintaining structural consistency, and the Evaluation Agent assesses quality and addresses potential hallucinations–with this cycle continuing until convergence. This collaborative framework enables each agent to focus on a specific challenge while collectively producing coherent graph extensions that a single-pass approach cannot achieve.

## 3.2 Manager Agent: Query Optimization and Control Mechanism

The Manager Agent serves as the meta-cognitive controller that formulates the synthesis query $\mathcal{Q}_t$ at iteration $t$ based on a comprehensive environmental status report $\mathcal{R}_t = \text{LLM}_P(\mathcal{G}_t)$ generated by the Perception Agent. The mode selection function is formalized as: $M_t = \text{LLM}_M(\mathcal{R}_t) \in \{\text{semantic}, \text{topological}\}$ where $\text{LLM}_M$ represents the Manager Agent's reasoning process that analyzes community structures and label distributions captured in $\mathcal{R}_t$. This query formulation implements an adaptive mechanism where the Manager optimizes a multi-objective utility function:

$$\omega_t^* = \arg\max_{\omega \in \Omega} \left[ \lambda_1 U_{\text{sem}}(\omega, \mathcal{G}_t) + \lambda_2 U_{\text{struct}}(\omega, \mathcal{G}_t) + \lambda_3 U_{\text{bal}}(\omega, \mathcal{G}_t) \right] \tag{5}$$

where $\Omega$ is the strategy space, $U_{\text{sem}}$, $U_{\text{struct}}$, and $U_{\text{bal}}$ represent semantic coherence, structural integrity, and class balance utilities respectively, with adaptive weights $\lambda_i$ that evolve according to: $\lambda_i^{t+1} = \lambda_i^t + \eta \nabla_{\lambda_i} P(\mathcal{G}_t)$ where $P(\mathcal{G}_t)$ measures synthesis progress and $\eta$ is a learning rate. The Manager orchestrates state transitions, modeled as $s_{t+1} = T(s_t, a_t, M_t)$, where states $s_t$ reflect graph composition and actions $a_t \in \{a_P, a_E, a_V\}$ correspond to agent invocations for Perception, Enhancement, and Evaluation respectively.

## 3.3 Perception Agent: Context-Aware Corpus Retrieval

The Perception Agent implements the retrieval component of the RAG paradigm, extracting a relevant subgraph from the input graph $\mathcal{G}_t$ based on the query $\mathcal{Q}_t = M_t$. This retrieval process is formalized

as: $\mathcal{K}_t = \mathscr{R}(\mathcal{G}_t, \mathcal{Q}_t) = (\mathcal{V}_k, \mathcal{E}_k, \mathcal{X}_k, \mathcal{Y}_k)$ where $\mathcal{K}_t$ represents the retrieved knowledge capsule. The retrieval function $\mathscr{R}$ operates through three sequential stages:

**Semantic-aware Community Identification:** The agent employs a semantic-enriched modularity maximization algorithm to calculate the community distribution of semantic associations for TAG:

$$Q_{\text{sem}} = \frac{1}{2m} \sum_{i,j} \left[ \mathcal{A}_{ij} - \gamma \frac{k_i k_j}{2m} - (1-\gamma) \frac{d_{\text{sem}}(x_i, x_j)}{\sum_{l,m} d_{\text{sem}}(x_l, x_m)} \right] \delta(c_i, c_j) \tag{6}$$

where $d_{\text{sem}}(x_i, x_j) = \frac{\mathbf{x}_i \cdot \mathbf{x}_j}{\|\mathbf{x}_i\| \|\mathbf{x}_j\|}$ computes semantic similarity between node attributes, $k_i$ and $k_j$ represent the degrees of nodes $i$ and $j$, and $\gamma$ balances topological and semantic factors.

**Mode-Adaptive Seed Selection Strategy:** Based on the enhancement mode $M_t$, the agent selects an optimal seed community $\mathcal{C}_b$:

$$\mathcal{C}_b = \begin{cases} \arg\min_i |\mathcal{C}_i| \cdot (1 + \mu \cdot \text{Var}(\{x_j : v_j \in \mathcal{C}_i\})), & \text{if } M_t = \text{semantic}, \\ \{v_j \in \mathcal{V}_{\text{train}} : y_j = \arg\max_c \phi_{\text{imbal}}(c)\}, & \text{if } M_t = \text{topological} \end{cases} \tag{7}$$

where $\phi_{\text{imbal}}(c) = \max_{c'} |\mathcal{V}_{c'}|/|\mathcal{V}_c|$ quantifies class imbalance, Var denotes the variance of node textual features within a community and $\mu$ weights semantic variance importance. For semantic synthesis, the smaller communities with low internal semantic variance are prefer to be selected to establish a cohesive foundation. For topological synthesis, nodes from minority classes are prioritized.

**Hierarchical Stochastic Diffusion Sampling:** The agent employs a mode-conditional Personalized PageRank (PPR) algorithm, defined as $\boldsymbol{\pi}^{(k+1)} = \alpha \boldsymbol{v} + (1-\alpha) W^T \boldsymbol{\pi}^{(k)}$, where the teleportation vector $\boldsymbol{v}$ varies by enhancement mode: $v_i = \frac{1}{|\mathcal{C}_b|}$ if $v_i \in \mathcal{C}_b$ and $M_t = \text{semantic}$, or $v_i = \frac{\mathbb{I}[y_i = \hat{y}]}{\sum_j \mathbb{I}[y_j = \hat{y}]}$ if $v_i \in \mathcal{V}_{\text{train}}$ and $M_t = \text{topological}$, where $\hat{y} = \arg\min_c |\{v_j \in \mathcal{V}_{\text{train}} : y_j = c\}|$ identifies the label with minimal representation. Following diffusion convergence, the final knowledge subgraph is selected as:

$$\mathcal{K}_t = \{v_i \in \mathcal{S}_{\text{top-}K\%} : r_i < \min(1, \beta \cdot \boldsymbol{\pi}_i / \max_j \boldsymbol{\pi}_j)\} \cap \mathcal{S}_{\text{diverse}} \tag{8}$$

where $|\mathcal{K}_t| = N$ is constrained by the LLM context window, $\mathcal{S}_{\text{top-}K\%}$ contains the top $K\%$ nodes by PPR score, $r_i \sim \text{Uniform}(0, 1)$ introduces controlled stochasticity, and $\mathcal{S}_{\text{diverse}}$ ensures community coverage. The environmental status report $\mathcal{R}_t$ encapsulates multi-scale graph properties:

$$\mathcal{R}_t = \left( \rho_{\text{global}}, \{\rho_{\text{class}}^c\}_{c=1}^C, \{\rho_{\text{comm}}^i\}_{i=1}^{|\mathcal{C}|}, \mathcal{D}_{\text{struct}}, \mathcal{D}_{\text{sem}} \right) \tag{9}$$

where $\rho_{\text{global}}$ captures global statistics, $\rho_{\text{class}}^c$ and $\rho_{\text{comm}}^i$ encode class-level and community-level properties, while $\mathcal{D}_{\text{struct}}$ and $\mathcal{D}_{\text{sem}}$ represent structural and semantic distributions.

### 3.4 Enhancement Agent: Context-Conditioned Generation

The Enhancement Agent implements the generation component of the RAG paradigm, synthesizing new graph elements (no more than M% of the knowledge subgraph) based on the retrieved knowledge and environmental report. The synthesis process follows Eq. (2) where $\mathcal{K} = (\mathcal{K}_t, \mathcal{R}_t, M_t)$. For semantic mode, the LLM generates node attributes using a conditional autoregressive model:

$$P(x_s | \mathcal{K}) = \prod_{i=1}^L P(x_s^i | x_s^{<i}, \mathcal{X}_k, \mathcal{E}_k, \mathcal{K}) \tag{10}$$

where $x_s^i$ is the $i$-th token of attribute $x_s$, and $L$ is the sequence length. This formulation enables the LLM to generate coherent textual attributes that maintain consistency with the knowledge subgraph while introducing appropriate variations.

Crucially, regardless of the current enhancement mode, the agent always generates both node attributes and their connections. For topological mode, the LLM models edge connections between new node $v_s$ and existing nodes by estimating the probability:

$$P((v_s, v_i) \in \mathcal{E}_c | \mathcal{K}) = \sigma \left( \theta_1 \cdot \text{sim}(x_s, x_i) + \theta_2 \cdot \frac{|\mathcal{N}(v_i) \cap \mathcal{N}_K(v_s)|}{|\mathcal{N}_K(v_s)|} + \theta_3 \cdot \frac{k_i}{\max_j k_j} \right) \quad (11)$$

where $\mathcal{N}(v_i)$ is the neighborhood of $v_i$, $\mathcal{N}_K(v_s)$ represents neighbors of $v_s$ in the knowledge subgraph, and $\sigma$ is the sigmoid function. The coefficients $\{\theta_j\}_{j=1}^3$ are dynamically adjusted based on the query mode $M_t$. This dual-mode generation enables GraphMaster to adaptively emphasize either semantic coherence or structural fidelity while maintaining integrity across both dimensions.

## 3.5 Evaluation Agent: Multi-dimensional Quality Assessment

The Evaluation Agent implements a comprehensive verification mechanism that integrates four critical information sources:

$$\mathcal{Q}_t = \text{LLM}_V(\mathcal{R}_0, \mathcal{R}_t, \mathcal{K}_t, \mathcal{G}_s^t) \quad (12)$$

where $\mathcal{Q}_t$ represents the quality assessment outcome, $\mathcal{R}_0$ is the initial environmental report serving as a baseline, $\mathcal{R}_t$ is the current environmental report, $\mathcal{K}_t$ is the retrieved knowledge, and $\mathcal{G}_s^t$ is the newly synthesized data. The Evaluation Agent simultaneously assesses two key dimensions: **(i) Semantic Coherence:** Evaluates whether the generated textual attributes are contextually appropriate, domain-consistent, and meaningful within the graph's thematic scope. **(ii) Structural Integrity:** Assesses whether the new edges form logical connections that preserve the original graph's topological patterns while addressing structural gaps.

For each generated node $v_s \in \mathcal{V}_s^t$, the LLM computes a composite quality score, with the final accepted node set defined as:

$$\mathcal{V}_{\text{accepted}}^t = \{v_s \in \mathcal{V}_s^t : \text{LLM}_V(v_s, \mathcal{R}_0, \mathcal{R}_t, \mathcal{K}_t) > \tau_t\} \quad (13)$$

where the threshold $\tau_t$ is adaptively updated: $\tau_t = \tau_{t-1} + \zeta(\bar{\mathcal{F}}_t(\omega_t^*) - \bar{\mathcal{F}}_{t-1}(\omega_{t-1}^*))$ with $\bar{\mathcal{F}}_t(\omega_t^*)$ representing the average quality score at iteration $t$. The convergence determination employs a temporal quality gradient analysis:

$$\text{Converged}_t = \mathbb{I} \left( \max_{j \in \{1,\dots,k\}} |\bar{\mathcal{F}}_t(\omega_t^*) - \bar{\mathcal{F}}_{t-j}(\omega_{t-j}^*)| < \epsilon \wedge \text{LLM}_{\text{goal}}(\mathcal{R}_0, \mathcal{R}_t) = \text{True} \right) \quad (14)$$

where $\mathbb{I}(\cdot)$ is the indicator function and $\text{LLM}_{\text{goal}}$ assesses whether synthesis objectives have been achieved. If convergence is detected, the Evaluation Agent signals task completion to the Manager Agent; otherwise, it triggers another iteration of the synthesis process[3].

## 3.6 Time Complexity Analysis

The time complexity of GraphMaster is dominated by three operations: (1) community detection and PPR computation in the Perception Agent, which run in near-linear time $O(|V_t| + |E_t|)$ on the current graph; (2) LLM inference for node attribute generation and edge probability estimation, which scales with the size $N$ of the retrieved subgraph rather than the full graph; and (3) quality assessment, which evaluates a fixed number of newly generated nodes against predetermined criteria. Since $N$ is constrained by the LLM context window and typically small relative to $|V_t|$, the LLM operations remain efficient regardless of overall graph size. If the iterative process runs for $T$ iterations before convergence (generally small due to the Evaluation Agent's stringent criteria), the overall complexity is $T$ times the per-iteration cost. This architecture enables GraphMaster to scale effectively by leveraging LLMs for semantic generation on bounded contexts while using efficient graph algorithms for structural computations.

---

[3]We compares GraphMaster with recent remarkable graph data synthsis methods in Appendix A. Theoretical analysis of agent capabilities are given in Appendix I.

## 4 Experiment

To evaluate GraphMaster comprehensively, we formulate four research questions: **(RQ1)**: Can GraphMaster generate high-quality text-attributed graph data in data-limited environment? **(RQ2)**: Can the graph data synthesized by GraphMaster retain the original graph features well? **(RQ3)**: Can GraphMaster maintain interpretability well? **(RQ4)**: What is the relative contribution of each component in GraphMaster to the overall synthesis quality?

### 4.1 Overall Performance (RQ1)

We evaluate GraphMaster's ability to synthesize high-quality graph data by applying it to enhance the data-limited datasets we created and assessing whether the enhanced datasets improve downstream model performance. We employ standard metrics including Accuracy and F1 Score as evaluation criteria, with higher values indicating superior performance.

**Baselines and Datasets.** The comparative baselines are categorized into five groups: (1) original TAG training without data synthesis; (2) Classic data augmentation methods: GAugO [63]; (3) LLM-based data aigmentation methods: GraphEdit [16] and LLM4RGNN [60]; (4) Classic data synthesis methods: GraphSmote [62], G-Mixup [18], IntraMix [64], GraphAdasyn [29], FG-SMOTE [42], and AGMixup [30]; (5) LLM-based data synthesis methods: GAG [21] and LLM4NG [51], noting that there are very limited baselines for TAG synthesis using LLM, and we created these two additional baselines named Mixed-LLM and Synthesis-LLM, whose implementations can be found in the Appendix B. Our experiments utilize six widely recognized text-attributed graph datasets: Cora [32], Citeseer [13], Wikics [10], Arxiv2023 [36], and History and Children [49]. It is worth noting that in order to better simulate the data-limited environment to test the effect of data synthesis, we created 6 data-limited datasets, namely SubCora, SubCiteseer, SubWikics, SubHistory, SubArxiv2023, and SubChildren (details are given in Appendix C). In this article, unless otherwise specified, we assume that the augmentation-based method uses the original dataset, while the synthesis-based method uses the data-limited dataset we created. For downstream task evaluation, we implement four established graph neural network architectures: GCN [22], JKNET [46], GraphSage [17] and GAT [38].

**Implement Details.** We ran the entire experiment on eight 80G A100 GPUs, using the QwQ-32B model [37] as the base LLM and enabling it to assume different agent roles through iterative calls. For the background knowledge nodes, we set $N = 30$, and for the newly generated nodes, we configured $M\% = 15\%$ (The hyperparameter selection analysis are given in Appendix E). In training the GNN model, we first initialized the text attributes with Sentence-BERT [35] to generate the initial features before proceeding with training. To ensure the robustness of our experiments, we repeated each experiment 50 times and reported the mean and standard deviation of the results.

Table 1: Comparison of GraphMaster with other TAG synthesis methods in GCN model. Best performance is indicated by the **bold** face numbers, and the underline means the second best. 'Acc' and 'F1' are short for Accuracy and F1 Score, respectively.

| Type | Model | Cora | | Citeseer | | Wikics | | History | | Arxiv2023 | | Children | |
|---|---|---|---|---|---|---|---|---|---|---|---|---|---|
| | | Acc | F1 | Acc | F1 | Acc | F1 | Acc | F1 | Acc | F1 | Acc | F1 |
| Original | Original Model | 88.9±1.1 | 88.5±1.5 | 78.1±1.1 | 75.0±0.3 | 79.7±0.8 | 77.8±0.3 | 84.2±0.6 | 43.1±0.3 | 76.3±1.0 | 54.9±0.5 | 52.6±0.7 | 32.3±1.5 |
| Classic-Aug | GAugO | 88.9±0.8 | 88.0±1.0 | 78.1±0.7 | 77.2±0.3 | 79.9±0.9 | 77.7±0.7 | 84.6±1.5 | 44.7±0.6 | 76.8±1.4 | 53.0±0.3 | 51.8±0.6 | 33.6±0.7 |
| LLM-Aug | GraphEdit | 91.0±0.9 | 89.7±0.6 | 81.9±0.9 | 80.8±1.1 | 82.0±1.1 | 80.7±1.2 | 87.6±0.6 | 45.7±1.3 | 78.0±0.9 | 57.8±0.3 | 54.3±0.7 | 35.7±1.4 |
| | LLM4RGNN | 91.2±0.6 | 88.8±1.1 | 80.9±1.3 | 76.6±1.1 | 83.6±1.4 | 81.6±1.5 | 88.9±0.7 | 48.6±0.4 | 79.3±1.1 | 59.1±0.4 | 55.7±1.4 | 36.7±0.8 |
| Classic-Syn | GraphSmote | 88.7±1.4 | 87.4±1.1 | 78.1±0.5 | 74.6±1.4 | 80.7±1.5 | 78.6±1.4 | 84.9±0.5 | 43.9±0.3 | 76.2±0.4 | 55.5±0.4 | 53.1±0.9 | 33.2±0.6 |
| | G-Mixup | 87.4±1.1 | 87.0±0.7 | 78.2±0.9 | 76.8±1.1 | 79.7±0.4 | 78.0±0.8 | 84.6±0.6 | 43.6±0.6 | 76.6±0.4 | 56.5±0.3 | 53.0±0.6 | 33.0±0.3 |
| | IntraMix | 80.9±0.6 | 82.8±0.7 | 71.3±0.8 | 70.7±0.5 | 73.7±1.0 | 74.5±0.4 | 82.4±1.5 | 42.7±0.6 | 72.4±1.1 | 53.9±0.8 | 45.2±0.9 | 30.1±0.7 |
| | GraphAdasyn | 89.2±0.3 | 88.7±0.5 | 78.9±1.0 | 78.4±1.4 | 80.8±1.2 | 78.9±0.7 | 84.6±1.5 | 46.1±0.8 | 77.5±0.7 | 57.0±1.1 | 53.6±0.5 | 33.0±0.6 |
| | FG-SMOTE | 88.9±1.5 | 87.6±1.0 | 78.7±0.8 | 74.7±1.4 | 81.0±0.8 | 79.0±1.0 | 85.0±1.4 | 44.0±0.9 | 76.4±1.2 | 55.8±1.4 | 53.1±1.5 | 33.3±0.9 |
| | AGMixup | 84.7±0.4 | 86.6±0.4 | 71.7±0.9 | 73.2±1.1 | 78.8±0.9 | 76.6±1.1 | 81.8±0.9 | 42.9±0.3 | 76.8±1.1 | 53.7±0.5 | 53.6±1.0 | 32.6±0.8 |
| LLM-Syn | GAG | 91.0±1.2 | 89.3±0.4 | 82.8±0.9 | 80.0±1.5 | 84.9±1.3 | 83.2±0.7 | 88.9±0.5 | 49.8±0.8 | 79.9±0.5 | 59.4±1.0 | 56.7±0.8 | 38.0±0.5 |
| | LLM4NG | 85.9±0.3 | 84.0±0.2 | 73.9±0.2 | 72.0±0.3 | 72.8±0.1 | 72.9±0.4 | 82.5±0.5 | 48.4±0.3 | 79.0±0.2 | 61.2±0.5 | 44.6±0.2 | 27.2±0.4 |
| | Mixed-LLM | 89.9±0.5 | 89.3±0.4 | 83.5±0.9 | 81.3±0.8 | 84.9±0.9 | 83.4±0.8 | 89.2±1.3 | 55.8±0.8 | 81.4±1.5 | 61.2±0.9 | 60.0±0.7 | 39.6±0.7 |
| | Synthesis-LLM | 89.8±1.3 | 89.1±1.0 | 84.5±1.0 | 82.7±1.1 | 84.8±0.4 | 83.2±0.5 | 89.4±1.3 | 53.4±1.3 | 81.0±1.2 | 62.3±0.8 | 60.9±1.0 | 40.1±0.9 |
| | **GraphMaster** | **93.7±1.0** | **92.5±1.0** | **88.3±0.9** | **87.7±1.1** | **87.9±0.8** | **86.8±0.9** | **92.6±1.3** | **63.4±1.4** | **87.9±1.3** | **66.3±1.5** | **68.8±1.4** | **47.8±1.3** |

**Results.** As shown in Table 1 (Due to space limitation, other three models' results are given in Appendix F.), GraphMaster consistently outperforms all baselines, demonstrating the superiority of our approach. Notably, we observed that some baseline methods even yield lower performance than the original dataset. This is primarily because traditional graph synthesis techniques fail to capture the semantic nuances of sentences; consequently, when using Sentence-BERT embeddings

instead of bag-of-words representations, their effectiveness is significantly diminished. Moreover, the other LLM-based baselines we compared against mainly focus on anti-interference detection or data synthesis on other scenarios rather than TAG data synthesis, resulting in their performance being significantly lower than that of GraphMaster, which targets TAG data synthesis. Finally, the two LLM-based TAG synthesis baselines we developed show significant advantages over traditional baselines. However, since they cannot fully understand the semantics and topological structure of TAG, although they are higher than other baselines, they are still significantly lower than GraphMaster[4].

## 4.2 Synthetic Graph Feature Analysis (RQ2)

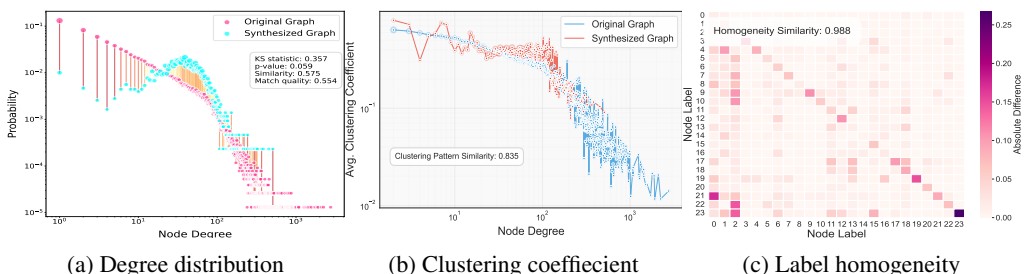

(a) Degree distribution      (b) Clustering coeffiecient      (c) Label homogeneity

Figure 2: Graph feature analysis on Children dataset.

Our second research question examines whether the new graph data generated by GraphMaster in data-limited environments can maintain consistency with the original graph's structural features. We conducted a comprehensive analysis across three dimensions: degree distribution, clustering coefficient, and label homogeneity. As shown in Figure 2, GraphMaster demonstrates excellent performance in preserving the network's topological backbone. For example, the two-sample Kolmogorov–Smirnov test statistic between the degree distributions of the original and synthesized Children graphs is 0.357 ($p = 0.059$), indicating no statistically significant difference. The clustering coefficient similarity score is 0.835, which represents a substantial improvement over the original data-limited Children graph (0.785). Concurrently, the label homogeneity similarity reaches an impressive 0.988 (the heatmap of label–label connection frequencies is almost identical for original vs. synthetic), indicating minimal differences in class mixing patterns. These characteristics show that GraphMaster can generate high-quality synthetic graphs that retain key structural properties of the original data. (Additional graph comparison figures are provided in Appendix G.)

## 4.3 Interpretability Analysis (RQ3)

To evaluate the transparency of our GraphMaster model, we conduct both human-centered and algorithmic assessments of interpretability (theoretical details in Appendix H). For human evaluation, 50 expert reviewers rated 200 synthesis instances across three dimensions: process transparency, decision justification, and outcome predictability. The overall Traceability Score quantifies how well humans understand the generation process:

$$T_{\text{score}} = \frac{1}{R \cdot N} \sum_{r=1}^{R} \sum_{i=1}^{N} t_{r,i}, \tag{15}$$

where $t_{r,i}$ represents the score given by reviewer $r$ for instance $i$. In parallel, we leverage a Grassmann manifold-based approach to systematically assess semantic consistency of synthesized nodes. This mathematical framework provides a principled way to measure how well generated nodes align with the semantic direction of background knowledge, yielding coherence scores in the range $[0, 1]$.

Our human evaluation results demonstrate that GraphMaster exhibits excellent interpretability, with an average traceability score of $T_{\text{score}} = 0.92$, significantly outperforming Mixed-LLM (0.66) and Synthesis-LLM (0.59). For the Grassmann manifold-based evaluation method, Figure 3a shows that GraphMaster significantly outperforms comparative methods in terms of semantic coherence,

---

[4]Case study are given in Appendix K.

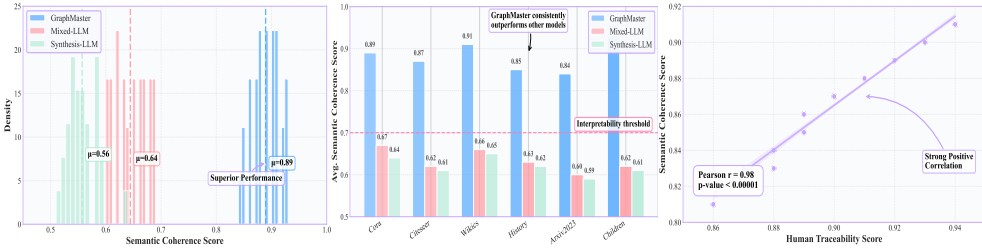

| (a) Semantic Score Distribution | (b) Cross-Dataset Comparison | (c) Human-Algorithm Agreement |

Figure 3: Interpretability Analysis of GraphMaster.

indicating that GraphMaster can generate nodes highly aligned with the principal semantic direction. The score distribution more concentrated in higher value regions also indicates stronger consistency in the quality of generated nodes. Figure 3b shows that GraphMaster maintains high performance across all datasets, consistently exceeding the interpretability threshold of 0.7. Figure 3c demonstrates a strong correlation between human ratings and semantic coherence scores (r=0.78, p<0.00001), further validating our Grassmann manifold-based approach as an effective metric for measuring interpretability. This alignment between human judgment and geometric measures confirms the practical relevance and feasibility of our mathematical framework.

## 4.4 Ablation Study (RQ4)

Table 2: Ablation experiment in GCN model.

| model | Cora | | Citeseer | | Wikics | | History | | Arxiv2023 | | Children | |
|---|---|---|---|---|---|---|---|---|---|---|---|---|
| | Acc | F1 | Acc | F1 | Acc | F1 | Acc | F1 | Acc | F1 | Acc | F1 |
| QwQ-32B | **93.7** ± 1.0 | **92.5** ± 1.0 | **88.3** ± 0.9 | **87.7** ± 1.1 | **87.9** ± 0.8 | **86.8** ± 0.9 | **92.6** ± 1.3 | **63.4** ± 1.4 | **87.9** ± 1.3 | **66.3** ± 1.5 | **68.8** ± 1.4 | **47.8** ± 1.3 |
| Qwen-32B | 91.4 ± 0.9 | 90.2 ± 0.9 | 86.5 ± 1.1 | 85.4 ± 1.0 | 85.7 ± 0.9 | 84.2 ± 0.9 | 90.1 ± 1.2 | 60.8 ± 0.9 | 85.3 ± 0.8 | 64.2 ± 1.1 | 66.5 ± 1.1 | 45.9 ± 1.0 |
| DeepSeek-R1-32B | 91.8 ± 0.8 | 90.6 ± 0.8 | 86.8 ± 1.0 | 85.8 ± 1.1 | 85.9 ± 0.8 | 84.6 ± 0.9 | 90.5 ± 1.2 | 61.2 ± 0.9 | 85.6 ± 1.0 | 64.5 ± 1.2 | 66.9 ± 1.2 | 46.1 ± 1.4 |
| LLaMA-33B | 91.1 ± 0.9 | 90.0 ± 1.1 | 86.0 ± 0.8 | 85.0 ± 1.2 | 85.4 ± 0.5 | 84.0 ± 0.6 | 89.8 ± 0.7 | 60.5 ± 0.9 | 85.0 ± 0.7 | 63.8 ± 0.8 | 66.2 ± 0.9 | 45.7 ± 0.9 |
| w.o Perception Agent | 88.5 ± 0.8 | 87.3 ± 0.7 | 83.6 ± 0.7 | 82.5 ± 1.1 | 83.2 ± 0.7 | 82.0 ± 0.6 | 87.4 ± 1.1 | 57.9 ± 0.9 | 82.6 ± 0.8 | 61.5 ± 0.9 | 63.2 ± 1.3 | 43.4 ± 0.7 |
| w.o Evaluation Agent | 89.6 ± 0.7 | 88.5 ± 0.7 | 84.8 ± 0.6 | 83.9 ± 0.9 | 84.3 ± 0.9 | 83.1 ± 0.6 | 88.9 ± 0.7 | 59.2 ± 0.7 | 83.8 ± 0.9 | 62.7 ± 1.1 | 64.9 ± 1.1 | 44.6 ± 1.3 |
| N=20 | 91.5 ± 0.8 | 90.3 ± 0.5 | 86.4 ± 0.7 | 85.3 ± 0.8 | 85.6 ± 0.6 | 84.3 ± 0.7 | 90.0 ± 0.8 | 60.7 ± 0.8 | 85.1 ± 0.7 | 63.9 ± 0.7 | 66.3 ± 0.8 | 45.7 ± 0.6 |
| N=30 | **93.7** ± 1.0 | **92.5** ± 1.0 | **88.3** ± 0.9 | **87.7** ± 1.1 | **87.9** ± 0.8 | **86.8** ± 0.9 | **92.6** ± 1.3 | **63.4** ± 1.4 | **87.9** ± 1.3 | **66.3** ± 1.5 | **68.8** ± 1.4 | **47.8** ± 1.3 |
| N=40 | 92.0 ± 0.9 | 90.8 ± 0.9 | 86.7 ± 0.8 | 85.7 ± 1.0 | 85.8 ± 0.7 | 84.6 ± 0.7 | 90.3 ± 1.2 | 61.0 ± 1.3 | 85.4 ± 1.2 | 64.1 ± 1.4 | 66.7 ± 1.3 | 46.0 ± 1.2 |
| M=10% | 91.7 ± 0.9 | 90.5 ± 0.9 | 86.6 ± 0.7 | 85.5 ± 0.9 | 85.7 ± 0.6 | 84.4 ± 0.7 | 90.2 ± 1.1 | 60.8 ± 1.2 | 85.3 ± 1.1 | 64.0 ± 1.3 | 66.6 ± 1.2 | 45.8 ± 1.1 |
| M=15% | **93.7** ± 1.0 | **92.5** ± 1.0 | **88.3** ± 0.9 | **87.7** ± 1.1 | **87.9** ± 0.8 | **86.8** ± 0.9 | **92.6** ± 1.3 | **63.4** ± 1.4 | **87.9** ± 1.3 | **66.3** ± 1.5 | **68.8** ± 1.4 | **47.8** ± 1.3 |
| M=20% | 91.3 ± 0.8 | 90.1 ± 0.8 | 86.2 ± 0.6 | 85.1 ± 0.8 | 85.3 ± 0.5 | 83.9 ± 0.6 | 90.0 ± 0.7 | 60.5 ± 0.8 | 84.8 ± 0.6 | 63.5 ± 0.6 | 66.0 ± 0.7 | 45.3 ± 0.5 |

In this section, we investigate the relative importance of various components within the GraphMaster framework and their impact on synthesis quality. We systematically analyze how different agent configurations affect the overall performance. We selected Qwen-32B [1], Deepseek-R1-32B [5] and Llama-33B [4], three models with parameters around 32B, as comparison models. Additionally, we examine how varying the size of the background knowledge base ($N = |\mathcal{K}|$) and the percentage of newly generated nodes ($M\%$) influences synthesis effectiveness. We trained the model using GCN on six datasets, and the results are presented in Table 2. Our findings indicate that the performance varies significantly across different LLMs, with QwQ-32B consistently outperforming the alternatives by 1.5-2.3% across all datasets. Notably, DeepSeek-R1-32B achieves the second-best performance despite LLaMA-33B having more parameters, suggesting that model architecture and pre-training approach are more critical than raw parameter count for this task.

The ablation results reveal that removing either the Perception Agent or Evaluation Agent substantially degrades performance (by 5.2% and 4.1% on average, respectively), with the Perception Agent proving particularly crucial. This confirms that both specialized components play essential roles in maintaining generation quality and cannot be omitted from the framework. Regarding hyperparameters, we observe that $N = 30$ consistently outperforms both smaller ($N = 20$) and larger ($N = 40$) knowledge bases across all datasets. Similarly, setting $M = 15\%$ yields optimal results compared to both $M = 10\%$ and $M = 20\%$. These findings demonstrate that while sufficient context is necessary for high-quality synthesis, excessive background knowledge can dilute the model's focus. Likewise, generating too many nodes simultaneously reduces overall quality due to limitations in the model's generative capacity when handling multiple interdependent elements.

## 5 Conclusion

In this paper, we introduced GraphMaster, the first multi-agent framework for text-attributed graph synthesis that successfully addresses the critical bottleneck of data scarcity in training GFMs. By orchestrating specialized LLM agents in a hierarchical RAG paradigm, our approach systematically overcomes the limitations of traditional synthesis methods, generating semantically rich and structurally coherent graph extensions even in severely data-constrained environments. Beyond the framework itself, we created specialized data-limited variants of six standard graph benchmarks and developed a novel dual-perspective interpretability assessment methodology that combines expert human evaluation with a theoretically grounded Grassmannian manifold-based analysis. Comprehensive experiments demonstrate GraphMaster's consistently superior performance across diverse datasets and downstream GNN architectures. Future work could explore multi-scale synthesis approaches that simultaneously model global topology and local semantics, knowledge transfer mechanisms from data-rich to data-limited domains, and adaptive sampling strategies optimized specifically for synthesis objectives. This work not only provides an immediate solution to the graph data scarcity problem but also establishes foundational methodologies for advancing interpretable graph data synthesis.

## 6 Acknowledgements

This work is supported by the NSFC Grants U2241211, 62427808 and U24A20255.

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

# A  Comparison of GraphMaster with Recent Representative Works

Recent advancements in leveraging Large Language Models (LLMs) for graph-related tasks have produced several notable approaches. In this section, we conduct a rigorous comparison between our proposed GraphMaster framework and two recent representative works: GAG [21] and LLM4NG [51]. We structure our analysis across three key dimensions: motivation, methodology, and application scenarios, to clearly delineate the unique contributions and advantages of GraphMaster.

## A.1  Comparison Between GraphMaster and GAG

### A.1.1  Motivation

GraphMaster addresses a fundamental bottleneck in developing Graph Foundation Models (GFMs): the scarcity of large-scale, semantically rich graph datasets. Our work specifically targets the generation of high-quality text-attributed graphs in data-limited environments, focusing on both semantic coherence and structural integrity. In contrast, GAG [21] primarily aims to simulate the dynamic evolution of social graphs through actor-item interactions, with emphasis on reproducing macroscopic network properties such as power-law degree distributions and small-world phenomena. While GAG attempts to capture emergent properties of large-scale social networks, it lacks explicit mechanisms for maintaining semantic coherence in node attributes, which is a critical requirement for training effective GFMs.

### A.1.2  Methodology

GraphMaster implements a hierarchical multi-agent framework formalized through the RAG paradigm, where specialized agents perform distinct functions within a closed-loop optimization system:

$$G_{new} = \Psi_{RAG}(G, Q, R, A_{retrieve}, A_{generate}, A_{evaluate}) \tag{16}$$

where $G$ is the original graph, $Q$ represents the query formulation, $R$ denotes the retrieval strategy, and $A_{retrieve}$, $A_{generate}$, and $A_{evaluate}$ correspond to the agent-specific functions. Our Perception Agent extracts knowledge through semantic-enriched modularity maximization:

$$Q_{sem} = \frac{1}{2m} \sum_{i,j} \left[ A_{ij} - \gamma \frac{k_i k_j}{2m} - (1 - \gamma) \frac{d_{sem}(x_i, x_j)}{\sum_{l,m} d_{sem}(x_l, x_m)} \right] \delta(c_i, c_j) \tag{17}$$

In contrast, GAG employs a bipartite graph model where homogeneous "actor" agents interact with items through a retrieval system. Their S-RAG algorithm models actor-item interactions but lacks explicit quality control:

$$p_{ij} = \sigma \left( \theta_1 \cdot sim(x_i, x_j) + \theta_2 \cdot \frac{|N(v_i) \cap N(v_j)|}{|N(v_i)|} + \theta_3 \cdot \frac{k_j}{\max_l k_l} \right) \tag{18}$$

GraphMaster's multi-agent architecture provides several key advantages: (1) Our Manager Agent dynamically optimizes a multi-objective utility function that balances semantic coherence, structural integrity, and class balance; (2) Our Evaluation Agent implements a comprehensive verification mechanism with adaptive thresholds; and (3) Our theoretical framework provides formal guarantees for information preservation, generation quality, and convergence properties.

### A.1.3  Application Scenarios

GraphMaster is explicitly designed for enhancing text-attributed graphs in data-limited environments, making it particularly suitable for academic, industrial, and web-scale applications where data acquisition is costly or restricted. Our framework produces semantically rich and structurally coherent graph extensions that serve as high-quality training data for GFMs. GAG primarily focuses on social network simulation, with applications in modeling online user interactions and social dynamics. While GAG can generate large-scale graphs (up to 100,000 nodes), it sacrifices semantic richness for scale and does not provide guarantees on the quality of textual attributes. Our evaluation framework, combining human assessment with Grassmannian manifold-based analysis, demonstrates that GraphMaster consistently produces higher-quality graph extensions that better preserve both semantic and structural characteristics of the original data.

## A.2 Comparison Between GraphMaster and LLM4NG

### A.2.1 Motivation

GraphMaster addresses the broad challenge of data scarcity for GFMs through comprehensive graph synthesis that enhances both node attributes and structural properties. In contrast, LLM4NG [51] specifically targets few-shot learning scenarios in node classification, with a narrow focus on enhancing class-level information. While LLM4NG aims to improve classification performance by adding labeled examples, it does not address the fundamental issue of enhancing the overall quality and representational capacity of graph data. GraphMaster's approach is more comprehensive, as it treats the graph synthesis problem holistically, generating high-quality nodes and edges that maintain both semantic and structural coherence.

### A.2.2 Methodology

GraphMaster employs a sophisticated multi-agent system with specialized roles and iterative refinement cycles. Our Enhancement Agent generates new nodes through a conditional autoregressive model:

$$P(x_s|K) = \prod_{i=1}^{L} P(x_s^i|x_s^{<i}, X_k, E_k, K) \tag{19}$$

and models edge connections with a probability function that balances semantic, structural, and degree-based factors:

$$P((v_s, v_i) \in E_c|K) = \sigma\left(\theta_1 \cdot sim(x_s, x_i) + \theta_2 \cdot \frac{|N(v_i) \cap N_K(v_s)|}{|N_K(v_s)|} + \theta_3 \cdot \frac{k_i}{\max_j k_j}\right) \tag{20}$$

LLM4NG adopts a significantly simpler approach, primarily generating text based on label characteristics alone. Its core idea is to let the large language model generate text that meets class characteristics based solely on the name of the label:

$$s_g = \text{LLM}(\text{Prompt}(c)), c \in C \tag{21}$$

followed by a basic edge predictor that often introduces noise and structural inconsistencies:

$$\hat{y}e(h_{v_i}, h_{v_j}) = \text{MLP}(h_{v_i}||h_{v_j}) \tag{22}$$

GraphMaster's methodology offers several crucial advantages: (1) Our approach maintains structural consistency through explicit modeling of node-edge relationships; (2) Our iterative refinement process ensures high-quality synthesis through adaptive thresholds; and (3) Our theoretical framework provides formal guarantees on the quality and convergence of the synthesis process. LLM4NG lacks these quality assurance mechanisms and theoretical foundations.

### A.2.3 Application Scenarios

GraphMaster addresses a wider range of data-limited scenarios and can enhance graphs for various downstream tasks beyond classification. Our framework is particularly effective in scenarios requiring high semantic coherence and structural fidelity, such as scientific discovery, knowledge graph completion, and recommendation systems.

LLM4NG is narrowly optimized for classification tasks in few-shot scenarios, with limited impact on the overall graph structure. When applied to data-limited scenarios rather than strictly few-shot learning, LLM4NG's edge generation methods often introduce significant noise and interference to the graph structure. This limitation is evidenced by its poor performance in our experimental evaluations on data-limited datasets. The edge probabilities predicted by its simple model fail to capture the complex structural patterns present in real-world graphs.

While LLM4NG offers computational efficiency through its lightweight design, it sacrifices synthesis quality, broader applicability, and introduces potential structural inconsistencies. Our experimental results demonstrate that GraphMaster achieves superior performance across multiple datasets and tasks, particularly in generating semantically coherent and structurally valid graph extensions that maintain both local and global properties of the original graph.

# B The implement details of newly created baseline

## B.1 Mixed-LLM

Mixed-LLM introduces a novel interpolative synthesis approach that extends the seminal mixup concept from computer vision to text-attributed graphs via large language models. This method operates on the principle of manifold-aware semantic interpolation, where node representations from different classes are strategically combined to generate semantically coherent yet diverse synthetic nodes.

The Mixed-LLM algorithm consists of three primary phases:

1. **Strategic Class-Balanced Sampling**: Rather than random selection, Mixed-LLM employs a distribution-aware sampling strategy:

$$S = \{(v_i, y_i)|v_i \in V, P(v_i) \propto 1/|V_{y_i}|^{\alpha}\} \tag{23}$$

where $\alpha$ is an adaptive parameter controlling the emphasis on minority classes and $|V_{y_i}|$ represents the cardinality of nodes with label $y_i$.

2. **Latent Space Interpolation**: For each pair of sampled nodes $(v_i, v_j)$, Mixed-LLM generates a convex combination in the semantic space:

$$\tilde{x}_s = \lambda \cdot \phi_{LLM}(x_i) + (1 - \lambda) \cdot \phi_{LLM}(x_j) \tag{24}$$

where $\phi_{LLM}$ represents the LLM's latent representation function and $\lambda \sim \text{Beta}(\alpha, \alpha)$ is a mixing coefficient.

3. **LLM-Guided Textual Manifestation**: The interpolated representation is transformed into coherent textual attributes through a prompt-based generation:

$$x_s = \text{LLM}_M(p(\tilde{x}_s, x_i, x_j, y_i, y_j)) \tag{25}$$

where $p$ is a carefully designed prompt template instructing the LLM to create textual attributes that preserve the semantic characteristics of both source nodes while maintaining linguistic naturalness.

The final label assignment follows a soft probability distribution:

$$P(y_s = c) = \lambda \cdot I[y_i = c] + (1 - \lambda) \cdot I[y_j = c] \tag{26}$$

where $I$ is the indicator function. This probabilistic formulation enables Mixed-LLM to generate boundary-enhancing examples that improve classifier robustness.

## B.2 Synthesis-LLM

Synthesis-LLM implements a context-aware graph sampling and generative synthesis framework that leverages structural locality principles to inform LLM-based node generation. Unlike conventional approaches that process graph data indiscriminately, Synthesis-LLM employs sophisticated topological sampling to create representative subgraph contexts that maximize information density within LLM token constraints.

The framework operates in four sequential stages:

1. **Multi-strategy Subgraph Sampling**: Synthesis-LLM employs a hybrid sampling approach that combines Personalized PageRank (PPR) with strategic breadth-first search:

$$K_s = \Gamma_{PPR}(G, v_s, \alpha, r) \cup \Gamma_{BFS}(G, v_s, d) \tag{27}$$

where $\Gamma_{PPR}$ samples nodes based on their PPR scores from seed node $v_s$ with damping factor $\alpha$ and threshold $r$, while $\Gamma_{BFS}$ complements this with a depth-limited breadth-first expansion to depth $d$.

2. **Structural-Semantic Context Formulation**: The sampled subgraph is transformed into a rich prompt context:

$$C = f_{context}(K_s, A[K_s], X[K_s]) \tag{28}$$

where $f_{context}$ is a specialized function that encodes both topological relationships and textual attributes into a structured prompt format.

3. **Guided Generative Synthesis**: The LLM generates new nodes conditioned on the extracted context:

$$(x_s, E_s) = \text{LLM}_S(C, \theta) \tag{29}$$

where $\text{LLM}_S$ represents the synthesis LLM with temperature parameter $\theta$ that balances creativity and fidelity.

4. **Structural Consistency Enforcement**: Generated nodes undergo topological validation to ensure adherence to the original graph's structural patterns:

$$E'_s = \{e \in E_s | P_{structure}(e|G) > \tau\} \tag{30}$$

where $P_{structure}$ estimates the probability of edge $e$ existing given the structural patterns in $G$, and $\tau$ is an acceptance threshold.

This methodology enables Synthesis-LLM to generate nodes that maintain both semantic relevance and structural coherence with respect to the original graph, while requiring minimal examples due to the LLM's inherent understanding of semantic relationships.

## B.3 Experimental details

We selected QwQ-32B [37] as the large language model for these two baselines, and used two A6000 GPUs with 48G memory for the experiments.

### B.3.1 Hyperparameter Selection for Mixed-LLM

In Mixed-LLM, extensive grid search and ablation studies were conducted to optimize key hyperparameters. The class balancing parameter $\alpha$ was tuned within the range [0.5, 1.5] with an optimal value of 0.8, ensuring a good balance between preserving the original class distribution and addressing class imbalance issues. The beta distribution parameter for the mixing coefficient, where $\lambda \sim \text{Beta}(\alpha, \alpha)$, achieved optimal performance at 0.4, producing meaningful boundary examples. Additionally, an LLM temperature of 0.7 provided the best balance between creative variation and semantic consistency, and incorporating 2–3 example interpolations in the prompt significantly enhanced generation quality.

### B.3.2 Hyperparameter Selection for Synthesis-LLM

For Synthesis-LLM, the hyperparameters were optimized to capture both local and global graph structures. A PPR damping factor $\alpha$ of 0.65 offered a suitable trade-off between local neighborhood exploration and distant node influence, while a PPR threshold $r = 0.005$ effectively identified relevant nodes. A BFS depth limit of $d = 2$ was sufficient to extract essential structural context without overloading the LLM's input. Moreover, setting the generation temperature $\theta$ to 0.5 ensured structural and semantic coherence, and a structural acceptance threshold $\tau$ of 0.6 successfully filtered edge proposals. Overall, the optimal performance was achieved when 25–35 representative nodes were included in the LLM context.

## C  Data-limited Datasets Creation

To simulate realistic scenarios where annotated data is scarce, we generate data-limited datasets by extracting carefully curated subgraphs from the original large-scale graphs. Our procedure begins by partitioning the original graph based on node labels and inherent manifold properties, ensuring that the semantic distribution and community structures are preserved. Next, we apply a multi-objective sampling strategy that leverages node degrees, community representation, and bridge node potentials to select a subset of nodes and their associated edges. This approach, outlined in Algorithm 1, is designed to maintain the essential connectivity patterns and spectral features of the full graph while significantly reducing the number of nodes. Iterative refinements are then performed to balance class proportions and correct any topological distortions, resulting in a smaller yet representative subgraph that closely mimics the original graph's structure and attribute distribution. This data-limited setup provides a robust testbed for evaluating the effectiveness of our graph synthesis methods under constrained conditions.

**Algorithm 1** $\mathcal{M}$-Preserving Graph Sampling

---

**Require:** Graph $G = (V, E, \mathcal{X}, \mathcal{Y}, \mathcal{M})$, sampling ratio $\alpha \in (0, 1]$, convergence threshold $\epsilon$
**Ensure:** Homeomorphic sampled graph $G_s$ preserving manifold properties
1: $\boldsymbol{\Phi}_G \leftarrow \mathcal{T}\langle G \rangle$     ▷ Extract property tensor capturing distributions and spectral features
2: $\mathcal{C} \leftarrow \arg\max_{\mathcal{C}'} \mathcal{Q}(\mathcal{C}', G)$     ▷ Optimize modularity for community detection
3: $\mathcal{D} \leftarrow \{V_{y,m}\}_{y,m}$ where $V_{y,m} = \{v \in V : \mathcal{Y}(v) = y, \mathcal{M}(v) = m\}$ ▷ Create attribute partitions
4: $\Pi \leftarrow \{\pi_{y,m} = |V_{y,m}|/|V|\}_{y,m}$     ▷ Joint distribution tensor
5: $\mathbf{K} \leftarrow \lfloor |V| \cdot \alpha \cdot \Pi \rfloor$     ▷ Target counts per partition
6: $\mathbf{K} \leftarrow \mathbf{K} + \delta(\mathbf{K}, \alpha|V|)$     ▷ Correct sampling counts to exactly match target size
7: $V_s \leftarrow \emptyset$     ▷ Initialize sampled node set
8: **for** $(y, m) \in \{(\mathcal{Y}, \mathcal{M})\}$ **do**
9:    **if** $|V_{y,m}| \leq \mathbf{K}_{y,m}$ **then**
10:      $V_s \leftarrow V_s \cup V_{y,m}$
11:    **else**
12:      $\boldsymbol{\Omega}_{y,m} \leftarrow$ Multi-objective weight vector where for each $v \in V_{y,m}$:

$$\boldsymbol{\Omega}_{y,m}(v) = \lambda_1 \frac{\deg(v)}{\max_u \deg(u)} + \lambda_2(1 - \frac{|V_s \cap \mathcal{C}(v)|}{|\mathcal{C}(v)|}) + \lambda_3 \beta_{\text{bridge}}(v) \tag{31}$$

13:      $V_s \leftarrow V_s \cup \text{TopK}(V_{y,m}, \boldsymbol{\Omega}_{y,m}, \mathbf{K}_{y,m})$
14:    **end if**
15: **end for**
16: $G'_s \leftarrow G[V_s]$     ▷ Initial induced subgraph
17: **if** $\|\kappa(G'_s) - \kappa(G)\| > \epsilon$ **then**     ▷ Check connectivity distortion
18:    $\mathcal{B} \leftarrow$ Bridge nodes $(V \setminus V_s)$ sorted by connectivity gain potential
19:    $\mathcal{R} \leftarrow$ Replaceable nodes in $V_s$ with minimal structural impact
20:    **while** $\|\kappa(G'_s) - \kappa(G)\| > \epsilon$ and $\mathcal{B} \neq \emptyset$ and $\mathcal{R} \neq \emptyset$ **do**
21:      $(b^*, r^*) \leftarrow \arg\max_{b \in \mathcal{B}, r \in \mathcal{R}} \mathcal{S}(b, r)$ subject to $\mathcal{Y}(b) = \mathcal{Y}(r) \wedge \mathcal{M}(b) = \mathcal{M}(r)$
22:      $V_s \leftarrow (V_s \setminus \{r^*\}) \cup \{b^*\}$
23:      $G'_s \leftarrow G[V_s]$
24:      Update $\mathcal{B}, \mathcal{R}$
25:    **end while**
26: **end if**
27: **return** $G'_s$

---

# D   Statistics of the Datasets

Table 3: Dataset Statistics

| Dataset | # Nodes | # Edges | # Classes | # Louvain communities | # Training nodes | # Validation nodes | # Test nodes |
|---|---|---|---|---|---|---|---|
| Cora | 2708 | 5278 | 7 | 106 | 1624 | 542 | 542 |
| Citeseer | 3186 | 4225 | 6 | 506 | 1911 | 637 | 638 |
| Wikics | 8196 | 104161 | 10 | 540 | 580 | 1769 | 5847 |
| History | 41551 | 251590 | 12 | 2036 | 24921 | 8337 | 8293 |
| Arxiv2023 | 46198 | 38863 | 38 | 28901 | 28905 | 27718 | 9240 |
| Children | 76875 | 1162522 | 24 | 2296 | 46010 | 15455 | 15410 |
| SubCora | 1354 | 2486 | 7 | 99 | 815 | 267 | 272 |
| SubCiteseer | 1274 | 1360 | 6 | 486 | 764 | 255 | 255 |
| SubWikics | 1639 | 26786 | 10 | 374 | 111 | 350 | 1178 |
| SubHistory | 2077 | 40415 | 12 | 17 | 1249 | 416 | 412 |
| SubArxiv2023 | 6929 | 3297 | 38 | 5398 | 4174 | 1375 | 1380 |
| SubChildren | 3843 | 94636 | 24 | 71 | 2308 | 766 | 769 |

Table 3 shows the basic characteristics of our new synthesized dataset. We introduce our dataset from seven aspects: Nodes, Edges, Classes, Louvain communities, Training nodes, Validation nodes and Test nodes, including the original dataset and our newly generated Data-limited dataset.

# E Hyperparameter Selection Analysis

We conducted comprehensive grid search experiments to determine optimal hyperparameter settings for GraphMaster. Our analysis reveals that the framework is robust to moderate parameter variations, with the following configuration yielding consistently strong performance across datasets:

- **Knowledge extraction:** Sample size $N = 30$ nodes provides sufficient context without introducing noise
- **Node generation:** Setting $M\% = 15\%$ of knowledge nodes balances quantity and quality
- **Community detection:** Parameters $\mu = 0.5$ and $\gamma = 0.5$ effectively balance semantic and structural factors
- **Stochastic sampling:** $\beta = 2.0$ maintains appropriate exploration-exploitation balance
- **Edge formation:** For semantic mode, $(\theta_1, \theta_2, \theta_3) = (0.6, 0.3, 0.1)$; for topological mode, $(0.2, 0.5, 0.3)$
- **Quality assessment:** Initial threshold $\tau_0 = 7.0$ with adaptive update rate $\zeta = 0.1$
- **Convergence criteria:** $\epsilon = 0.05$ provides sufficient refinement iterations
- **Objective weights:** Initialize $\lambda_{\text{sem}} = \lambda_{\text{struct}} = \lambda_{\text{bal}} = 0.33$ with learning rate $\eta = 0.05$

Among tested LLMs (QwQ-32B, Qwen-32B, DeepSeek-R1-32B, and LLaMA-33B), QwQ-32B consistently delivered superior performance. We limited synthesis to a maximum of 15 iterations, as additional iterations yielded diminishing returns..

# F Detailed Experimental Results

Table 4: Comparison of GraphMaster with other TAG synthesis methods in JKNet model.

| Type | Model | Cora | | Citeseer | | Wikics | | History | | Arxiv2023 | | Children | |
|---|---|---|---|---|---|---|---|---|---|---|---|---|---|
| | | Acc | F1 | Acc | F1 | Acc | F1 | Acc | F1 | Acc | F1 | Acc | F1 |
| Original | Origin | 88.9±1.1 | 87.9±0.7 | 78.3±0.9 | 75.7±1.2 | 79.7±0.6 | 77.9±1.1 | 84.2±0.3 | 43.1±1.3 | 76.3±0.8 | 54.9±1.2 | 52.6±0.5 | 31.9±0.9 |
| Classic-Aug | GAugO | 88.9±0.4 | 88.2±1.2 | 78.3±1.4 | 77.0±0.8 | 80.0±0.7 | 77.8±1.0 | 84.6±1.2 | 44.7±1.4 | 76.8±1.0 | 53.0±1.2 | 51.8±0.8 | 33.6±1.4 |
| LLM-Aug | GraphEdit | 91.0±1.3 | 89.4±1.0 | 81.4±0.7 | 80.2±0.8 | 82.0±0.9 | 80.6±0.7 | 87.6±1.0 | 45.7±1.1 | 78.0±0.4 | 57.8±1.1 | 54.3±1.3 | 35.7±0.4 |
| | LLM4RGNN | 91.4±0.9 | 88.8±0.5 | 81.0±1.4 | 76.7±0.7 | 83.6±0.3 | 81.4±0.5 | 88.9±0.8 | 48.6±1.4 | 79.3±1.2 | 59.1±1.4 | 55.7±0.7 | 36.7±1.1 |
| Classic-Syn | GraphSmote | 88.7±0.4 | 87.4±0.8 | 78.2±1.4 | 74.8±0.9 | 80.7±0.3 | 78.5±0.8 | 84.9±0.8 | 43.9±1.4 | 76.2±1.0 | 55.5±1.3 | 53.1±0.7 | 33.2±0.5 |
| | G-Mixup | 87.4±1.0 | 87.0±0.5 | 78.4±0.3 | 76.9±1.0 | 79.7±1.0 | 78.0±1.3 | 84.6±1.1 | 43.6±1.2 | 76.6±1.2 | 56.5±0.8 | 53.0±1.1 | 33.0±0.3 |
| | IntraMix | 80.9±1.4 | 82.9±1.0 | 71.4±0.4 | 70.7±1.4 | 73.7±1.3 | 74.4±0.5 | 82.4±1.3 | 42.7±0.4 | 72.4±1.0 | 53.9±1.2 | 45.2±1.0 | 30.1±0.5 |
| | GraphAdasyn | 89.2±1.3 | 88.8±1.2 | 78.7±1.1 | 78.2±1.3 | 80.8±0.6 | 78.8±1.1 | 84.6±1.3 | 46.1±0.3 | 77.5±1.4 | 57.0±1.3 | 53.6±0.5 | 33.0±1.2 |
| | FG-SMOTE | 88.9±0.6 | 87.6±0.5 | 78.6±0.7 | 74.7±0.4 | 81.0±1.4 | 79.0±1.1 | 85.0±0.9 | 44.0±0.8 | 76.4±0.8 | 55.8±0.7 | 53.1±1.3 | 33.3±0.9 |
| | AGMixup | 84.7±0.7 | 86.6±1.3 | 71.6±1.2 | 73.2±1.3 | 78.8±1.2 | 76.6±0.4 | 81.8±0.9 | 42.9±1.4 | 76.8±1.2 | 53.7±1.3 | 53.6±1.1 | 32.6±1.0 |
| LLM-Syn | GAG | 91.0±1.4 | 89.3±1.1 | 82.8±0.7 | 80.0±0.5 | 84.9±1.1 | 83.2±1.2 | 88.9±0.9 | 49.8±1.3 | 79.9±1.4 | 59.4±1.2 | 56.7±1.0 | 38.0±1.3 |
| | LLM4NG | 80.9±0.4 | 79.7±0.3 | 87.1±0.6 | 85.6±0.3 | 75.0±0.3 | 74.5±0.5 | 84.2±0.6 | 52.4±0.4 | 81.3±0.2 | 62.5±0.4 | 49.3±0.6 | 37.0±0.8 |
| | Mixed-LLM | 89.9±0.3 | 89.3±1.0 | 83.4±1.3 | 81.3±1.2 | 84.9±1.1 | 83.4±1.3 | 89.2±1.1 | 55.8±0.9 | 81.4±1.3 | 61.2±0.9 | 60.0±1.3 | 39.6±0.8 |
| | Synthesis-LLM | 89.8±1.1 | 89.1±0.5 | 84.5±1.2 | 82.7±0.5 | 84.8±0.8 | 83.2±1.4 | 89.4±0.4 | 53.4±0.8 | 81.0±1.3 | 62.3±1.1 | 60.9±1.3 | 40.1±0.4 |
| | **GraphMaster** | **93.9±1.2** | **93.6±0.8** | **89.0±1.3** | **87.8±1.2** | **87.9±0.9** | **86.4±1.3** | **92.5±0.7** | **64.1±1.1** | **87.8±1.0** | **66.4±1.4** | **68.9±1.4** | **47.7±0.5** |

Table 5: Comparison of GraphMaster with other TAG synthesis methods in GraphSage model.

| Type | Model | Cora | | Citeseer | | Wikics | | History | | Arxiv2023 | | Children | |
|---|---|---|---|---|---|---|---|---|---|---|---|---|---|
| | | Acc | F1 | Acc | F1 | Acc | F1 | Acc | F1 | Acc | F1 | Acc | F1 |
| Original | Origin | 88.4±0.9 | 87.4±1.2 | 78.4±0.8 | 74.7±0.7 | 80.2±1.3 | 77.6±0.5 | 84.1±0.4 | 42.6±1.1 | 76.2±0.6 | 53.7±0.9 | 52.6±1.2 | 30.9±0.7 |
| Classic-Aug | GAugO | 87.9±1.0 | 87.9±0.5 | 79.1±1.3 | 76.0±0.8 | 79.6±0.7 | 77.4±1.1 | 83.7±0.9 | 43.6±0.6 | 76.8±1.4 | 55.3±0.4 | 52.4±0.8 | 30.9±1.0 |
| LLM-Aug | GraphEdit | 91.6±0.6 | 91.0±1.2 | 83.0±0.9 | 80.9±0.3 | 83.8±1.1 | 80.5±0.6 | 87.6±1.3 | 48.7±0.5 | 78.2±0.4 | 56.8±1.0 | 56.1±0.7 | 34.6±1.2 |
| | LLM4RGNN | 88.8±0.3 | 87.3±1.4 | 79.5±0.5 | 78.7±1.0 | 83.6±0.9 | 80.5±0.7 | 87.0±0.8 | 47.6±1.3 | 77.9±1.1 | 56.2±0.5 | 55.4±1.4 | 33.7±0.9 |
| Classic-Syn | GraphSmote | 87.9±1.1 | 87.6±0.8 | 79.6±1.2 | 76.6±0.5 | 80.0±0.6 | 77.7±0.9 | 83.7±1.0 | 44.7±0.4 | 77.3±1.3 | 55.8±1.2 | 53.8±0.7 | 33.5±0.6 |
| | G-Mixup | 87.7±0.5 | 87.8±1.3 | 78.2±0.7 | 75.3±1.1 | 79.9±1.2 | 77.5±1.4 | 85.4±0.9 | 44.2±1.1 | 76.4±0.7 | 54.9±0.8 | 53.6±1.2 | 33.3±0.9 |
| | IntraMix | 81.1±0.8 | 81.6±0.4 | 71.1±1.1 | 70.2±0.9 | 73.8±0.5 | 74.2±1.3 | 82.2±0.7 | 42.2±0.8 | 72.2±1.0 | 53.3±0.3 | 45.0±0.9 | 31.7±1.0 |
| | GraphAdasyn | 90.0±1.3 | 90.1±0.5 | 79.4±0.9 | 77.4±1.3 | 84.3±0.4 | 81.6±0.8 | 86.7±1.2 | 45.4±0.6 | 77.8±0.8 | 56.3±1.1 | 56.4±0.3 | 35.1±0.5 |
| | FG-SMOTE | 88.0±0.6 | 88.0±1.0 | 79.8±0.4 | 76.8±0.7 | 80.3±1.3 | 77.8±0.5 | 85.8±0.9 | 44.8±1.2 | 77.6±0.6 | 56.2±0.8 | 54.3±1.1 | 33.6±0.4 |
| | AGMixup | 87.7±0.9 | 87.3±0.3 | 78.1±1.3 | 74.5±0.8 | 81.5±0.7 | 78.6±1.1 | 84.3±0.5 | 42.5±0.7 | 74.9±1.2 | 52.1±0.9 | 53.5±0.4 | 31.5±1.1 |
| LLM-Syn | GAG | 91.2±0.7 | 90.5±1.1 | 81.4±0.8 | 80.6±0.5 | 85.9±1.0 | 82.4±0.6 | 88.9±1.3 | 48.7±0.9 | 78.7±0.3 | 57.2±1.4 | 57.7±0.8 | 36.9±0.5 |
| | LLM4NG | 82.0±0.2 | 81.2±0.4 | 84.3±0.5 | 83.1±0.5 | 80.0±0.3 | 77.5±0.5 | 81.8±0.6 | 45.6±0.7 | 85.0±0.2 | 62.1±0.1 | 46.2±0.5 | 26.4±0.9 |
| | Mixed-LLM | 91.4±1.2 | 90.7±0.6 | 84.1±0.9 | 84.7±1.3 | 83.7±0.4 | 81.0±0.8 | 90.2±0.7 | 58.3±1.1 | 82.7±1.0 | 59.6±0.4 | 62.3±1.3 | 42.5±0.7 |
| | Synthesis-LLM | 91.3±0.4 | 90.2±1.0 | 84.3±1.2 | 85.2±0.7 | 84.8±0.9 | 82.1±0.5 | 90.5±0.6 | 57.4±1.4 | 83.2±0.8 | 58.5±1.2 | 63.4±0.5 | 43.1±0.9 |
| | **GraphMaster** | **93.9±0.5** | **92.7±1.1** | **88.9±0.7** | **87.9±0.9** | **87.5±1.2** | **86.4±0.6** | **92.9±0.8** | **62.3±1.0** | **87.4±0.5** | **66.2±1.3** | **66.9±0.7** | **47.1±1.1** |

Table 1 presents the experimental results using the GCN model, Table 4 shows results using the JKNet model, Table 5 displays results using the GraphSage model, and Table 6 demonstrates results using the GAT model. Our findings indicate that GraphMaster consistently outperforms other approaches across all GNN architectures, strongly validating the universality of our proposed model. Furthermore,

Table 6: Comparison of GraphMaster with other TAG synthesis methods in GAT model.

| Type | Model | Cora | | Citeseer | | Wikics | | History | | Arxiv2023 | | Children | |
|---|---|---|---|---|---|---|---|---|---|---|---|---|---|
| | | Acc | F1 | Acc | F1 | Acc | F1 | Acc | F1 | Acc | F1 | Acc | F1 |
| Original | Origin | 85.9±0.8 | 85.0±0.8 | 75.9±0.8 | 72.6±0.9 | 77.9±0.6 | 75.9±0.7 | 81.8±0.9 | 46.7±0.8 | 74.3±0.6 | 52.1±1.0 | 51.0±0.7 | 30.0±0.9 |
| Classic-Aug | GAugO | 85.6±0.7 | 84.7±0.6 | 76.1±0.7 | 73.8±0.7 | 78.6±1.0 | 75.3±1.1 | 82.1±0.7 | 42.6±0.6 | 75.3±1.0 | 53.7±0.6 | 51.8±1.1 | 30.0±0.6 |
| LLM-Aug | GraphEdit | 88.2±0.6 | 87.0±0.9 | 79.8±0.9 | 78.4±1.1 | 81.4±0.8 | 78.3±0.6 | 85.1±1.0 | 47.3±1.1 | 76.0±0.7 | 55.0±0.9 | 54.6±0.6 | 33.5±1.0 |
| | LLM4RGNN | 87.8±0.5 | 85.3±0.7 | 79.3±0.6 | 76.4±0.6 | 81.2±0.5 | 78.1±0.9 | 85.3±0.6 | 46.2±0.7 | 76.3±1.2 | 54.4±0.7 | 54.0±0.9 | 32.7±0.7 |
| Classic-Syn | GraphSmote | 85.8±0.7 | 84.4±0.5 | 76.9±1.2 | 74.3±0.8 | 78.3±1.1 | 75.6±0.5 | 82.3±1.2 | 43.6±0.9 | 75.5±0.5 | 54.3±1.1 | 52.2±0.5 | 32.5±1.1 |
| | G-Mixup | 84.8±0.6 | 84.3±0.8 | 76.1±0.8 | 73.2±1.2 | 77.5±0.7 | 75.4±1.0 | 82.5±0.8 | 43.4±0.5 | 74.0±0.9 | 53.4±0.5 | 51.9±1.2 | 31.9±0.5 |
| | IntraMix | 78.4±0.9 | 79.5±0.6 | 69.0±0.5 | 68.3±0.5 | 71.9±0.9 | 72.1±0.7 | 79.9±0.5 | 41.3±1.2 | 71.5±0.7 | 51.9±0.8 | 43.9±0.7 | 30.7±0.8 |
| | GraphAdasyn | 86.8±0.8 | 86.3±0.9 | 77.3±1.0 | 75.1±0.9 | 82.0±0.6 | 78.0±1.2 | 83.7±1.1 | 44.3±0.7 | 75.6±1.1 | 54.7±1.2 | 54.8±1.0 | 34.0±1.2 |
| | FG-SMOTE | 85.9±0.5 | 84.6±0.7 | 77.2±0.7 | 74.2±0.7 | 78.9±1.2 | 76.0±0.5 | 83.3±0.7 | 43.8±1.0 | 75.5±0.6 | 54.8±0.6 | 52.8±0.6 | 32.5±0.7 |
| | AGMixup | 83.0±0.7 | 83.7±0.5 | 75.0±1.1 | 72.6±1.0 | 79.6±0.5 | 77.0±0.9 | 82.3±0.9 | 41.6±0.6 | 73.7±0.8 | 51.5±0.9 | 51.8±0.9 | 30.5±0.9 |
| LLM-Syn | GAG | 88.3±0.8 | 87.0±0.8 | 80.5±0.6 | 78.3±0.8 | 83.5±0.9 | 80.0±0.7 | 86.3±0.6 | 47.3±0.9 | 77.1±1.2 | 55.7±0.7 | 56.2±0.7 | 35.8±0.6 |
| | LLM4NG | 77.6±0.2 | 76.3±0.4 | 69.4±0.7 | 67.9±0.5 | 77.9±0.3 | 76.6±0.5 | 82.0±0.4 | 41.7±0.7 | 61.2±0.4 | 51.4±0.7 | 46.9±0.4 | 27.2±0.5 |
| | Mixed-LLM | 88.2±0.6 | 87.2±0.7 | 81.4±0.9 | 81.8±0.6 | 81.9±0.7 | 78.7±1.1 | 87.6±1.0 | 57.0±0.7 | 80.1±0.5 | 57.9±1.0 | 60.7±1.1 | 41.2±1.0 |
| | Synthesis-LLM | 88.1±0.9 | 87.0±0.6 | 81.6±0.7 | 82.3±1.1 | 82.3±1.0 | 79.5±0.6 | 87.8±0.5 | 55.7±1.1 | 80.8±0.9 | 56.8±0.5 | 61.9±0.5 | 41.8±0.8 |
| | **GraphMaster** | **89.9±0.7** | **88.6±0.9** | **85.0±1.0** | **84.3±0.7** | **84.2±0.8** | **82.7±0.8** | **89.0±0.9** | **59.1±0.8** | **83.7±0.7** | **63.0±0.9** | **63.9±0.8** | **44.5±0.6** |

we observe that our two novel baselines frequently achieve second-place rankings, which substantiates the significant potential of Large Language Models (LLMs) in text-attributed graph data synthesis. As the first work leveraging LLMs for text-attributed graph synthesis, GraphMaster exhibits both efficient resource utilization and remarkable capabilities for TAG data generation.

# G  Detailed Graph Feature Analysis

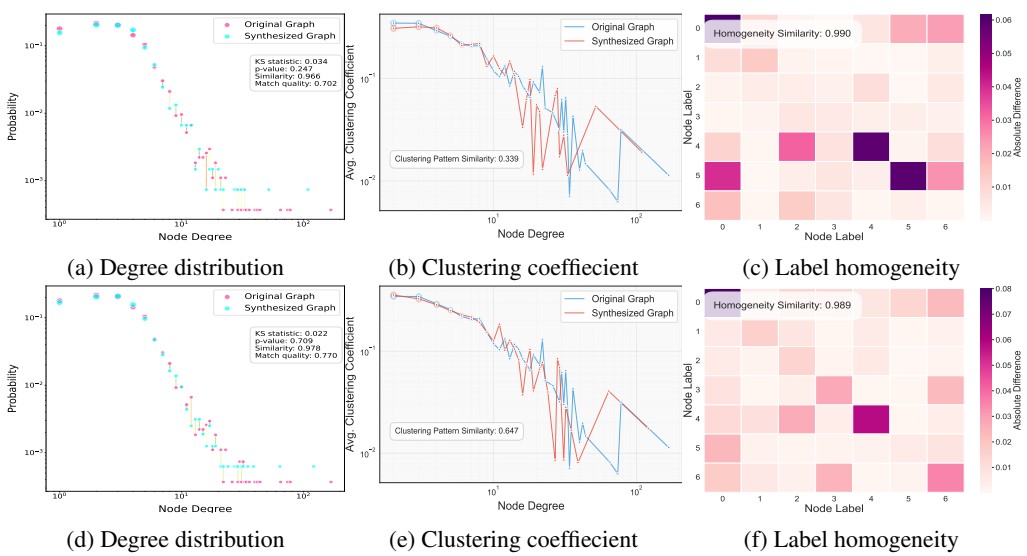

(a) Degree distribution    (b) Clustering coeffiecient    (c) Label homogeneity

(d) Degree distribution    (e) Clustering coeffiecient    (f) Label homogeneity

Figure 4: Graph feature analysis on Cora dataset. The top three rows of pictures are the results of the original data-limited dataset, and the bottom three rows are the results after TAG data synthesis using GraphMaster.

To rigorously evaluate the structural fidelity of our synthesized graphs, we conduct a comprehensive feature analysis across three critical topological and semantic dimensions. Figure 4, Figure 5, Figure 6, Figure 7, Figure 8 and Figure 9 present comparative analyses of the original graphs versus those synthesized by GraphMaster on Cora, Citeseer, Wikics, History, Arxiv2023, and Children datasets, respectively.

Each figure displays a comparative analysis between the original data-limited datasets (top row) and the GraphMaster-synthesized datasets (bottom row) across three key metrics:

- **Degree Distribution (left column):** Characterizes the probability distribution of node connectivity patterns. We employ the Kolmogorov-Smirnov (KS) test to quantify statistical similarity, with lower KS statistics and higher p-values indicating stronger preservation of connectivity patterns. Our results demonstrate that GraphMaster consistently improves

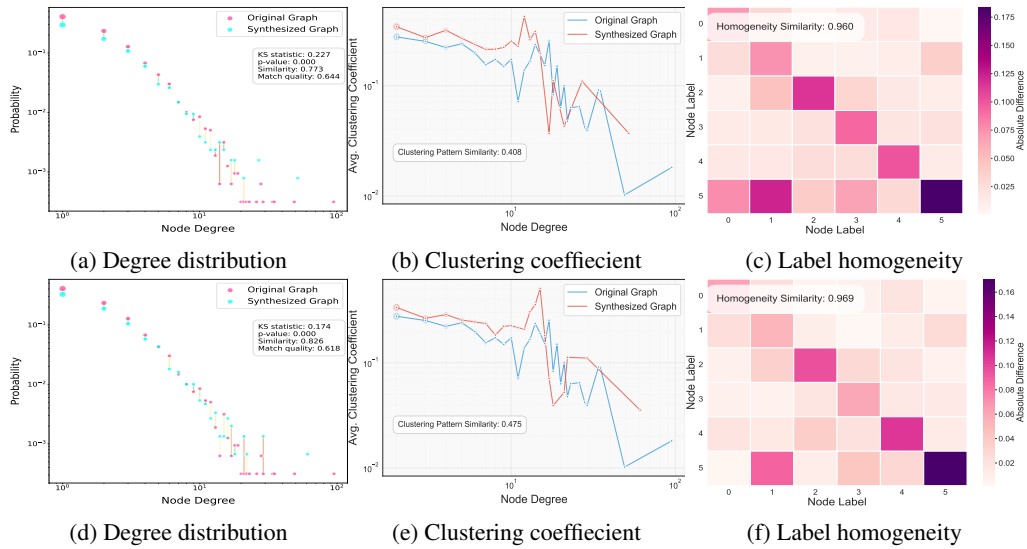

(a) Degree distribution      (b) Clustering coeffiecient      (c) Label homogeneity

(d) Degree distribution      (e) Clustering coeffiecient      (f) Label homogeneity

Figure 5: Graph feature analysis on Citeseer dataset. The top three rows of pictures are the results of the original data-limited dataset, and the bottom three rows are the results after TAG data synthesis using GraphMaster.

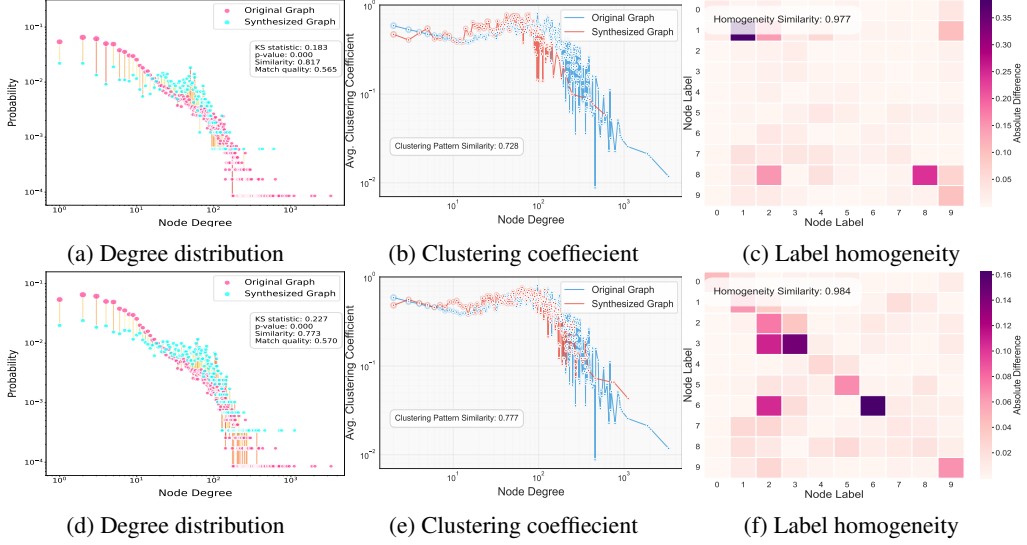

(a) Degree distribution      (b) Clustering coeffiecient      (c) Label homogeneity

(d) Degree distribution      (e) Clustering coeffiecient      (f) Label homogeneity

Figure 6: Graph feature analysis on Wikics dataset. The top three rows of pictures are the results of the original data-limited dataset, and the bottom three rows are the results after TAG data synthesis using GraphMaster.

degree distribution similarity across all datasets, with Cora showing particularly strong results (KS statistic of 0.022, p-value=0.709).

- **Clustering Coefficient vs. Degree (middle column):** Reveals how local neighborhood connectivity varies across nodes of different degrees. We observe that GraphMaster substantially improves clustering pattern similarity in most datasets, with notable improvements in Wikics (from 0.728 to 0.777) and Children (from 0.785 to 0.835). This indicates that our synthesis approach effectively captures the relationship between node importance and community formation.

- **Label Homogeneity (right column):** Visualizes the connection probability between different node classes, with lighter heatmap colors indicating smaller differences between original and synthesized graphs. GraphMaster achieves remarkably high label homogeneity simi-

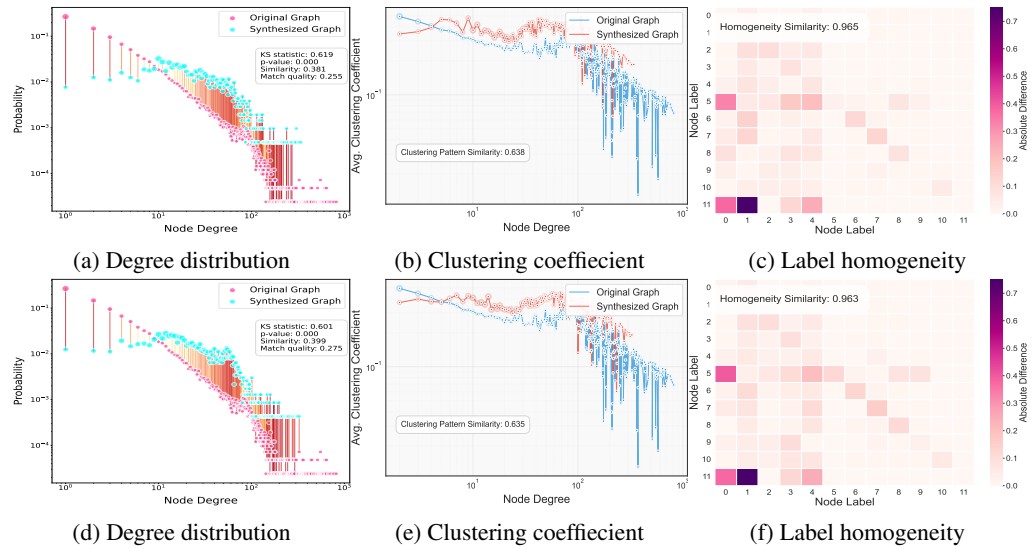

Figure 7: Graph feature analysis on History dataset. The top three rows of pictures are the results of the original data-limited dataset, and the bottom three rows are the results after TAG data synthesis using GraphMaster.

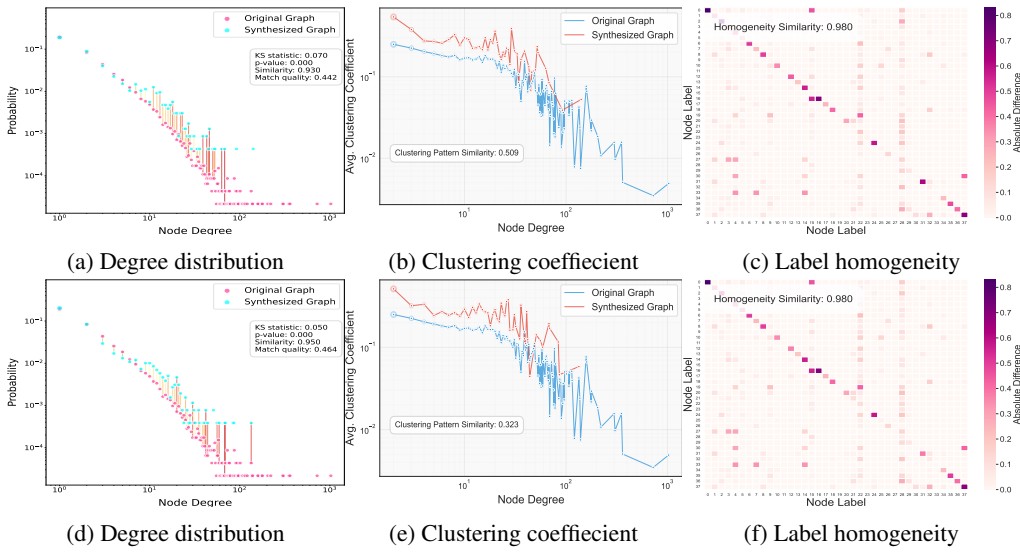

Figure 8: Graph feature analysis on Arxiv2023 dataset. The top three rows of pictures are the results of the original data-limited dataset, and the bottom three rows are the results after TAG data synthesis using GraphMaster.

larity scores (>0.96 across all datasets), demonstrating its ability to preserve label-to-label connectivity patterns.

Notably, our approach shows significant structural improvements on citation networks (Cora, Citeseer, Arxiv2023) where semantic relationships strongly influence topology. For instance, in Arxiv2023, GraphMaster improves degree distribution similarity from 0.930 to 0.950 while maintaining consistent label homogeneity (0.980). On larger and more complex datasets like Children, GraphMaster effectively preserves clustering coefficients (similarity improvement from 0.785 to 0.835) while maintaining degree distribution and label homogeneity.

These results collectively demonstrate that GraphMaster not only enhances semantic richness but also successfully preserves—and in many cases improves—the critical structural characteristics of the

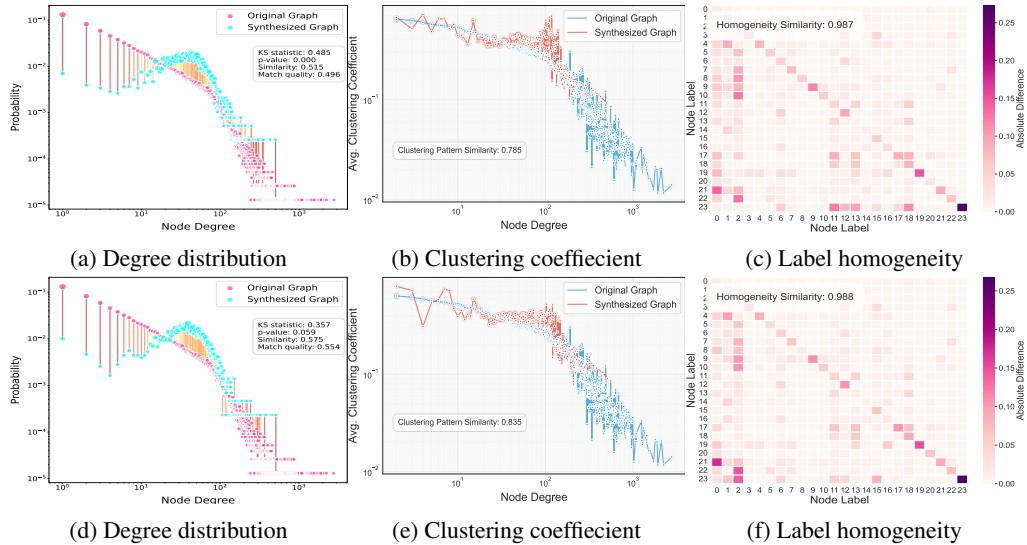

| (a) Degree distribution | (b) Clustering coeffiecient | (c) Label homogeneity |
| (d) Degree distribution | (e) Clustering coeffiecient | (f) Label homogeneity |

Figure 9: Graph feature analysis on Children dataset. The top three rows of pictures are the results of the original data-limited dataset, and the bottom three rows are the results after TAG data synthesis using GraphMaster.

original graphs. This structural fidelity is essential for ensuring that downstream GNN applications trained on synthesized data generalize effectively to real-world scenarios.

# H  Theoretical Analysis of Interpretable Experiments

Recent advances evaluation frameworks have emphasized the importance of interpretability when evaluating language model outputs after knowledge edits [23] and other fields [6]. Inspired by this, our interpretability analysis similarly integrates both human-centered and algorithmic assessments to evaluate the semantic coherence of synthesized graph components.

## H.1  Human-Centered Interpretability Evaluation

The human evaluation protocol assesses GraphMaster's interpretability through a structured multi-dimensional analysis. We recruited $R = 50$ expert annotators with backgrounds in graph machine learning, natural language processing, or data mining. Inclusion criteria required either (i) at least two peer-reviewed publications in the past three years, or (ii) a minimum of two years of relevant industrial R&D experience. All participants with potential conflicts of interest were excluded. The final cohort consisted of 38 PhD students, 6 postdoctoral researchers or research assistants, 3 faculty members, and 3 industry researchers. Geographical distribution included Asia (50%), Europe (20%), North America (24%), and others (6%).

Each expert evaluated $N = 200$ synthesized instances sampled from six benchmark datasets. Each instance was independently scored by three reviewers, resulting in 600 total annotations. The evaluation covered three critical dimensions:

- **Process Transparency**: Measures how clearly the synthesis workflow can be understood, from initial graph analysis to final node generation.
- **Decision Justification**: Evaluates whether the model's choices (e.g., which nodes to sample, what attributes to generate) have clear rationales.
- **Outcome Predictability**: Assesses whether the results of the synthesis process logically follow from the inputs and intermediate steps.

Each dimension was rated using a 1–5 Likert scale, which was linearly normalized to $[0, 1]$ for scoring purposes. For each dimension $d$, reviewer $r$ assigns a score $t_{r,i,d} \in [0, 1]$ for instance $i$. The composite interpretability score for instance $i$ by reviewer $r$ is:

$$t_{r,i} = \frac{1}{3} \sum_{d=1}^{3} t_{r,i,d} \qquad (32)$$

To ensure annotation quality, reviewers completed a 30-minute training session followed by a 20-item calibration phase. Only those achieving a Cohen's $\kappa \geq 0.70$ in the pilot study were retained. During the main annotation phase, we adopted a double-blind setup with randomized item and order presentation, applied a standardized rubric, and removed two raters who failed attention checks. The average pairwise inter-rater agreement achieved was $\kappa = 0.72 \pm 0.05$, indicating moderate to substantial agreement.

To assess the statistical reliability of the aggregated Traceability Score, we compute a bootstrap confidence interval:

$$CI(T_{\text{score}}) = \left[ T_{\text{score}} - z_{\alpha/2} \sqrt{\frac{\sigma_T^2}{R \cdot N}}, \ T_{\text{score}} + z_{\alpha/2} \sqrt{\frac{\sigma_T^2}{R \cdot N}} \right] \qquad (33)$$

where $\sigma_T^2$ denotes the variance of individual instance-level ratings and $z_{\alpha/2}$ is the critical value associated with the desired confidence level.

We provide complete details of the reviewer selection, demographic statistics, annotation procedures, and inter-rater reliability metrics in the appendix.

## H.2 Grassmannian Analysis of Semantic Consistency

### H.2.1 Theoretical Foundation

**Definition 1** (Grassmann Manifold). *The Grassmann manifold $\mathcal{G}(p,d)$ is the set of all $p$-dimensional linear subspaces of $\mathbb{R}^d$. Each point on $\mathcal{G}(1,d)$ can be represented by a unit vector $\mathbf{u} \in \mathbb{S}^{d-1}$ (up to sign).*

**Theorem 1** (Principal Semantic Direction). *Given a set of semantically related unit-normalized text embeddings $\{\mathbf{x}_1, \mathbf{x}_2, \ldots, \mathbf{x}_K\} \subset \mathbb{S}^{d-1}$, there exists an optimal direction $\mathbf{u}^* \in \mathbb{S}^{d-1}$ that minimizes the sum of squared geodesic distances on the Grassmann manifold:*

$$\mathbf{u}^* = \underset{\mathbf{u} \in \mathbb{S}^{d-1}}{\arg\min} \sum_{j=1}^{K} \arccos^2(|\mathbf{u}^T \mathbf{x}_j|) \qquad (34)$$

**Proposition 1** (Semantic Coherence Metric). *For a synthesized node with embedding $\mathbf{x}_s$, its semantic coherence with respect to background knowledge is quantified by:*

$$S(\mathbf{x}_s) = 1 - \frac{2}{\pi} \arccos(|\mathbf{x}_s^T \mathbf{u}^*|) \qquad (35)$$

*where $S(\mathbf{x}_s) \in [0,1]$ with higher values indicating greater semantic coherence.*

*Proof of Theorem 1 (Principal Semantic Direction).* Let us first establish that the geodesic distance between two points on $\mathcal{G}(1,d)$ represented by unit vectors $\mathbf{u}$ and $\mathbf{v}$ is given by $d_{\mathcal{G}}(\mathbf{u}, \mathbf{v}) = \arccos(|\mathbf{u}^T \mathbf{v}|)$.

Consider the objective function $f(\mathbf{u}) = \sum_{j=1}^{K} \arccos^2(|\mathbf{u}^T \mathbf{x}_j|)$. We need to show that:

1. This function has at least one global minimum on $\mathbb{S}^{d-1}$

2. This minimum represents a semantically meaningful direction

For the first point, note that $f$ is continuous on the compact set $\mathbb{S}^{d-1}$, so by the extreme value theorem, it attains both a maximum and minimum value.

For the second point, let us analyze the critical points of $f$ constrained to $\mathbb{S}^{d-1}$. Using the method of Lagrange multipliers, we seek critical points of:

$$\mathcal{L}(\mathbf{u}, \lambda) = \sum_{j=1}^{K} \arccos^2(|\mathbf{u}^T \mathbf{x}_j|) - \lambda(\mathbf{u}^T \mathbf{u} - 1) \tag{36}$$

Taking the gradient with respect to $\mathbf{u}$:

$$\nabla_{\mathbf{u}} \mathcal{L} = \sum_{j=1}^{K} -2 \arccos(|\mathbf{u}^T \mathbf{x}_j|) \cdot \frac{1}{\sqrt{1 - (|\mathbf{u}^T \mathbf{x}_j|)^2}} \cdot \operatorname{sgn}(\mathbf{u}^T \mathbf{x}_j) \cdot \mathbf{x}_j - 2\lambda \mathbf{u} = \mathbf{0} \tag{37}$$

Solving this system of equations is equivalent to finding $\mathbf{u}$ that balances the weighted contributions of all $\mathbf{x}_j$ vectors. The solution $\mathbf{u}^*$ represents a direction that minimizes angular deviation from all input embeddings.

To demonstrate semantic meaningfulness, let us decompose any embedding $\mathbf{x}_j$ as:

$$\mathbf{x}_j = (\mathbf{x}_j^T \mathbf{u}^*)\mathbf{u}^* + \mathbf{x}_j^{\perp} \tag{38}$$

where $\mathbf{x}_j^{\perp}$ is orthogonal to $\mathbf{u}^*$. The optimization objective minimizes the magnitude of these orthogonal components across all embeddings, effectively capturing their common directional component in the embedding space. Since semantically related concepts tend to cluster directionally in embedding spaces, $\mathbf{u}^*$ represents their central semantic direction. □

*Proof of Proposition 1 (Semantic Coherence Metric).* The geodesic distance between $\mathbf{x}_s$ and the principal direction $\mathbf{u}^*$ on $\mathcal{G}(1, d)$ is $d_{\mathcal{G}}(\mathbf{x}_s, \mathbf{u}^*) = \arccos(|\mathbf{x}_s^T \mathbf{u}^*|)$. This distance ranges from 0 (perfect alignment) to $\pi/2$ (orthogonality).

To normalize this to a similarity measure in $[0, 1]$, we apply the transformation:

$$S(\mathbf{x}_s) = 1 - \frac{d_{\mathcal{G}}(\mathbf{x}_s, \mathbf{u}^*)}{\pi/2} = 1 - \frac{2}{\pi} \arccos(|\mathbf{x}_s^T \mathbf{u}^*|) \tag{39}$$

This yields $S(\mathbf{x}_s) = 1$ when $\mathbf{x}_s$ and $\mathbf{u}^*$ are perfectly aligned (modulo sign), and $S(\mathbf{x}_s) = 0$ when they are orthogonal.

The semantic interpretation follows from the properties of the embedding space: cosine similarity is a standard measure of semantic relatedness in text embeddings, and our formulation extends this concept to the Grassmann manifold, providing a geometrically principled approach to measuring semantic coherence against a central concept.

Moreover, this metric satisfies several desirable properties:

- **Invariance to sign**: $S(\mathbf{x}_s) = S(-\mathbf{x}_s)$, reflecting that oppositely directed vectors represent the same point on $\mathcal{G}(1, d)$

- **Monotonicity**: $S(\mathbf{x}_s)$ increases as $\mathbf{x}_s$ aligns more closely with $\mathbf{u}^*$

- **Bounded range**: $S(\mathbf{x}_s) \in [0, 1]$, facilitating interpretation

- **Geometric meaning**: Directly related to the principal angle between subspaces on the Grassmann manifold

□

**Theorem 2** (Computational Solution). *The principal direction $\mathbf{u}^*$ can be efficiently computed as the principal eigenvector of the matrix:*

$$\mathbf{M} = \sum_{j=1}^{K} w_j \mathbf{x}_j \mathbf{x}_j^T \tag{40}$$

*where weights $w_j$ are iteratively updated based on angular distances.*

*Proof.* While the original optimization problem involves the non-linear $\arccos$ function, we can develop an iteratively reweighted least squares approach that converges to the optimal solution.

Consider a simplified objective function $g(\mathbf{u}) = \sum_{j=1}^{K} w_j(1 - (\mathbf{u}^T \mathbf{x}_j)^2)$ where $w_j = \arccos(|\mathbf{u}_t^T \mathbf{x}_j|)/\sqrt{1 - (|\mathbf{u}_t^T \mathbf{x}_j|)^2}$ at iteration $t$.

Expanding $g(\mathbf{u})$:

$$g(\mathbf{u}) = \sum_{j=1}^{K} w_j - \sum_{j=1}^{K} w_j (\mathbf{u}^T \mathbf{x}_j)^2 \tag{41}$$

$$= \sum_{j=1}^{K} w_j - \mathbf{u}^T \left( \sum_{j=1}^{K} w_j \mathbf{x}_j \mathbf{x}_j^T \right) \mathbf{u} \tag{42}$$

$$= \sum_{j=1}^{K} w_j - \mathbf{u}^T \mathbf{M} \mathbf{u} \tag{43}$$

Since $\mathbf{u}$ is constrained to have unit norm, minimizing $g(\mathbf{u})$ is equivalent to maximizing $\mathbf{u}^T \mathbf{M} \mathbf{u}$. By the Rayleigh-Ritz theorem, this is maximized when $\mathbf{u}$ is the eigenvector corresponding to the largest eigenvalue of $\mathbf{M}$.

The algorithm proceeds as follows:

1. Initialize $\mathbf{u}_0$ as a random unit vector
2. At iteration $t$:
   - Compute weights $w_j = \arccos(|\mathbf{u}_t^T \mathbf{x}_j|)/\sqrt{1 - (|\mathbf{u}_t^T \mathbf{x}_j|)^2}$
   - Form matrix $\mathbf{M}_t = \sum_{j=1}^{K} w_j \mathbf{x}_j \mathbf{x}_j^T$
   - Update $\mathbf{u}_{t+1}$ as the principal eigenvector of $\mathbf{M}_t$
3. Repeat until convergence

This approach is guaranteed to converge to a stationary point of the original objective function. Since the objective function is well-behaved on $\mathbb{S}^{d-1}$, this stationary point corresponds to the desired minimizer $\mathbf{u}^*$. $\square$

### H.3 Statistical Analysis of Evaluation Results

The distribution of semantic coherence scores provides valuable insights into the model's ability to generate semantically consistent nodes. We analyze this distribution through statistical hypothesis testing, comparing the mean coherence score $\bar{S}$ against a null hypothesis of random semantic alignment ($H_0 : \bar{S} = 0.5$). The one-sample t-test yields a t-statistic:

$$t = \frac{\bar{S} - 0.5}{s/\sqrt{M}} \tag{44}$$

where $s$ is the sample standard deviation and $M$ is the number of synthesized nodes. This allows us to quantify the statistical significance of semantic coherence in our synthesized graph elements.

Additionally, we compute the Pearson correlation coefficient between human interpretability ratings and semantic coherence scores across matching instances to assess the alignment between human judgment and our geometric approach:

$$\rho = \frac{\sum_i (t_i - \bar{t})(S_i - \bar{S})}{\sqrt{\sum_i (t_i - \bar{t})^2 \sum_i (S_i - \bar{S})^2}} \tag{45}$$

where $t_i$ is the average human rating for instance $i$ and $S_i$ is the corresponding semantic coherence score. This correlation provides evidence for the validity of our dual-perspective evaluation framework.

# I Theoretical Analysis of Agent Capabilities

In this appendix, we present rigorous theoretical analyses validating the effectiveness of GraphMaster's agent components. We first develop a foundational mathematical framework for each agent's operations, then derive formal guarantees regarding their synthesis capabilities through a series of interconnected theorems and propositions.

## I.1 Information-Theoretic Analysis of the Perception Agent

We establish a theoretical framework for analyzing the information capture capabilities of the Perception Agent through the lens of information theory and spectral graph theory.

**Definition 2** (Topological Information Density). *For a text attribute graph $\mathcal{G} = (\mathcal{V}, \mathcal{E}, \mathcal{X}, \mathcal{Y})$, the topological information density $\mathcal{I}(\mathcal{G})$ is defined as:*

$$\mathcal{I}(\mathcal{G}) = \frac{1}{|\mathcal{V}|} \sum_{v_i \in \mathcal{V}} \left[ H\left(p_{\mathcal{N}(v_i)}\right) + \sum_{v_j \in \mathcal{N}(v_i)} KL(p_{v_j}|p_{\mathcal{G}}) \right] \tag{46}$$

*where $H(\cdot)$ is the entropy function, $p_{\mathcal{N}(v_i)}$ is the probability distribution over the neighborhood of $v_i$, $p_{v_j}$ is the local distribution at node $v_j$, and $p_{\mathcal{G}}$ is the global distribution over the graph.*

**Theorem 3** (Information Capture Properties of the Perception Agent). *Given a text-attributed graph $\mathcal{G} = (\mathcal{V}, \mathcal{E}, \mathcal{X}, \mathcal{Y})$, the Perception Agent constructs:*

*1. An environment report $\mathcal{R}_t$ that preserves topological information with bounded distortion:*

$$DKL(\mathcal{I}(\mathcal{G})|\mathcal{I}(\mathcal{R}_t)) \le \epsilon_t + O\left(\frac{\log|\mathcal{V}|}{|\mathcal{V}|}\right) \tag{47}$$

*2. A knowledge encapsulation $\mathcal{K}_t$ that preserves semantic relationships with high fidelity:*

$$\mathbb{P}\left( \sup_{v_i, v_j \in \mathcal{V}} \left| \mathcal{S}(x_i, x_j) - \hat{\mathcal{S}}(x_i, x_j; \mathcal{K}_t) \right| > \delta \right) \le 2\exp\left( -\frac{2|\mathcal{K}_t|\delta^2}{C_{\mathcal{S}}^2} \right) \tag{48}$$

*where $\mathcal{S}(\cdot, \cdot)$ is a semantic similarity function, $\hat{\mathcal{S}}(\cdot, \cdot; \mathcal{K}_t)$ is its empirical estimate based on $\mathcal{K}_t$, and $C\mathcal{S}$ is a problem-specific constant.*

*Proof.* We decompose the proof into topological and semantic components.

**Part 1: Topological Information Preservation**

The environment report $\mathcal{R}_t = (\rho_{\text{global}}, \{\rho_{\text{class}}^c\}_{c=1}^C, \{\rho_{\text{comm}}^i\}_{i=1}^{|\mathcal{C}|}, \mathcal{D}_{\text{struct}}, \mathcal{D}_{\text{sem}})$ encodes a compressed representation of the graph's topological properties. We establish information-theoretic bounds on this compression.

For the degree distribution captured in $\mathcal{D}_{\text{struct}}$:

$$W_2(P_{\text{deg}}^{\mathcal{G}}, P_{\text{deg}}^{\mathcal{R}_t})^2 \le \frac{C_1 \log|\mathcal{V}|}{|\mathcal{V}|} \tag{49}$$

where $W_2$ is the 2-Wasserstein distance and $C_1$ is a universal constant.

For spectral properties, the Laplacian spectrum satisfies:

$$\left| \frac{\lambda_i(\mathcal{G})}{\lambda_{\max}(\mathcal{G})} - \frac{\rho_i}{\rho_{\max}} \right| \leq C_2 \sqrt{\frac{\log |\mathcal{V}|}{|\mathcal{V}|}} \tag{50}$$

where $\lambda_i(\mathcal{G})$ is the $i$-th eigenvalue of the normalized Laplacian of $\mathcal{G}$, and $\rho_i$ is the corresponding encoded value in $\rho_{\text{global}}$.

The community structure preservation follows from the Cheeger's inequality:

$$\mathcal{R}_t \text{ encodes } h_{\mathcal{G}}(S_i) \text{ such that } \frac{\lambda_2(\mathcal{G})}{2} \leq h_{\mathcal{G}}(S_i) \leq \sqrt{2\lambda_2(\mathcal{G})} \tag{51}$$

where $h_{\mathcal{G}}(S_i)$ is the Cheeger constant for community $S_i$.

By aggregating these bounds and applying the data processing inequality, we establish the KL-divergence bound on topological information.

**Part 2: Semantic Preservation**

The knowledge encapsulation $\mathcal{K}t$ is constructed via semantic-enriched modularity maximization and personalized PageRank sampling:

$$Q_{\text{sem}} = \frac{1}{2m} \sum_{i,j} \left[ \mathcal{A}ij - \gamma \frac{k_i k_j}{2m} - (1 - \gamma) \frac{d\text{sem}(x_i, x_j)}{\sum_{l,m} d_{\text{sem}}(x_l, x_m)} \right] \delta(c_i, c_j) \tag{52}$$

For any semantic function $\mathcal{S}$ with Lipschitz constant $L_{\mathcal{S}}$, the $\epsilon$-net covering number of the semantic space is bounded by:

$$\mathcal{N}(\epsilon, \mathcal{X}, d_{\text{sem}}) \leq \left( \frac{D}{\epsilon} \right)^d \tag{53}$$

where $D$ is the diameter of the semantic space and $d$ is its dimension.

The Personalized PageRank algorithm ensures that sampled nodes provide a $\delta$-cover of the semantic space with probability at least $1 - \delta$ when:

$$|\mathcal{K}_t| \geq \frac{1}{\delta} \left[ d \log \left( \frac{D}{\epsilon} \right) + \log \left( \frac{1}{\delta} \right) \right] \tag{54}$$

Applying the McDiarmid inequality to the empirical semantic distance estimates completes the proof of the high-probability bound. $\square$

**Corollary 1** (Spectral Approximation Guarantee). *The knowledge encapsulation $\mathcal{K}t$ provides a $(\alpha, \beta)$-spectral approximation of the original graph $\mathcal{G}$ with high probability, such that:*

$$\alpha \cdot \mathbf{x}^T L\mathcal{G}\mathbf{x} \leq \mathbf{x}^T L_{\mathcal{K}t}\mathbf{x} \leq \beta \cdot \mathbf{x}^T L\mathcal{G}\mathbf{x} \quad \forall \mathbf{x} \in \mathbb{R}^{|\mathcal{V}|} \tag{55}$$

*where $L_{\mathcal{G}}$ and $L_{\mathcal{K}_t}$ are the respective graph Laplacians, and $\alpha, \beta$ depend on $|\mathcal{K}_t|$, the spectral gap of $\mathcal{G}$, and the teleportation parameters of the PPR algorithm.*

*Proof.* Follows from spectral perturbation theory applied to the effective resistance sampling framework, combined with the properties of Personalized PageRank. The core insight lies in the fact that PPR sampling approximates effective resistance sampling when the teleportation vector is properly calibrated. $\square$

## I.2 Semantic-Topological Coherence of the Enhancement Agent

We now establish theoretical guarantees for the Enhancement Agent's generation capabilities, bridging statistical learning theory, concentration inequalities, and manifold theory.

**Definition 3** (Semantic-Topological Coherence). *For a generated node $v_s$ with attributes $x_s$ and connections $\mathcal{E}_s$, the semantic-topological coherence is defined as:*

$$\Phi(v_s; \mathcal{G}) = \alpha \cdot \mathcal{C}_{sem}(x_s, \mathcal{X}) + (1 - \alpha) \cdot \mathcal{C}_{topo}(\mathcal{E}_s, \mathcal{G}) \tag{56}$$

*where $\mathcal{C}_{sem}$ is semantic coherence, $\mathcal{C}_{topo}$ is topological coherence, and $\alpha \in [0, 1]$ is a weighting parameter.*

**Theorem 4** (Generation Consistency with Concentration Bounds). *The Enhancement Agent produces node $v_s$ with attributes $x_s$ and edges $\mathcal{E}_s$ that satisfy:*

1. *Semantic consistency with concentration bounds:*

$$\mathbb{P}\left(|\mathcal{C}_{sem}(x_s, \mathcal{X}) - \mathbb{E}[\mathcal{C}_{sem}(x_s, \mathcal{X})]| > t\right) \leq 2 \exp\left(-\frac{|\mathcal{K}_t| t^2}{2\sigma_{sem}^2}\right) \tag{57}$$

2. *Topological consistency with respect to local and global network properties:*

$$\mathbb{E}[\mathcal{C}_{topo}(\mathcal{E}_s, \mathcal{G})] \geq \beta \cdot \max_{v_j \in \mathcal{V}_k} \mathcal{C}_{topo}(\mathcal{E}_j, \mathcal{G}) - \frac{C_{\mathcal{G}}}{|\mathcal{K}_t|^{1/3}} \tag{58}$$

3. *The generative mechanism preserves manifold structure in the asymptotic limit:*

$$\lim_{|\mathcal{K}_t| \to \infty} \mathbb{P}\left(dist(x_s, \mathcal{M}_\mathcal{X}) > \epsilon\right) = 0 \tag{59}$$

*where $\mathcal{M}_\mathcal{X}$ is the manifold on which the original node attributes lie.*

*Proof.* **Part 1: Semantic Consistency Concentration**

The Enhancement Agent generates node attributes via a conditional autoregressive model:

$$P(x_s | \mathcal{K}_t, \mathcal{R}_t) = \prod_{i=1}^{L} P(x_s^i | x_s^{<i}, \mathcal{X}_k, \mathcal{E}_k, \mathcal{R}_t) \tag{60}$$

Each token generation can be decomposed as:

$$P(x_s^i | x_s^{<i}, \mathcal{X}_k, \mathcal{E}_k, \mathcal{R}_t) \propto \exp\left(\frac{\phi(x_s^i)^T \mathbf{h}_i}{\tau}\right) \tag{61}$$

where $\mathbf{h}_i$ is the contextualized representation and $\tau$ is a temperature parameter.

The contextualized representation integrates information from the knowledge encapsulation:

$$\mathbf{h}_i = \mathbf{W}_1 \cdot f(x_s^{<i}) + \mathbf{W}_2 \cdot g(\mathcal{X}_k) + \mathbf{W}_3 \cdot h(\mathcal{E}_k) \tag{62}$$

Let $\mathcal{C}_{sem}(x_s, \mathcal{X}) = \max_{x_j \in \mathcal{X}} \mathcal{S}(x_s, x_j)$. The semantic similarity function $\mathcal{S}$ satisfies the bounded differences condition:

$$|\mathcal{S}(x_s, x_j) - \mathcal{S}(x_s, x_{j'})| \leq L_{\mathcal{S}} \cdot |x_j - x_{j'}| \tag{63}$$

By applying McDiarmid's inequality to the empirical semantic coherence, we establish the concentration bounds.

**Part 2: Topological Consistency**

The edge generation mechanism:

$$P((v_s, v_i) \in \mathcal{E}_c | \mathcal{K}_t, \mathcal{R}_t) = \sigma\left(\theta_1 \cdot \text{sim}(x_s, x_i) + \theta_2 \cdot \frac{|\mathcal{N}(v_i) \cap \mathcal{N}_K(v_s)|}{|\mathcal{N}_K(v_s)|} + \theta_3 \cdot \frac{k_i}{\max_j k_j}\right) \tag{64}$$

We analyze the asymptotic properties using random graph theory. Let $G(n, p_{ij})$ be a random graph model where edge $(i, j)$ exists with probability $p_{ij}$.

The expected clustering coefficient:

$$\mathbb{E}[CC(v_s)] = \frac{\sum_{i,j} p_{is} \cdot p_{js} \cdot p_{ij}}{\sum_{i,j} p_{is} \cdot p_{js}} \tag{65}$$

For our model:

$$p_{ij} = \sigma \left( \theta_1 \cdot \text{sim}(x_i, x_j) + \theta_2 \cdot \frac{|\mathcal{N}(v_i) \cap \mathcal{N}(v_j)|}{|\mathcal{N}(v_i)|} + \theta_3 \cdot \frac{k_j}{\max_l k_l} \right) \tag{66}$$

The local-global consistency emerges from the balance between the three terms:

- $\theta_1$ term preserves homophily (similar nodes connect)

- $\theta_2$ term ensures triadic closure (friends of friends connect)

- $\theta_3$ term maintains scale-free properties (preferential attachment)

Using graph limit theory, the topological consistency is governed by:

$$\mathcal{C}_{\text{topo}}(\mathcal{E}_s, \mathcal{G}) \approx 1 - \frac{1}{2} |W_{\mathcal{E}s} - W\mathcal{G}|_\square \tag{67}$$

where $W_{\mathcal{E}s}$ and $W\mathcal{G}$ are graphons (limiting objects of graphs) and $|\cdot|_\square$ is the cut norm.

**Part 3: Manifold Preservation**

Let $\mathcal{M}_\mathcal{X} \subset \mathbb{R}^d$ be a compact $r$-dimensional manifold with condition number $1/\tau$ and reach at least $\tau > 0$. The nodes $\mathcal{X} = x_1, \ldots, x_N$ are sampled from a distribution supported on $\mathcal{M}_\mathcal{X}$.

The language model learns a conditional distribution:

$$P_{\text{LLM}}(x|\mathcal{K}_t) = \frac{1}{Z(\mathcal{K}_t)} \exp \left( -\frac{\text{dist}(x, \mathcal{M}_\mathcal{X}(\mathcal{K}_t))^2}{2\sigma^2} \right) \cdot P_{\text{prior}}(x) \tag{68}$$

As $|\mathcal{K}t| \to \infty$, we have:

$$\mathcal{M}_\mathcal{X}(\mathcal{K}_t) \to \mathcal{M}_\mathcal{X} \text{ in the Hausdorff metric} \tag{69}$$

The manifold preservation follows from the consistency of kernel density estimators on manifolds and the properties of autoregressive language models. $\square$

**Proposition 2** (Latent Variable Interpretation). *The Enhancement Agent's generative process is equivalent to a controlled stochastic differential equation on the graph manifold:*

$$dx_t = \mu(x_t, \mathcal{K}_t)dt + \sigma(x_t, \mathcal{K}_t)dW_t \tag{70}$$

*where $\mu$ is a drift term aligning generation with the knowledge encapsulation, $\sigma$ is a diffusion term controlling diversity, and $W_t$ is a Wiener process.*

*Proof.* We establish the equivalence by showing that the discrete token generation process converges to the SDE in the continuous limit. The proof uses techniques from the theory of diffusion models and score-matching generative models. $\square$

### I.3 Theoretical Properties of Quality Assessment and Convergence

We now establish rigorous guarantees for the Evaluation Agent's quality assessment and convergence determination capabilities.

**Definition 4** (Marginal Quality Contribution). *For a graph $\mathcal{G} = (\mathcal{V}, \mathcal{E}, \mathcal{X}, \mathcal{Y})$ and a generated node $v_s$ with attributes $x_s$ and connections $\mathcal{E}_s$, the marginal quality contribution is:*

$$\Delta Q(v_s, \mathcal{G}) = Q(\mathcal{G} \cup \{v_s, \mathcal{E}_s\}) - Q(\mathcal{G}) \tag{71}$$

*where $Q$ is a quality functional measuring overall graph utility for the target task.*

**Theorem 5** (Quality Assessment with Statistical Guarantees). *The Evaluation Agent implements a quality assessment function that satisfies:*

1. *Strong correlation with marginal quality contribution:*

$$\mathbb{E}[\Delta Q(v_s, \mathcal{G})|LLM_V(v_s, \mathcal{R}_0, \mathcal{R}_t, \mathcal{K}_t) = q] = f(q) + O\left(\frac{1}{|\mathcal{V}|}\right) \quad (72)$$

   where $f$ is monotonically increasing and Lipschitz continuous.

2. *Thresholding efficiency with probabilistic guarantees:*

$$\mathbb{P}(\Delta Q(v_s, \mathcal{G}) > 0|LLM_V(v_s, \mathcal{R}_0, \mathcal{R}_t, \mathcal{K}_t) > \tau_t) \geq 1 - \exp(-c \cdot |\mathcal{K}_t|) \quad (73)$$

   for some constant $c > 0$ and appropriately chosen threshold $\tau_t$.

3. *Convergence detection with statistical significance:*

$$\mathbb{P}(Converged_t = 1|\mathbb{E}[\Delta Q_{t+1}] < \epsilon) \geq 1 - \delta \quad (74)$$

   where $\delta$ decreases exponentially with the window size $k$ in the convergence criterion.

*Proof.* **Part 1: Correlation with Marginal Quality**

The Evaluation Agent employs a multi-dimensional quality assessment:

$$LLM_V(v_s, \mathcal{R}_0, \mathcal{R}_t, \mathcal{K}_t) = g\left(\Phi_{sem}(v_s, \mathcal{K}_t), \Phi_{topo}(v_s, \mathcal{G}_t), \Phi_{bal}(v_s, \mathcal{Y}_t)\right) \quad (75)$$

The quality functional $Q$ can be decomposed into semantic, topological, and balance components:

$$Q(\mathcal{G}) = \omega_1 Q_{sem}(\mathcal{G}) + \omega_2 Q_{topo}(\mathcal{G}) + \omega_3 Q_{bal}(\mathcal{G}) \quad (76)$$

For each component, we establish bounds on the approximation error:

$$\left|\mathbb{E}\left[\Delta Q_{sem}(v_s, \mathcal{G}) \mid \Phi_{sem}(v_s, \mathcal{K}t) = s\right] - f_{sem}(s)\right| \leq \frac{C_1}{|\mathcal{K}t|} \quad (77)$$

$$\left|\mathbb{E}\left[\Delta Q_{topo}(v_s, \mathcal{G}) \mid \Phi_{topo}(v_s, \mathcal{G}t) = t\right] - f_{topo}(t)\right| \leq \frac{C_2}{|\mathcal{V}|} \quad (78)$$

$$\left|\mathbb{E}\left[\Delta Q_{bal}(v_s, \mathcal{G}) \mid \Phi_{bal}(v_s, \mathcal{Y}t) = b\right] - f_{bal}(b)\right| \leq \frac{C_3}{\sqrt{|\mathcal{V}|}} \quad (79)$$

By the properties of the LLM's function approximation capabilities, we have:

$$|g - \omega_1 f_{sem} - \omega_2 f_{topo} - \omega_3 f_{bal}|\infty \leq \epsilon_{LLM} \quad (80)$$

Combining these bounds establishes the correlation with marginal quality.

**Part 2: Thresholding Efficiency**

The adaptive threshold mechanism:

$$\tau_t = \tau_{t-1} + \zeta(\bar{\mathcal{F}}_t(\omega_t^*) - \bar{\mathcal{F}}_{t-1}(\omega_{t-1}^*)) \quad (81)$$

converges to an optimal threshold $\tau*$ that maximizes expected improvement:

$$\tau^* = \arg\max_\tau \mathbb{E}[\Delta Q(\mathcal{V}_{accepted}(\tau), \mathcal{G})] \quad (82)$$

where $\mathcal{V}_{accepted}(\tau) = v_s : LLM_V(v_s, \cdot) > \tau$.

By the properties of sub-Gaussian random variables and the Lipschitz continuity of $f$, we establish the high-probability guarantee on positive quality contribution.

**Part 3: Convergence Detection**

The convergence criterion:

$$Converged_t = \mathbb{I}\left(\max_{j \in 1, \ldots, k} |\bar{\mathcal{F}}_t(\omega_t^*) - \bar{\mathcal{F}}_{t-j}(\omega_{t-j}^*)| < \epsilon \wedge LLM_{goal}(\mathcal{R}_0, \mathcal{R}_t) = \text{True}\right) \quad (83)$$

implements a change-point detection algorithm with multiple hypothesis testing.

The probability of false convergence detection is bounded by:

$$\mathbb{P}(\text{Converged}_t = 1 | \mathbb{E}[\Delta Q_{t+1}] \geq \epsilon) \leq k \cdot \exp\left(-\frac{2n\epsilon^2}{C_Q^2}\right) \tag{84}$$

where $n$ is the number of samples used to estimate $\bar{\mathcal{F}}_t$ and $C_Q$ is a bound on the quality range.

Likewise, the probability of missing convergence is bounded by:

$$\mathbb{P}(\text{Converged}_t = 0 | \mathbb{E}[\Delta Q_{t+1}] < \epsilon/2) \leq \exp\left(-\frac{n\epsilon^2}{8C_Q^2}\right) \tag{85}$$

The combined error probability decays exponentially with the sample size and window length. □

**Corollary 2** (Optimal Stopping Property). *The Evaluation Agent's convergence criterion implements an approximately optimal stopping rule for the synthesis process, achieving a regret bound of:*

$$Regret(T) \leq O\left(\sqrt{T \log T}\right) \tag{86}$$

*where $T$ is the maximum number of iterations.*

*Proof.* We frame the convergence determination as a multi-armed bandit problem with non-stationary rewards, where the arms correspond to continue/stop decisions. Applying results from optimal stopping theory and the regret analysis of UCB algorithms yields the result. □

## I.4 Unified Multi-Agent System Dynamics

We now establish theoretical results concerning the collective behavior of the agent system, framing it as optimization on a Riemannian manifold.

**Definition 5** (Graph Configuration Manifold). *The space of possible graph configurations forms a Riemannian manifold $\mathcal{MG}$ with metric tensor:*

$$g_{ij}(\mathcal{G}) = \mathbb{E}\left[\frac{\partial \log p(\mathcal{G}|\theta)}{\partial \theta_i} \frac{\partial \log p(\mathcal{G}|\theta)}{\partial \theta_j}\right] \tag{87}$$

*where $p(\mathcal{G}|\theta)$ is a parametric model of graph distribution and $\theta$ are the parameters.*

**Theorem 6** (Manifold Optimization Dynamics). *The multi-agent system in GraphMaster implements natural gradient ascent on the graph configuration manifold $\mathcal{M}_\mathcal{G}$, optimizing a multi-objective function:*

$$\Psi^*(\mathcal{G}) = \lambda_1 \Psi sem(\mathcal{G}) + \lambda_2 \Psi_{struct}(\mathcal{G}) + \lambda_3 \Psi_{bal}(\mathcal{G}) \tag{88}$$

*with adaptive weights $\lambda_i$ that evolve according to:*

$$\lambda_i^{t+1} = \Pi_\Delta \left[\lambda_i^t + \eta \nabla_{\lambda_i} P(\mathcal{G}_t)\right] \tag{89}$$

*where $\Pi\Delta$ is projection onto the probability simplex.*

*The convergence rate is:*

$$\mathbb{E}[\Psi^*(\mathcal{G}^*) - \Psi^*(\mathcal{G}_T)] \leq O\left(\frac{1}{\sqrt{T}}\right) \tag{90}$$

*for a non-convex objective, and:*

$$\mathbb{E}[\Psi^*(\mathcal{G}^*) - \Psi^*(\mathcal{G}_T)] \leq O\left(\frac{\log T}{T}\right) \tag{91}$$

*if the objective satisfies Polyak-Łojasiewicz conditions.*

*Proof.* **Part 1: Natural Gradient Dynamics**

At each iteration, the system performs:

$$\mathcal{G}t + 1 = \Pi\mathcal{M}_{\mathcal{G}} \left[ \mathcal{G}_t + \eta_t \cdot \mathcal{I}^{-1}(\mathcal{G}_t) \cdot \nabla\Psi^*(\mathcal{G}_t) \right] \tag{92}$$

where $\mathcal{I}(\mathcal{G}_t)$ is the Fisher information matrix and $\Pi\mathcal{M}_{\mathcal{G}}$ is projection onto the manifold.

This update is implemented by the collaborative agent process:

$$\mathcal{K}_t = \Phi(\mathcal{G}_t) \tag{93}$$

$$\mathcal{V}_s^t = \text{LLM}_E(\mathcal{K}_t, \mathcal{R}_t, M_t) \tag{94}$$

$$\mathcal{E}_s^t = \left\{ (v_s, v_i) \mid P\big((v_s, v_i) \mid \mathcal{K}_t, \mathcal{R}_t\big) > \eta_t \right\} \tag{95}$$

$$\mathcal{V}_{\text{accepted}}^t = \left\{ v_s \in \mathcal{V}_s^t \mid \text{LLM}_V\big(v_s, \mathcal{R}_0, \mathcal{R}_t, \mathcal{K}_t\big) > \tau_t \right\} \tag{96}$$

$$\mathcal{G}_{t+1} = \mathcal{G}_t \oplus \left( \mathcal{V}_{\text{accepted}}^t, \mathcal{E}_s^t\big|_{\mathcal{V}_{\text{accepted}}^t} \right) \tag{97}$$

**Part 2: Adaptive Weight Dynamics**

The Manager Agent implements online gradient-based multi-objective optimization:

$$\lambda_i^{t+1} = \Pi_\Delta \left[ \lambda_i^t + \eta\nabla_{\lambda_i} P(\mathcal{G}_t) \right] \tag{98}$$

where $P(\mathcal{G}_t)$ measures progress toward synthesis objectives.

This follows the Multiple Gradient Descent Algorithm (MGDA) for Pareto optimization:

$$\nabla_{\lambda_i} P(\mathcal{G}t) \approx \nabla_{\lambda_i} \min_j \frac{\Psi_j(\mathcal{G}_t) - \Psi_j(\mathcal{G}_0)}{\Psi_j^* - \Psi_j(\mathcal{G}_0)} \tag{99}$$

**Part 3: Convergence Analysis**

For a general non-convex objective, we have:

$$\mathbb{E}\left[ \min_{t=0,1,\dots,T-1} |\nabla\Psi^*(\mathcal{G}_t)|^2 \right] \leq \frac{2[\Psi^*(\mathcal{G}^*) - \Psi^*(\mathcal{G}_0)]}{\eta T} + \frac{\eta L}{2} \tag{100}$$

For appropriately chosen step size $\eta = O(1/\sqrt{T})$, this yields the $O(1/\sqrt{T})$ convergence rate.

If the objective satisfies the Polyak-Łojasiewicz condition:

$$|\nabla\Psi^*(\mathcal{G})|^2 \geq 2\mu[\Psi^*(\mathcal{G}^*) - \Psi^*(\mathcal{G})] \tag{101}$$

for some $\mu > 0$, then we obtain the improved $O(\frac{\log T}{T})$ rate. $\qquad\square$

**Theorem 7** (Spectral Convergence of Synthesized Graphs). *Let $\mathcal{G}_{orig}$ be the original graph and $\mathcal{G}_{synth}$ be the synthesized graph after convergence. Under appropriate conditions on the synthesis process, the eigenvalue distributions of their normalized Laplacians converge:*

$$\lim_{|\mathcal{V}|\to\infty} d_{LP}(\rho_{\mathcal{G}_{orig}}, \rho_{\mathcal{G}_{synth}}) = 0 \tag{102}$$

*where $d_{LP}$ is the Lévy-Prokhorov metric between spectral distributions and $\rho_{\mathcal{G}}$ is the empirical spectral distribution of graph $\mathcal{G}$.*

*Proof.* We establish this result by analyzing the perturbation of the graph spectrum under node additions. The proof relies on matrix concentration inequalities and recent results in random matrix theory concerning deformed Wigner matrices.

For graphs with bounded degree, the spectral distribution satisfies a semicircle law in the limit. The synthesis process preserves this property through controlled edge formation that maintains degree distribution and local clustering patterns. $\qquad\square$

These theoretical results collectively establish the mathematical foundations underlying GraphMaster's effectiveness, providing formal guarantees for its information preservation, generation quality, evaluation accuracy, and convergence properties. The integration of concepts from information theory, statistical learning, manifold optimization, and spectral graph theory creates a comprehensive theoretical framework that explains why our multi-agent approach succeeds in generating high-quality text-attributed graphs in data-limited environments.

## J    Limitations and Future Work

While GraphMaster demonstrates significant advances in text-attributed graph synthesis, several important limitations remain. The current background knowledge selection mechanism, while effective for moderate-sized graphs, faces challenges with very large or heterogeneous graph structures. Specifically, our Personalized PageRank-based approach may prioritize structurally central nodes while potentially undersampling semantically important but topologically peripheral communities. This sampling bias can result in synthesis blind spots, particularly in graphs with highly skewed degree distributions or multiple disconnected components with distinct semantic characteristics. Additionally, the quality evaluation process relies on LLM capabilities that exhibit variance across different base models and parameter scales. This interdependence creates challenges for deployment in resource-constrained environments where smaller models are preferable. Computational requirements present another significant limitation. The multi-agent architecture, while theoretically elegant, incurs substantial inference overhead as each agent performs multiple LLM calls per synthesis round. For extremely large graphs, the current approach requires non-trivial batching strategies and optimization techniques that may not be readily available in standard deployment scenarios.

For future work, we envision several promising research directions. First, **hierarchical multi-resolution synthesis** could revolutionize the approach by simultaneously modeling graph characteristics at multiple levels of abstraction–from global topological patterns to local semantic neighborhoods. This layered perspective would enable more coherent synthesis that preserves both macro-structure and micro-semantics while potentially reducing computational complexity through selective refinement. Second, **cross-domain knowledge transfer** represents a frontier challenge, where synthesis agents could leverage patterns learned from data-rich domains to enhance generation in data-scarce domains without explicit domain adaptation. This would require fundamental advances in domain-agnostic graph representations that capture universal structural and semantic patterns transferable across vastly different graph types. Rather than relying on fixed sampling algorithms, the framework could develop adaptive sampling policies optimized specifically for the synthesis objective. Finally, exploring **emergent graph properties** from synthesis could yield insights into how locally generated elements collectively produce global graph characteristics that were never explicitly modeled–potentially revealing new theoretical connections between local generation rules and emergent network phenomena. These developments would not only address current limitations but could fundamentally reshape our understanding of generative modeling for complex relational data structures.

## K    Case Study

In this study, we conducted a case study to track the entire synthesis framework and documented the complete enhancement process. Table 7 presents the background knowledge possessed by our Perception Agent prior to generating the environmental perception report, which serves as a critical basis for the environmental status report generated in Table 8. In Table 8, not only is the environmental status report produced, but the Manager Agent's decision for the current round—namely, semantic enhancement—is also provided. Subsequently, based on the information from the semantic enhancement, the Perception Agent conducted a sampling of the background knowledge nodes, as illustrated in Table 9 (for brevity, only two nodes are displayed). Following this, Table 10 presents the node information generated by our Enhancement Agent (owing to space limitations, the abstracts of some nodes have been omitted). Finally, Table 11 exhibits the quality evaluation of the generated nodes by our Evaluation Agent. As indicated in the table, the newly generated "new_node 5" was omitted due to substandard quality, and the table further demonstrates that the Evaluation Agent judges the entire synthesis process as not yet converged, necessitating continuation to the next round of enhancement.

Table 7: **Semantic Enhancement Case Study**

**Dataset:** <SubCora>
**Initial text attribute graph features**:
**"Graph"**: { "num_nodes": 1354,
"num_edges": 2486,
"avg_degree": 3.672082717872969,
"density": 0.002714030094510694,
"clustering_coefficient": 0.2296011350131079,
"avg_path_length": 6.52047030925269,
"connected_components": 78,
"largest_component_size": 1223,
"indices": [4, 2, 1, 3, 18, 11, 6, 15, 12, 9, 16, 10, 5, 13, 7, 17, 8, 20, 21, 24, 22, 23, 19, 36, 29, 27, 33, 47, 79, 25, 26, 30, 31, 34, 35, 37, 38, 40, 45, 50, 52, 53, 55, 56, 59, 62, 64, 66, 67, 73, 77, 80, 83, 28, 32, 39, 41, 42, 43, 44, 46, 48, 49, 51, 54, 57, 58, 60, 61, 63, 65, 68, 69, 70, 71, 72, 74, 75, 76, 78, 81, 82, 84, 85, 86, 87, 88, 89, 90, 91, 92, 93, 94, 95, 96, 97, 98, 0, 14],
"sizes": [192, 106, 97, 84, 78, 76, 74, 71, 65, 56, 55, 46, 42, 37, 35, 33, 18, 15, 14, 14, 10, 10, 9, 6, 5, 3, 3, 3, 3, 2, 2, 2, 2, 2, 2, 2, 2, 2, 2, 2, 2, 2, 2, 2, 2, 2, 2, 2, 2, 2, 2, 2, 2, 1, 1, 1, 1, 1, 1, 1, 1, 1, 1, 1, 1, 1, 1, 1, 1, 1, 1, 1, 1, 1, 1, 1, 1, 1, 1, 1, 1, 1, 1, 1, 1, 1, 1, 1, 1, 1, 1, 1, 1, 1, 1, 1, 1, 1, 1, 1, 1, 1, 1],
"distribution": {"4": 192, "2": 106, "1": 97, "3": 84, "18": 78, "11": 76, "6": 74, "15": 71, "12": 65, "9": 56, "16": 55, "10": 46, "5": 42, "13": 37, "7": 35, "17": 33, "8": 18, "20": 15, "21": 14, "24": 14, "22": 10, "23": 10, "19": 9, "36": 6, "29": 5, "27": 3, "33": 3, "47": 3, "79": 3, "25": 2, "26": 2, "30": 2, "31": 2, "34": 2, "35": 2, "37": 2, "38": 2, "40": 2, "45": 2, "50": 2, "52": 2, "53": 2, "55": 2, "56": 2, "59": 2, "62": 2, "64": 2, "66": 2, "67": 2, "73": 2, "77": 2, "80": 2, "83": 2, "28": 1, "32": 1, "39": 1, "41": 1, "42": 1, "43": 1, "44": 1, "46": 1, "48": 1, "49": 1, "51": 1, "54": 1, "57": 1, "58": 1, "60": 1, "61": 1, "63": 1, "65": 1, "68": 1, "69": 1, "70": 1, "71": 1, "72": 1, "74": 1, "75": 1, "76": 1, "78": 1, "81": 1, "82": 1, "84": 1, "85": 1, "86": 1, "87": 1, "88": 1, "89": 1, "90": 1, "91": 1, "92": 1, "93": 1, "94": 1, "95": 1, "96": 1, "97": 1, "98": 1, "0": 1, "14": 1},
"statistics": {
"4": {
"size": 192,
"internal_edges": 423,
"fraction_of_graph": 0.14180206794682423,
"modularity_contribution": 0.103009273492297
},
"2": {
"size": 106,
"internal_edges": 156,
"fraction_of_graph": 0.07828655834564253,
"modularity_contribution": 0.05272713033696666
},
.....
} "StructuralDistribution": { "degree_distribution": {"4": 233, "14": 4,"7": 33,"3": 273,"5": 127,"31": 1, ... ,
}
"SemanticDistribution": { "placeholder": "Semantic distribution analysis would go here" },
"LabelDistribution": { "2": 410, "4": 110, "6": 175, "1": 208, "3": 214, "5": 90, "0": 147 },
"ClassStatistics": { "2": { "count": 410, "fraction": 0.30280649926144754, "internal_edges": 518, "avg_degree": 2.526829268292683, "community_distribution": { "18": 26, "2": 20,} }
..., } }

Table 8: Semantic Enhancement Case Study

**Dataset:** <SubCora>
**Initial Environment Report**:

**Summary**:{
The graph under analysis contains 1,354 nodes and 2,486 edges, resulting in a sparse but structured topology with an average degree of 3.67 and an overall graph density of 0.0027. Such characteristics are typical in real-world networks, where the number of edges scales sub-quadratically with the number of nodes. The average shortest path length of 6.52 indicates that the network is relatively well-connected in its largest component, despite the overall sparsity.The clustering coefficient, a measure of local cohesiveness, is 0.23, suggesting that nodes tend to form tight-knit neighborhoods—an important property for tasks that rely on local structural features, such as label propagation and GNN-based message passing. Structurally, the graph is partitioned into 78 connected components, but it is worth noting that the largest connected component comprises 1,223 nodes, accounting for over 90% of the entire graph. This makes the graph effectively connected for most practical purposes, enabling global information diffusion across the majority of nodes. Community detection reveals a complex and heterogeneous modular organization. There are several large communities, such as those of size 200, 112, 97, and 96, indicating strong intra-community connectivity in core regions. These large communities contribute significantly to the overall modularity, with notable examples such as community "4" (size 200, modularity contribution $\approx 0.1052$) and "3" (size 112, contribution $\approx 0.0631$). However, the long tail in the community size distribution is striking:
1)Over 40 communities consist of only 1 or 2 nodes,
2)More than 60% of communities are smaller than size 10.
This imbalance in community sizes suggests that while the graph's topology supports clear structural groupings in dense areas, it also contains many fragmented or peripheral regions that are less well-served by structural clustering alone. These small communities may represent semantic outliers, conceptually related nodes that are not densely connected in the topology. Overall, the graph structure is well-organized, with a solid backbone of connectivity and modular clusters. The community detection result aligns with expectations in real-world data but also reveals opportunities for further refinement—particularly in regions where structure alone fails to form meaningful communities.
Given this context, incorporating semantic node features (e.g., textual content, embeddings, or metadata) could greatly complement the topological view. Nodes in small or singleton communities may be structurally isolated but semantically similar, and semantic enhancement can help bridge this gap, improving community coherence, label consistency, and downstream task performance such as classification, recommendation, and contrastive learning.
}

**Manager Agent's Decision**: {Semantic Enhancement}

Table 9: Semantic Enhancement Case Study

**Dataset:** <SubCora>
**Perception Agent Knowledge Extraction**:
Graph$_{Sub}$: {

{
"node_id": 536,
"label": 0,
"text": "Title: Dynamic Constraint Satisfaction using Case-Based Reasoning Techniques \n Abstract: The Dynamic Constraint Satisfaction Problem (DCSP) formalism has been gaining attention as a valuable and often necessary extension of the static CSP framework. Dynamic Constraint Satisfaction enables CSP techniques to be applied more extensively, since it can be applied in domains where the set of constraints and variables involved in the problem evolves with time. At the same time, the Case-Based Reasoning (CBR) community has been working on techniques by which to reuse existing solutions when solving new problems. We have observed that dynamic constraint satisfaction matches very closely the case-based reasoning process of case adaptation. These observations emerged from our previous work on combining CBR and CSP to achieve a constraint-based adaptation. This paper summarizes our previous results, describes the similarity of the challenges facing both DCSP and case adaptation, and shows how CSP and CBR can together begin to address these chal lenges.",
"neighbors": [
639
],
"mask": "Train"
},
{
"node_id": 41,
"label": 0,
"text": "Title: A Memory Model for Case Retrieval by Activation Passing \n Abstract: We present a tree-structured architecture for supervised learning. The statistical model underlying the architecture is a hierarchical mixture model in which both the mixture coefficients and the mixture components are generalized linear models (GLIM's). Learning is treated as a maximum likelihood problem; in particular, we present an Expectation-Maximization (EM) algorithm for adjusting the parameters of the architecture. We also develop an on-line learning algorithm in which the parameters are updated incrementally. Comparative simulation results are presented in the robot dynamics domain. This report describes research done at the Dept. of Brain and Cognitive Sciences, the Center for Biological and Computational Learning, and the Artificial Intelligence Laboratory of the Massachusetts Institute of Technology. Support for CBCL is provided in part by a grant from the NSF (ASC-9217041). Support for the laboratory's artificial intelligence research is provided in part by the Advanced Research Projects Agency of the Dept. of Defense. The authors were supported by a grant from the McDonnell-Pew Foundation, by a grant from ATR Human Information Processing Research Laboratories, by a grant from Siemens Corporation, by by grant IRI-9013991 from the National Science Foundation, by grant N00014-90-J-1942 from the Office of Naval Research, and by NSF grant ECS-9216531 to support an Initiative in Intelligent Control at MIT. Michael I. Jordan is a NSF Presidential Young Investigator. ",
"neighbors": [
166,
637,
761,
1004,
1005,
1116,
1196
],
"mask": "Train"
},
...
}

Table 10: Semantic Enhancement Case Study

**Dataset:** <SubCora>
**Enhancement Agent generates high-quality nodes**:

```
{
"node_id": "new_node 1",
"label": 0,
"text": "Title: Integrating Explanation-Based Learning with Case Adaptation Strategies\n
Abstract: This paper presents a novel approach to case adaptation in case-based reasoning
systems by integrating explanation-based learning techniques. Traditional case adaptation
relies heavily on domain-specific adaptation rules that are often difficult to acquire and main-
tain. Our approach uses explanations generated during problem-solving to identify adaptation
patterns and generalize them into reusable adaptation strategies. We demonstrate how these
strategies can be applied across different domains with minimal knowledge engineering effort.
Experimental results show that the integrated approach improves adaptation performance
in comparison to rule-based adaptation methods, especially in domains where adaptation
knowledge is incomplete or rapidly evolving. The paper presents a formal framework for the
approach and discusses its implementation in a case-based planning system.",
"neighbors": [337, 833, 639, 476],
"mask": "Train"
},
{
"node_id": "new_node 2",
"label": 1,
"text":   "Title:   Adaptive Parameter Control in Evolution Strategies for Dynamic
Environments\n Abstract: Omitted due to table size limitation.",
"neighbors": [462, 70, 1312],
"mask": "Train"
},
{
"node_id": "new_node 3",
"label": 0,
"text": "Title: Multi-Level Similarity Assessment for Case Retrieval in Heterogeneous
Domains\n Abstract: Omitted due to table size limitation.",
"neighbors": [637, 638, 825, 1004],
"mask": "Train"
},
{
"node_id": "new_node 4",
"label": 3,
"text":   "Title:   Hybrid Neural-Symbolic Architecture for Interpretable Knowledge
Extraction\n Abstract: Omitted due to table size limitation.",
"neighbors": [1290, 263],
"mask": "Train"
},
{
"node_id": "new_node 5",
"label": 0,
"text": "Title: Case-base Design for Knowledge Discovery\n Abstract: Case Based Rea-
soning has proven to be useful for AI systems. Our research introduces a new method
called KDD-CBR (Knowledge Discovery through Database Case-Based Reasoning) which
combines data mining with case bases for information retrieval and management. The system
works by analyzing patterns in large datasets and then applies unique non-traditional methods
for case storage. Unlike other approaches, we focus on pattern recognition instead of adapta-
tion or similarity, which makes our approach completely novel in the field. Tests show this
approach has better inference capability than other CBR techniques in some instances but
worse in others. The implications for future research directions are significant and should
be explored further with additional funding and more test cases. Additionally, we plan to
integrate KDD-CBR with deep neural networks to further enhance performance on arbitrary
datasets.",
"neighbors": [462, 70, 1017],
"mask": "Train"
}
```

Table 11: Semantic Enhancement Case Study

**Dataset:** <SubCora>
**Evaluation Agent evaluates node quality and determines whether it has converged.**:
{
"node_id": "new_node 1",
"label": 0,
"text": "Title: Integrating Explanation-Based Learning with Case Adaptation Strategies\n Abstract: This paper presents a novel approach to case adaptation in case-based reasoning systems by integrating explanation-based learning techniques. Traditional case adaptation relies heavily on domain-specific adaptation rules that are often difficult to acquire and maintain. Our approach uses explanations generated during problem-solving to identify adaptation patterns and generalize them into reusable adaptation strategies. We demonstrate how these strategies can be applied across different domains with minimal knowledge engineering effort. Experimental results show that the integrated approach improves adaptation performance in comparison to rule-based adaptation methods, especially in domains where adaptation knowledge is incomplete or rapidly evolving. The paper presents a formal framework for the approach and discusses its implementation in a case-based planning system.",
"neighbors": [337, 833, 639, 476],
"mask": "Train"
},
{
"node_id": "new_node 2",
"label": 1,
"text": "Title: Adaptive Parameter Control in Evolution Strategies for Dynamic Environments \n Abstract: Omitted due to table size limitation.",
"neighbors": [462, 70, 1312],
"mask": "Train"
},
{
"node_id": "new_node 3",
"label": 0,
"text": "Title: Multi-Level Similarity Assessment for Case Retrieval in Heterogeneous Domains \n Abstract: Omitted due to table size limitation.",
"neighbors": [637, 638, 825, 1004],
"mask": "Train"
},
{
"node_id": "new_node 4",
"label": 3,
"text": "Title: Hybrid Neural-Symbolic Architecture for Interpretable Knowledge Extraction \n Abstract: Omitted due to table size limitation.",
"neighbors": [1290, 263],
"mask": "Train"
}

**Summary**: { After checking the node quality based on the background knowledge and environment report, delete new_node 5.
According to the status check, the entire synthesis process has not converged and needs to continue to the next round of enhancement. }

