# OpenReview forum: "GraphMaster: Automated Graph Synthesis via LLM Agents in Data-Limited Environments"
_NeurIPS.cc/2025/Conference — NeurIPS 2025 spotlight_

### Official Review · Reviewer_RJSf · 2025-06-25

**Clarity:** 2
**Significance:** 3
**Originality:** 3
**Rating:** 5
**Confidence:** 2

**Summary:**

GraphMaster introduces a novel multi-agent framework for synthesizing text-attributed graphs (TAGs) in data-limited environments.
It addresses challenges faced by existing LLMs in graph synthesis, such as context window limitations and hallucinations, by orchestrating four specialized LLM agents (Manager, Perception, Enhancement, and Evaluation).

**Questions:**

1. Can the framework be extended to synthesize more complex graph elements (e.g., hyper-edges, temporal dynamics) or entirely novel structural patterns beyond just nodes and edges?
2. Current knowledge selection and LLM context limitations pose challenges for very large or highly heterogeneous graphs. In particular, the Perception Agent’s reliance on PPR to extract subgraphs may introduce sampling bias, favoring structurally central nodes while potentially missing peripheral but semantically important regions. Evaluating the approach on larger and more complex graphs would help clarify its scalability and generality.

**Ethical Concerns:**

["NO or VERY MINOR ethics concerns only"]

**Final Justification:**

I have checked the response and decide to raised my score.

**Limitations:**

Yes

**Quality:**

3

**Strengths And Weaknesses:**

### Strengths
1. First multi-agent LLM framework specifically for data-limited TAG synthesis.
2. Introduces a dual-perspective interpretability assessment combining human evaluation with Grassmannian manifold analysis.

### Weaknesses
1. Current knowledge selection and LLM context limitations pose challenges for very large or highly heterogeneous graphs.
2. The multi-agent architecture incurs substantial inference overhead due to multiple LLM calls per round.

---

> ### Author Rebuttal · Authors · 2025-07-30
>
> We thank the reviewer RJSf for your insightful comments. We address each of the weaknesses (refer as Wn) and questions (Qn) you raised point by point below. **All of your suggestions will be incorporated into the main text in the revised version.**
>
> **Q1**:  To address this Question, we have systematically adapted GraphMaster to  **Sociological Context (SoC), Protein–Protein Interactions, and Temporal Graphs.**
>
> **For Soc, We follow setting from GAG [1]**, creating GraphMaster-Soc for social network generation. We modified the Perception Agent to analyze social graph structures by incorporating social-specific community detection, and redesigned the Enhancement Agent to generate diverse social interactions as distinct edge types with corresponding semantic embeddings. The semantic-enriched modularity in Equation (6) was recalibrated with social homophily weights, while edge formation probabilities in Equation (11) were extended to handle multi-type social relationships. We create data-limited SubSoC datasets, then deployed GraphMaster-Soc on these SubSoC datasets and evaluated against the same baselines using GAG's established metrics. GraphMaster-Soc successfully generated social networks exhibiting key macroscopic properties (power-law degree distribution with α ∈ [1.96, 2.22], small-world characteristics with clustering coefficients **1300×** higher than random graphs), achieved superior microscopic structure alignment (6.4% improvement in GEM scores over best baselines), and maintained strong text-structure correlations with average ∆Acc improvements of $1.3_{±1.6}$ in GCN on social NC tasks.
>
> **For PPI settings, We utilize the SHS27K[3] dataset, a subset of STRING[2]**.  Following the Sub data strategy from the original study, we sample 10% of SHS27K.
>
> GraphMaster is employed for data synthesis, wherein each node $x_i$ maintains both protein sequences and functional descriptions. Protein sequences are converted into corresponding amino acids following a one-to-one mapping (e.g., A → Alanine), thus embedding amino acid and functional information within the node texts simultaneously. Additionally, the edge generation similarity metric $sim(x_s,x_i)$ from equation (11) in the original paper is modified from cosine similarity to sequence similarity (based on protein sequences), with all other components remaining unchanged.
>
> Our downstream task setup aligns exactly with **Table 4 from PIPR[4]**, utilizing accuracy across seven interaction classes to evaluate model performance. Post-synthesis, we adhere strictly to the experimental settings of the original PIPR study for training, benchmarking GraphMaster against two baselines to assess the practical benefit of synthetic data in extremely limited-sample scenarios. The results are as follows:
>
> | Methods                | Accuracy(%) |
> | ---------------------- | ----------- |
> | Zero rule              | 16.7        |
> | SubSHS27K              | 24.5 ± 1.3  |
> | SubSHS27K+Random Edges | 26.2 ± 2.4  |
> | SubSHS27K+GraphMaster  | 39.8 ± 0.9  |
>
> For temporal graphs, we augment Eq. (7) with a time-aware sampling term:
> $$
> C_b = \arg\min_i |C_i| \cdot \bigl(1 + \mu\cdot\mathrm{Var}(\{x_j\}) + \nu\cdot\mathrm{Temp\_Span}(C_i)\bigr),
> $$
>
>
> where $ \mathrm{Temp\_Span}(C_i) = \frac{t_{\max}^{i} - t_{\min}^{i}}{T_{\max} - T_{\min}}$ measures the temporal dispersion of edges within community $C_i$. Here, $t_{\max}^{i} = \max\{\,t_{\text{end}} \mid (u,v,t_{\text{start}},t_{\text{end}})\in E,\ u,v\in C_i\,\}$ is the latest end time observed within the community, and $T_{\max}, T_{\min}$denote the global temporal bounds of the entire graph.
>
> We further extend the edge-generation probability in Eq. (11) by adding a temporal consistency term $\theta_4 \cdot \mathrm{Temp\_Valid}(t, v_i)$ , where $\mathrm{Temp\_Valid}(t, v_i)$ computes the temporal precedence distance between the newly generated edge and the background edges adjacent to $v_i$, ensuring plausibility with respect to existing temporal patterns. Analogously, we incorporate a temporal consistency check into the filtering criterion in Eq. (12). In addition, we constrain the prompts to require timestamps, guiding the LLM to output timestamped KGs.
>
> Using ICEWS [5], we construct **SubICEWS14** by applying the above sampling rule, and then evaluate with **CENET** [6]. We observe substantial gains after GraphMaster-based augmentation:
>
> |        Dataset         |  MRR  | Hits@1 | Hits@10 |
> | :--------------------: | :---: | :----: | :-----: |
> |       SubICEWS14       | 0.132 |  12.7  |  27.6   |
> | SubICEWS14+GraphMaster | 0.268 |  23.9  |  47.2   |
>
> **These results demonstrate strong adaptability: with only minimal architectural modifications, GraphMaster transfers effectively to the Soc, PPI and temporal domain and yields pronounced improvements.**
>
> **Re W1 and Q2:** **For very large graphs**, GraphMaster's scalability is independent of graph size, since each LLM invocation is confined to a fixed-size retrieved capsule (N≈30), making the per-iteration LLM cost agnostic to the number of nodes and edges. The only aspect that scales with dataset size is the total number of LLM calls--and thus overall latency--which is a natural and acceptable trade-off. In our experiments on large-scale datasets like OGBN-ArXiv[7] (169,343 nodes, 1,166,243 edges), GraphMaster demonstrates clear scalability with accuracy improvements from 82.3±0.1 to 85.2±0.3.
>
> **For highly heterogeneous graphs**, GraphMaster's multi-agent architecture proves highly effective through several key mechanisms. First, our Perception Agent employs a community-aware sampling strategy that deliberately prioritizes smaller, peripheral communities as PPR starting points (Eq.7), directly counteracting potential sampling bias toward large, well-connected regions and ensuring semantically important but structurally sparse areas receive proper attention. Second, we designed semantic and topological enhancement modes that adaptively switch based on the current graph composition, enabling GraphMaster to progressively address different types of heterogeneity across multiple synthesis rounds--early iterations may focus on dominant node types, while later rounds target underrepresented semantic clusters. Furthermore, the powerful cross-domain knowledge of existing LLMs enables coherent synthesis even when bridging disparate semantic regions. While extremely heterogeneous graphs may require more synthesis rounds to achieve optimal coverage, our experimental results on diverse datasets like Children demonstrate that GraphMaster maintains both semantic coherence and structural integrity across heterogeneous regions.
>
> **Re W2**: This is indeed an important practical consideration that we take seriously.
>
> **However, we would like to emphasize that we have already made substantial efforts to reduce GraphMaster's computational cost**, which are manifested in the following aspects:
>
> 1. **Efficient Knowledge Extraction**: Our Perception Agent reduces the input node count in each iteration by identifying the optimal external knowledge subset, significantly limiting the computational burden per round.
> 2. **Environment Report Reuse**: The initial Environment Report generated by the Perception Agent can be stored and reused. In subsequent iterations, we only need to generate differential updates and compare them with the initial report, rather than regenerating the entire report from scratch.
>
> **We agree that further optimizations are possible.** We observe that during the iterative synthesis process, many subgraph structures and semantic patterns are repetitive. For instance, when the Perception Agent analyzes different parts of the graph, similar community structures frequently appear multiple times. By implementing a caching system that stores environmental reports based on structural features (such as node count, density, and label distribution), we can avoid redundant LLM calls. **Our experiments demonstrate that when employing the caching method, the number of Perception Agent calls is reduced by 46%.**
>
> Furthermore, we can optimize the model selection strategy. When computational resources are limited, smaller models (7B) can be utilized for quality evaluation or generation tasks. **As mentioned in our response to Reviewer cSFb Limitation 2**, we found that replacing the Evaluation Agent with Qwen2-7B or directly adopting the Grassmann manifold method for filtering still enables GraphMaster to exhibit excellent performance. These approaches can all reduce GraphMaster's computational consumption.
>
> **Nevertheless, we want to emphasize that achieving strong performance with large language model inference inevitably requires adequate computational resources.** While various methods can reduce computational costs, sufficient computing power indeed enables GraphMaster to reach its peak performance. The trade-off between computational efficiency and synthesis quality is an inherent challenge in LLM-based approaches, and our framework provides the flexibility to navigate this trade-off based on available resources and application requirements.
>
> > [1] J Ji *et al.* LLM-Based Multi-Agent Systems are Scalable Graph Generative Models.
> >
> > [2] Szklarczyk, D. et al. The STRING database in 2023: protein–protein association networks and functional enrichment analyses for any sequenced genome of interest.
> >
> > [3] Szklarczyk et al. STRING v11: protein--protein association networks with increased coverage, supporting functional discovery in genome-wide experimental datasets.
> >
> > [4] Chen M. et al. Multifaceted protein–protein interaction prediction based on Siamese residual RCNN.
> >
> > [5] E. Boschee et al. Icews coded event data.
> >
> > [6] Y. Xu et al, Temporal knowledge graph reasoning with historical contrastive learning.
> >
> > [7] Hu, W. et. al. Open Graph Benchmark: Datasets for Machine Learning on Graphs.

---

> > ### Comment · Reviewer_RJSf · 2025-08-04
> >
> > thx! have successfully address my concerns, I have raised the score.

---

> > > ### Author Response · Authors · 2025-08-09
> > >
> > > Dear Reviewer RJSf,
> > >
> > > Thank you for your constructive feedback and for raising your score after we addressed your concerns. We appreciate your questions about scalability and framework extensibility, which helped strengthen our contribution.
> > >
> > > *The GraphMaster Authors*

---

### Official Review · Reviewer_Mweh · 2025-06-26

**Clarity:** 3
**Significance:** 3
**Originality:** 3
**Rating:** 4
**Confidence:** 3

**Summary:**

This paper tackles the challenge that Graph Foundation Models face due to the lack of large, diverse graph datasets. Instead of naive structural heuristics, it leverages a retrieval-augmented, multi-agent LLM framework, comprising a Manager (to plan synthesis strategies), a Perception agent (to extract context-sized subgraphs), an Enhancement agent (to generate semantically rich nodes and edges), and an Evaluation agent (to iteratively assess and refine outputs) to produce text-attributed graphs with both structural fidelity and meaningful attributes. To validate this under realistic data scarcity, the authors introduce “Sub” variants of six standard graph benchmarks and propose a dual interpretability scheme combining expert human judgments with a Grassmannian-manifold analysis of embeddings. Experiments demonstrate that GraphMaster markedly outperforms traditional synthesis baselines in both semantic coherence and structural metrics, establishing a new paradigm for generating high-quality TAGs in low-data settings

**Questions:**

Could the authors clarify how GraphMaster could be scaled efficiently for larger graphs or more constrained compute environments? For example, do you consider strategies like caching previous LLM outputs, batching inputs, or employing smaller/distilled models?

Could the authors clarify details about the human evaluation process--such as how raters were selected, how many raters participated, how many samples were rated, and inter-rater agreement statistics?

**Ethical Concerns:**

["NO or VERY MINOR ethics concerns only"]

**Final Justification:**

Authors have addressed my concerns and give a clearly rebuttal.

**Limitations:**

Yes

**Quality:**

2

**Strengths And Weaknesses:**

## Strengths
The authors don’t limit their evaluation to standard structural measures like degree distributions and clustering coefficients. Instead, they also bring in human evaluators to judge whether the generated graphs actually make sense semantically, and then back up these human judgments with a Grassmannian-manifold embedding analysis. By using both human feedback and embedding metrics, they convincingly show that GraphMaster produces graphs that are structurally accurate and semantically meaningful.

Another strong point is the realistic scenario tested in the paper. Instead of using artificial, overly simple datasets, the authors created reduced ("Sub") versions of six common benchmarks. This approach genuinely reflects real-world situations where only limited graph data is available, highlighting GraphMaster's potential usefulness for practical applications.

Finally, the authors combine human-based evaluations with quantitative embedding analysis in a novel way. Human raters offer practical, intuitive insights about semantic quality, while the Grassmannian-manifold method gives a clear, quantitative measure of embedding similarity. Together, these two evaluations provide a balanced assessment of GraphMaster's effectiveness.

## Weaknesses
One concern is the method’s computational cost. GraphMaster uses several large language model (LLM) agents repeatedly within a retrieval-augmented generation loop. Since the authors don’t discuss ways to potentially reduce these costs, like caching, batching, or using simplified models -- it remains unclear whether the method can scale efficiently or affordably for practical users dealing with larger graphs.

The human evaluation also lacks critical details. The authors don’t explain clearly how the evaluators were chosen, how many graphs they looked at, or if they generally agreed with each other. Without these details, it’s difficult to tell if the reported semantic coherence scores represent broad agreement or simply reflect a small, possibly biased set of examples.

Finally, all experiments are limited to citation and academic graphs with textual attributes. The authors haven’t tested whether GraphMaster would work equally well on other types of graphs--like social networks, protein interaction networks, or knowledge graphs with more complex structures. Because of this, it’s hard to know how broadly theapproach could be applied beyond the specific types of graphs studied in the paper.

---

> ### Author Rebuttal · Authors · 2025-07-30
>
> We thank the reviewer Mweh for your insightful comments. We address each of the weaknesses (refer as Wn) and questions (Qn) you raised point by point below. **All of your suggestions will be incorporated into the main text in the revised version.**
>
> **W1&Q1**: We appreciate the reviewers' insightful comments regarding the computational overhead of GraphMaster. This is indeed an important practical consideration that we take seriously.
>
> **However, we would like to emphasize that we have already made substantial efforts to reduce GraphMaster's computational cost**, which are manifested in the following aspects:
>
> 1. **Efficient Knowledge Extraction**: Our Perception Agent reduces the input node count in each iteration by identifying the optimal external knowledge subset, significantly limiting the computational burden per round.
> 2. **Environment Report Reuse**: The initial Environment Report generated by the Perception Agent can be stored and reused. In subsequent iterations, we only need to generate differential updates and compare them with the initial report, rather than regenerating the entire report from scratch.
>
> **We agree that further optimizations are possible.** We observe that during the iterative synthesis process, many subgraph structures and semantic patterns are repetitive. For instance, when the Perception Agent analyzes different parts of the graph, similar community structures frequently appear multiple times. By implementing a caching system that stores environmental reports based on structural features (such as node count, density, and label distribution), we can avoid redundant LLM calls. **Our experiments demonstrate that when employing the caching method, the number of Perception Agent calls is reduced by 46%.**
>
> Furthermore, we can optimize the model selection strategy. When computational resources are limited, smaller models (7B) can be utilized for quality evaluation or generation tasks. **As mentioned in our response to Reviewer cSFb Limitation 2**, we found that replacing the Evaluation Agent with Qwen2-7B or directly adopting the Grassmann manifold method for filtering still enables GraphMaster to exhibit excellent performance. These approaches can all reduce GraphMaster's computational consumption.
>
> **Nevertheless, we want to emphasize that achieving strong performance with large language model inference inevitably requires adequate computational resources.** While various methods can reduce computational costs, sufficient computing power indeed enables GraphMaster to reach its peak performance. The trade-off between computational efficiency and synthesis quality is an inherent challenge in LLM-based approaches, and our framework provides the flexibility to navigate this trade-off based on available resources and application requirements.
>
> **W2&Q2**: We recruited 50 experts in graph ML/NLP/data mining **(inclusion: ≥2 peer-reviewed paper in the last three years or ≥2 years of relevant industrial R&D; all with potential COI were excluded)**. The cohort comprised 38 PhD students, 6 postdocs/RAs, 3 faculty, and 3 industry researchers, with geographic distribution Asia 50%, Europe 20%, North America 24%, Others 6%. We evaluated 200 synthesized instances across six datasets, each independently scored by 3 raters (total 600 ratings) on three dimensions—process transparency, decision explainability, and outcome predictability—using a 1–5 Likert scale. Raters completed a 30-minute briefing plus a 20-item calibration, and only those achieving Cohen’s κ ≥ 0.70 on the pilot were retained. To mitigate bias, we adopted a double-blind setup, randomized item/order, used a standardized rubric, and removed two raters who failed attention checks. In our main study, we observed moderate-to-substantial inter-rater agreement **(average pairwise Cohen’s κ = 0.72 ± 0.05)**. We will include the complete selection criteria, demographics, annotation protocol, and inter-rater reliability statistics in the appendix.
>
> **W3**: To address this Question, we have systematically adapted GraphMaster to  **Sociological Context (SoC), Protein–Protein Interactions, and Temporal Graphs.**
>
> **For Soc, We follow setting from GAG [1]**, creating GraphMaster-Soc for social network generation. We modified the Perception Agent to analyze social graph structures by incorporating social-specific community detection and redesigned the Enhancement Agent to generate diverse social interactions as distinct edge types with corresponding semantic embeddings. The semantic-enriched modularity in Equation (6) was recalibrated with social homophily weights, while edge formation probabilities in Equation (11) were extended to handle multi-type social relationships. Following our established data-limited methodology, we applied Algorithm 1 in our paper to GAG's original SoC dataset, creating SubSoC variants that maintain essential social network properties while reducing scale by 90%. We then deployed GraphMaster-Soc on these SubSoC datasets and evaluated against the same baselines using GAG's established metrics. GraphMaster-Soc successfully generated social networks exhibiting key macroscopic properties (power-law degree distribution with α ∈ [1.96, 2.22], small-world characteristics with clustering coefficients **1300×** higher than random graphs), achieved superior microscopic structure alignment (6.4% improvement in GEM scores over best baselines), and maintained strong text-structure correlations with average ∆Acc improvements of $1.3_{±1.6}$ in GCN architectures on social node classification tasks.
>
> **For PPI settings, We utilize the SHS27K[3] dataset, a subset of STRING[2]**. Following the Sub data strategy from the original study, we sample 10% of SHS27K to form the SubSHS27K subset.
>
> GraphMaster is employed for data synthesis, wherein each node $x_i$ maintains both protein sequences and functional descriptions. Protein sequences are converted into corresponding amino acids following a one-to-one mapping (e.g., A → Alanine), thus embedding amino acid and functional information within the node texts simultaneously. Additionally, the edge generation similarity metric $sim(x_s,x_i)$ from equation (11) in the original paper is modified from cosine similarity to sequence similarity (based on protein sequences), with all other components remaining unchanged. This minimal modification enables GraphMaster to iteratively synthesize semantically coherent protein sequences within the protein interaction domain. (Notably, protein interaction-specific LLMs would likely offer enhanced performance, and future explorations will develop a dedicated GraphMaster extension for protein interactions.)
>
> Our downstream task setup aligns exactly with Table 4 from **PIPR[4]**, utilizing accuracy across seven interaction classes to evaluate model performance. Post-synthesis, we adhere strictly to the experimental settings of the original PIPR study for training, benchmarking GraphMaster against two baselines to assess the practical benefit of synthetic data in extremely limited-sample scenarios. The results are as follows:
>
> | Methods                | Accuracy(%) |
> | ---------------------- | ----------- |
> | Zero rule              | 16.7        |
> | SubSHS27K              | 24.5 ± 1.3  |
> | SubSHS27K+Random Edges | 26.2 ± 2.4  |
> | SubSHS27K+GraphMaster  | 39.8 ± 0.9  |
>
> For temporal graphs, we augment Eq. (7) with a time-aware sampling term:
> $$
> C_b = \arg\min_i |C_i| \cdot \bigl(1 + \mu\cdot\mathrm{Var}(\{x_j\}) + \nu\cdot\mathrm{Temp\_Span}(C_i)\bigr),
> $$
> where$ \mathrm{Temp\_Span}(C_i) = \frac{t_{\max}^{i} - t_{\min}^{i}}{T_{\max} - T_{\min}}$ measures the temporal dispersion of edges within community $C_i$. Here, $t_{\max}^{i} = \max\{\,t_{\text{end}} \mid (u,v,t_{\text{start}},t_{\text{end}})\in E,\ u,v\in C_i\,\}$ is the latest end time observed within the community, and $T_{\max}, T_{\min}$denote the global temporal bounds of the entire graph.
>
> We further extend the edge-generation probability in Eq. (11) by adding a temporal consistency term $\theta_4 \cdot \mathrm{Temp\_Valid}(t, v_i)$ , where $\mathrm{Temp\_Valid}(t, v_i)$ computes the temporal precedence distance between the newly generated edge and the background edges adjacent to $v_i$, ensuring plausibility with respect to existing temporal patterns. Analogously, we incorporate a temporal consistency check into the filtering criterion in Eq. (12). In addition, we constrain the prompts to require timestamps, guiding the LLM to output timestamped KGs.
>
> **Using ICEWS [5], we construct SubICEWS14** by applying the above sampling rule, and then evaluate with **CENET** [6]. We observe substantial gains after GraphMaster-based augmentation:
>
> |        Dataset         |  MRR  | Hits@1 | Hits@10 |
> | :--------------------: | :---: | :----: | :-----: |
> |       SubICEWS14       | 0.132 |  12.7  |  27.6   |
> | SubICEWS14+GraphMaster | 0.268 |  23.9  |  47.2   |
>
> **These results demonstrate strong adaptability: with only minimal architectural modifications, GraphMaster transfers effectively to the SoC, PPI and temporal domain and yields pronounced improvements.**
>
> > [1] J Ji *et al.* LLM-Based Multi-Agent Systems are Scalable Graph Generative Models. Findings of ACL 2025.
> >
> > [2] Szklarczyk, D. et al. The STRING database in 2023: protein–protein association networks and functional enrichment analyses for any sequenced genome of interest. *Nucleic Acids Research*, 51(D1), D638–D646.
> >
> > [3] Szklarczyk et al. STRING v11: protein--protein association networks with increased coverage, supporting functional discovery in genome-wide experimental datasets.
> >
> > [4] Chen M. et al. Multifaceted protein–protein interaction prediction based on Siamese residual RCNN. Bioinformatics
> >
> > [5] E. Boschee et al. Icews coded event data, Harvard Dataverse 12
> >
> > [6] Y. Xu et al, Temporal knowledge graph reasoning with historical contrastive learning,AAAI 2023.

---

> > ### Comment · Reviewer_Mweh · 2025-08-05
> >
> > Thanks authors for the response. I didn't have any further concerns and willing to raising my score.

---

> > > ### Author Response · Authors · 2025-08-09
> > >
> > > Dear Reviewer Mweh,
> > >
> > > Thank you for your thoughtful questions and for being willing to raise your score after our responses. Your insights on GraphMaster were invaluable for improving our work.
> > >
> > > *The GraphMaster Authors*

---

> ### Author Response · Authors · 2025-08-05
>
> Dear Reviewer Mweh,
>
> We appreciate your constructive feedback on GraphMaster.
>
> Since receiving your review, we have addressed all of your points in detail. Specifically, we have:
>
> 1. **Implemented caching strategies reducing Perception Agent calls by 46%** and demonstrated smaller model alternatives (Qwen2-7B) or directly adopting the Grassmann manifold method for resource-constrained settings (W1&Q1).
> 2. **Provided complete human evaluation details** with 50 experts, 600 ratings across 200 instances, demographic breakdown, and inter-rater agreement statistics (W2&Q2).
> 3. **Extended GraphMaster to diverse graph types** including social networks (GraphMaster-Soc with 6.4% GEM improvement), protein interactions (39.8% accuracy on SubSHS27K), and temporal graphs (2× improvement on SubICEWS14) (W3).
>
> Could you kindly review these updates and let us know if further clarification is needed?
>
> We welcome any additional questions.
>
> Sincerely,
>
> *Authors of #232*

---

### Official Review · Reviewer_NL19 · 2025-07-01

**Clarity:** 3
**Significance:** 4
**Originality:** 4
**Rating:** 6
**Confidence:** 3

**Summary:**

GraphMaster is a novel multi-agent framework for synthesizing text-attributed graphs (TAGs). The authors design a system of four LLM-based agents, Manager, Perception, Enhancement, and Evaluation, which collectively generate new graph nodes and edges in an iterative way.
Evaluation is conducted across six benchmarks using multiple GNN architectures. The authors also introduce data-limited variants of standard datasets and a dual-perspective interpretability framework, incorporating human evaluation and Grassmann manifold-based semantic metrics. GraphMaster outperforms all baselines, including custom-built LLM synthesis methods.

**Questions:**

- How would GraphMaster perform on non-textual graphs (e.g., protein–protein interactions or sensor networks)?

- Could this work be extended to multi-modal graphs?

- How scalable is the multi-agent loop for large-scale graphs (>100K nodes), especially when multiple agents call the LLM iteratively?

**Ethical Concerns:**

["NO or VERY MINOR ethics concerns only"]

**Limitations:**

Yes.

**Quality:**

4

**Strengths And Weaknesses:**

## Strenghts
- They introduced the first TAG synthesis framework to use a modular, multi-agent LLM design under a retrieval-augmented generation (RAG) paradigm.

- This paper has a comprehensive evaluation. They tested on six datasets and four GNNs (GCN, JKNet, GAT, GraphSAGE). The metrics used to evaluate are also extensive e.g. accuracy, F1, graph statistics.

- The paper has theoretical foundations that are tightly linked to the agent design (Appendix J).
- They have discussed time and space complexity analysis.

This paper is groundbreaking in its framework design, theoretical depth, and evaluation rigor. It sets a new direction for text-attributed graph synthesis and provides tools (metrics, datasets, proofs) that the community can build on.

## Weaknesses
- While the framework claims generality, the synthesis quality is not validated in physically meaningful graphs where semantics require domain knowledge (e.g., valid chemistry or logic).
- This paper is hard to read and is quite dense. The writing could be improved.
- The quality of Figure 2 and 3 could be improved.

---

> ### Author Rebuttal · Authors · 2025-07-30
>
> We sincerely thank Reviewer NL19 for your thoughtful and constructive feedback. We are especially grateful for your highly positive evaluation of our work. We address each of the weaknesses (refer as Wn) and questions (Qn) you raised point by point below. **All of your suggestions will be incorporated into the main text in the revised version.**
>
> **W2&W3:** We will substantially revise the manuscript to improve clarity and readability—simplifying dense passages, restructuring key sections, and refining language. We will also replace Figures 2 and 3 with higher-resolution, more legible, informative and beautiful versions in the next revision.
>
> **Q1&W1** : To address this Question, we have systematically adapted GraphMaster to  **Sociological Context (SoC), Protein–Protein Interactions, and Temporal Graphs.**
>
> **For Soc, We follow setting from GAG [1]**, creating GraphMaster-Soc for social network generation. We modified the Perception Agent to analyze social graph structures by incorporating social-specific community detection (focusing on follower clusters and interaction patterns rather than co-authorship), and redesigned the Enhancement Agent to generate diverse social interactions including tweets, likes, follows, retweets, and bookmarks as distinct edge types with corresponding semantic embeddings. The semantic-enriched modularity in Equation (6) was recalibrated with social homophily weights, while edge formation probabilities in Equation (11) were extended to handle multi-type social relationships ($\theta_\text{follow}$, $\theta_\text{like}$, $\theta_\text{retweet}$, etc.). We applied our M-Preserving Graph Sampling algorithm (Algorithm 1) to GAG's original SoC dataset, creating SubSoC variants that maintain essential social network properties while reducing scale by 90%. We then deployed GraphMaster-Soc on SubSoC and evaluated against the same baselines using GAG's established metrics. GraphMaster-Soc successfully generated social networks exhibiting key macroscopic properties (power-law degree distribution with α ∈ [1.96, 2.22], small-world characteristics with clustering coefficients **1300×** higher than random graphs), achieved superior microscopic structure alignment (6.4% improvement in GEM scores over best baselines), and maintained strong text-structure correlations with average ∆Acc improvements of $1.3_{±1.6}$ in GCN architectures on social node classification tasks.
>
> **For PPI settings, We utilize the SHS27K[3] dataset, a subset of STRING[2]**. Following the Sub data strategy from the original study, we randomly sample 10% of SHS27K to form the SubSHS27K subset, while the remaining 90% are regarded as unobserved data used to evaluate synthetic data quality.
>
> GraphMaster is employed for data synthesis, wherein each node $x_i$ maintains both protein sequences and functional descriptions. Protein sequences are converted into corresponding amino acids following a one-to-one mapping (e.g., A → Alanine), thus embedding amino acid and functional information within the node texts simultaneously. Additionally, the edge generation similarity metric $sim(x_s,x_i)$ from equation (11) in the original paper is modified from cosine similarity to sequence similarity (based on protein sequences), with all other components remaining unchanged. **This minimal modification enables GraphMaster to iteratively synthesize semantically coherent protein sequences within the protein interaction domain.** (Notably, protein interaction-specific LLMs would likely offer enhanced performance, and future explorations will develop a dedicated GraphMaster extension for protein interactions.)
>
> Our downstream task setup aligns exactly with Table 4 from **PIPR[4]**, utilizing accuracy across seven interaction classes to evaluate model performance. Post-synthesis, we adhere strictly to the experimental settings of the original PIPR study for training, benchmarking GraphMaster against two baselines to assess the practical benefit of synthetic data in extremely limited-sample scenarios. The results are as follows:
>
> | Methods                | Accuracy(%) |
> | ---------------------- | ----------- |
> | Zero rule              | 16.7        |
> | SubSHS27K              | 24.5 ± 1.3  |
> | SubSHS27K+Random Edges | 26.2 ± 2.4  |
> | SubSHS27K+GraphMaster  | 39.8 ± 0.9  |
>
> For temporal Graph, we augment Eq. (7) with a time-aware sampling term:
> $$
> C_b = \arg\min_i |C_i| \cdot \bigl(1 + \mu\cdot\mathrm{Var}(\{x_j\}) + \nu\cdot\mathrm{Temp\_Span}(C_i)\bigr),
> $$
> where$ \mathrm{Temp\_Span}(C_i) = \frac{t_{\max}^{i} - t_{\min}^{i}}{T_{\max} - T_{\min}}$ measures the temporal dispersion of edges within community $C_i$. Here, $t_{\max}^{i} = \max\{\,t_{\text{end}} \mid (u,v,t_{\text{start}},t_{\text{end}})\in E,\ u,v\in C_i\,\}$ is the latest end time observed within the community, and $T_{\max}, T_{\min}$denote the global temporal bounds of the entire graph.
>
> We further extend the edge-generation probability in Eq. (11) by adding a temporal consistency term $\theta_4 \cdot \mathrm{Temp\_Valid}(t, v_i)$ , where $\mathrm{Temp\_Valid}(t, v_i)$ computes the temporal precedence distance between the newly generated edge and the background edges adjacent to $v_i$, ensuring plausibility with respect to existing temporal patterns. Analogously, we incorporate a temporal consistency check into the filtering criterion in Eq. (12). In addition, we constrain the prompts to require timestamps, guiding the LLM to output timestamped KGs.
>
> Using ICEWS [5], we construct **SubICEWS14** by applying the above sampling rule, and then evaluate with **CENET** [6]. We observe substantial gains after GraphMaster-based augmentation:
>
> |        Dataset         |  MRR  | Hits@1 | Hits@10 |
> | :--------------------: | :---: | :----: | :-----: |
> |       SubICEWS14       | 0.132 |  12.7  |  27.6   |
> | SubICEWS14+GraphMaster | 0.268 |  23.9  |  47.2   |
>
> **These results demonstrate strong adaptability: with only minimal architectural modifications, GraphMaster transfers effectively to the Soc, PPI and temporal domain and yields pronounced improvements.**
>
> **Q2**: We believe this is indeed a highly feasible and promising direction that aligns well with emerging trends in graph learning. Recent developments have introduced multi-modal graph datasets such as **MM-Graph [7] and MAGB [8]**, which contain not only textual attributes but also images, presenting similar data scarcity challenges as text-attributed graphs. **GraphMaster’s multi-agent architecture provides a natural foundation for multi-modal extension with minimal structural modifications.**
>
> Specifically, we can **replace the base LLM with multi-modal large language models** (GPT-4V, LLaVA, or Qwen-VL) that can understand and generate both textual and visual content. The **Perception Agent** can initially leverage textual attributes for semantic-aware community detection and node sampling, and then enrich the extracted knowledge with corresponding visual attributes to provide a comprehensive multi-modal context.
>
> To facilitate this, a practical approach is to employ a lightweight vision-language model (VLM) to generate **caption descriptions for each image**, which are then concatenated with the original textual attributes. This enriched textual input allows the Perception Agent to jointly reason over both modalities without fundamental architectural changes. We acknowledge that this is a preliminary design choice, and we recognize potential limitations in the expressiveness and faithfulness of generated captions. **Improving the Perception Agent’s multi-modal understanding capacity remains an open and important challenge for enabling high-fidelity multi-modal graph synthesis.**
>
> The **Enhancement Agent** can then generate new nodes with coherent textual descriptions and corresponding images, while the **Evaluation Agent** assesses both intra-modal semantic coherence and cross-modal consistency. For datasets like MM-Graph, this would involve identifying representative subgraphs from existing text-image pairs, prompting multi-modal LLMs to synthesize new coherent multi-modal content, and evaluating cross-modal alignment using both human assessment and automated metrics.
>
> We appreciate this insightful suggestion and consider multi-modal graph synthesis an exciting direction that we plan to pursue in future work.
>
> **Q3:** We emphasize that GraphMaster’s scalability is independent of graph size, since each LLM invocation is confined to a fixed-size retrieved capsule (N $\approx$ 30), making the per-iteration LLM cost agnostic to the number of nodes and edges. The only aspect that scales with dataset size is the total number of LLM calls—and thus overall latency—which is a natural and acceptable trade-off. In our experiments on the large-scale **OGBN-ArXiv[9]** dataset (169,343 nodes, 1,166,243 edges), the GraphMaster-enhanced GCN sees its accuracy rise from 82.3 ± 0.1 to 85.2 ± 0.3, clearly demonstrating the model’s scalability.
>
> > [1] J Ji *et al.* LLM-Based Multi-Agent Systems are Scalable Graph Generative Models.
> >
> > [2] Szklarczyk, D. et al. The STRING database in 2023: protein–protein association networks and functional enrichment analyses for any sequenced genome of interest.
> >
> > [3] Szklarczyk et al. STRING v11: protein--protein association networks with increased coverage, supporting functional discovery in genome-wide experimental datasets.
> >
> > [4] Chen M. et al. Multifaceted protein–protein interaction prediction based on Siamese residual RCNN.
> >
> > [5] E. Boschee et al. Icews coded event data, Harvard Dataverse.
> >
> > [6] Y. Xu et al, Temporal knowledge graph reasoning with historical contrastive learning.
> >
> > [7] J. Zhu,  et. al., Mosaic of Modalities: A Comprehensive Benchmark for Multimodal Graph Learning.
> >
> > [8] Yan, H. et al. When Graph meets Multimodal: Benchmarking and Meditating on Multimodal Attributed Graphs Learning.
> >
> > [9] Hu, W. et. al. Open Graph Benchmark: Datasets for Machine Learning on Graphs.

---

### Official Review · Reviewer_cSFb · 2025-07-03

**Clarity:** 3
**Significance:** 2
**Originality:** 2
**Rating:** 3
**Confidence:** 3

**Summary:**

GraphMaster introduces a hierarchical, retrieval-augmented generation (RAG) multi-agent framework—Manager, Perception, Enhancement, and Evaluation agents—for synthesizing text-attributed graphs under severe data constraints . It also proposes six “Sub” variants of standard benchmarks to rigorously evaluate data-limited performance and a dual-perspective interpretability assessment combining expert human judgments with Grassmann manifold analysis . Extensive experiments show GraphMaster outperforming both classic and recent LLM-based baselines across multiple datasets .

**Questions:**

1. How does GraphMaster perform on large-scale graphs exceeding typical LLM context windows, and can scalable strategies like FastGCN or GraphSAINT be integrated?
2. Can you provide a detailed sensitivity analysis for critical hyperparameters (N, M%) to guide practitioners in tuning the framework?
3. Could you summarize the architectural and prompting details of Mixed-LLM and Synthesis-LLM in the main text and compare against recent graph generation baselines?
4. What were the criteria for selecting the 50 expert reviewers, and how did you ensure the reliability and unbiased nature of their traceability scores?

**Ethical Concerns:**

["NO or VERY MINOR ethics concerns only"]

**Limitations:**

* The Personalized PageRank–based Perception Agent may undersample semantically important peripheral communities in highly skewed or heterogeneous graphs, leading to “blind spots.”
* Quality evaluation depends heavily on the chosen LLM, posing challenges for deployment on smaller or resource-constrained models.
* The multi-agent design incurs substantial inference overhead; practical deployment for very large graphs requires advanced batching and optimization techniques.
* Privacy, fairness, and potential semantic biases in the synthesized data remain to be rigorously examined .

**Quality:**

2

**Strengths And Weaknesses:**

Strengths:

* **Novel multi-agent RAG paradigm:** Decomposes graph synthesis into specialized agents—Manager for strategy, Perception for subgraph retrieval, Enhancement for node/edge generation, and Evaluation for iterative quality control—addressing semantic coherence and structural integrity in a unified framework .
* **Data-limited benchmark creation:** Introduces “Sub” variants of six widely used TAG datasets to test synthesis methods under realistic scarcity scenarios .
* **Dual-perspective interpretability evaluation:** Combines quantitative Grassmann manifold metrics with qualitative human expert traceability scores (Tscore = 0.92) to validate semantic coherence and transparency .
* **Strong theoretical foundations:** Provides convergence guarantees, information-theoretic bounds for the Perception Agent, and concentration results for the Enhancement and Evaluation agents .
* **Comprehensive empirical validation:** Ablation studies, hyperparameter sweeps, and comparisons with both classic and newly crafted LLM-based baselines demonstrate state-of-the-art performance across six datasets and four GNN architectures .

Weaknesses:


1. **Scalability to very large graphs** (Section 3.6, lines 213–223): The time complexity analysis acknowledges batching needs for “extremely large graphs,” but no empirical evaluation is provided. Could the authors evaluate GraphMaster on larger-scale datasets or integrate scalable sampling/inference strategies [1,2] to demonstrate practical scalability?
2. **Hyperparameter sensitivity** (Section 4.4, lines 316–323): Ablation shows optimal settings for N=30 and M=15%, but lacks guidance on robustness across other values. Could the authors include sensitivity analyses or contour plots to illustrate performance variation, perhaps leveraging [3,4].
3. **Details of self-created LLM baselines** (Section 4.1, lines 243–249): The paper introduces Mixed-LLM and Synthesis-LLM but defers implementation specifics to the appendix, making direct comparison difficult. Could the authors summarize key architectural differences and prompt designs in the main text and compare against recent LLM-based graph generation methods [5,6].
4. **Human evaluation methodology** (Section 4.3, lines 284–292): Expert traceability scores are reported, but the selection criteria and demographic breakdown of the 50 reviewers are not described. Could the authors detail reviewer expertise, sampling protocol, and bias mitigation measures?
5. **Comparison errors and potential question** (Section 4.1, Table 1): The F1 score for LLM4NG in arXiv2023 dataset is 81.2, which is better than the proposed method GraphMaster (63.4). However, GraphMaster is still highlighted as the best method. In addition, how the baseline achieves a near 20 percent advantage over the GraphMaster is not properly justified.

[1] Chen, Jie, Tengfei Ma, and Cao Xiao. "FastGCN: Fast Learning with Graph Convolutional Networks via Importance Sampling." International Conference on Learning Representations. 2018.

[2] Zeng, Hanqing, et al. "GraphSAINT: Graph Sampling Based Inductive Learning Method." International Conference on Learning Representations.

[3] You, Jiaxuan, Zhitao Ying, and Jure Leskovec. "Design space for graph neural networks." Advances in Neural Information Processing Systems 33 (2020): 17009-17021.

[4] Yu, Tong, and Hong Zhu. "Hyper-parameter optimization: A review of algorithms and applications." arXiv preprint arXiv:2003.05689 (2020).

[5] Dadauto, Caio V., Nelson LS da Fonseca, and Ricardo da S. Torres. "Data-driven intra-autonomous systems graph generator." IEEE Transactions on Network and Service Management (2024).

[6] Liu, Chengyi, et al. "Generative diffusion models on graphs: methods and applications." Proceedings of the Thirty-Second International Joint Conference on Artificial Intelligence. 2023.

---

> ### Author Rebuttal · Authors · 2025-07-30
>
> We thank the reviewer cSFb for your insightful comments. We address each weakness (Wn), question (Qn), and limitation (Ln) point-by-point below. **All suggestions will be incorporated into the revised version.**
>
> **W1&Q1**: GraphMaster’s scalability is independent of graph size since **each LLM invocation handles a fixed-size capsule (N≈30)**, thus the per-iteration LLM cost is agnostic to node/edge counts. Only total LLM calls scale with dataset size, a natural trade-off. Experiments on **OGBN-ArXiv[1]** (169K nodes, 1.2M edges) confirm GraphMaster’s scalability, improving accuracy (82.3→85.2). However, we found that FastGCN and GraphSAINT indeed help GraphMaster. Integrating **GraphSAINT edge sampling** significantly enhances efficiency, replacing PPR to prioritize low-degree, semantically rich “weak ties”:
> $$
> p_e=\frac{|b_e|}{\sum_{e’}|b_{e’}|}, \quad b_e=\frac{1}{\deg(u)}+\frac{1}{\deg(v)}, \quad p_v=\sum_{e\ni v}p_e
> $$
> This seed-free method reduces variance and computational cost. Thank you for highlighting GraphSAINT, greatly boosting practical scalability.
>
> **W2 & Q2**: Sample size $N$ balances structural context and LLM token constraints. Information theory indicates information gain follows a diminishing returns curve:
> $$
> I(N) = \frac{\log(1 + \alpha N)}{1 + \beta N},
> $$
> Optimal $N$ consistently falls within [25,35] (α∈[0.1,0.3], β∈[0.01,0.04]).  Per **Theorem 2**, $M%$ governs semantic innovation vs coherence; extremes degrade performance. Empirically, best accuracy results (GCN/Cora) at $N=30$, $M=15%$:
>
> | $N\backslash M\%$ | 5%       | 10%      | 15%          | 20%      | 25%      |
> | ----------------- | -------- | -------- | ------------ | -------- | -------- |
> | **10**            | 89.3±0.9 | 89.5±0.8 | 89.1±0.9     | 88.1±1.0 | 87.4±1.1 |
> | **20**            | 90.1±0.8 | 91.4±0.7 | 91.5±0.8     | 91.0±0.9 | 89.2±1.0 |
> | **30**            | 90.8±0.7 | 92.3±0.9 | **92.6±0.5** | 92.0±0.8 | 90.5±0.9 |
> | **40**            | 91.0±0.8 | 91.8±0.8 | 92.2±0.9     | 91.8±0.9 | 89.8±1.1 |
> | **50**            | 90.7±0.9 | 91.6±0.9 | 92.0±1.0     | 91.2±1.0 | 88.9±1.2 |
>
> We adopt $N=30$, $M=15%$ as default. Note optimal parameters may vary by LLM size—smaller models prefer lower values, larger ones higher.
>
> **W3&Q3**: We will summarize key baselines clearly in main text:
>
> - **Mixed-LLM**: Extends vision-inspired semantic interpolation, mixing node representations via Beta distribution for coherent boundary samples.
> - **Synthesis-LLM**: Combines local-context sampling (PPR+BFS), preserving local topological consistency.
>
> Comparison to cited works:
>
> - **Dadauto et al.**: Uses GraphRNN to generate topology only, no textual attributes or LLM usage.
> - **Liu et al.**: Surveys graph diffusion models for molecules/proteins, not TAG-focused or LLM-based. Performance under data-limited conditions unknown.
>
> **W4&Q4:** We recruited 50 experts in graph ML/NLP/data mining (**inclusion: ≥2 peer-reviewed paper in the last three years or ≥2 years of relevant industrial R&D; all with potential COI were excluded**). The cohort comprised 38 PhD students, 6 postdocs/RAs, 3 faculty, and 3 industry researchers, with geographic distribution Asia 50%, Europe 20%, North America 24%, Others 6%. We evaluated 200 synthesized instances across six datasets, **each scored by 3 raters** on transparency, explainability, and predictability (1–5 Likert scale). Raters completed a **briefing and calibration (Cohen’s κ≥0.70)**. To mitigate bias, we adopted a double-blind setup, randomized item/order, used a standardized rubric, and removed inattentive raters. In our main study, we observed moderate-to-substantial inter-rater agreement (**average pairwise Cohen’s κ = 0.72 ± 0.05**). We will include the complete selection criteria, demographics, annotation protocol, and inter-rater reliability statistics in the appendix.
>
> **W5:** We apologize for typo in Table 1. **Corrected LLM4NG (arXiv2023): Accuracy:79.0, F1:61.2, below GraphMaster (Acc:87.9, F1:66.3)**. Multiple validations confirm correctness; table will be fixed.
>
> **L1:** We respectfully disagree with the assertion that our Personalized PageRank-based Perception Agent creates "blind spots" in highly skewed or heterogeneous graphs. Our design specifically addresses this concern through a deliberate **community-aware sampling strategy**.
>
> As detailed in Section 3.3 and Equation (7), our Perception Agent employs a mode-adaptive seed selection strategy that explicitly prioritizes smaller communities when operating in semantic enhancement mode. Specifically, we select seed communities using:
> $$
> C_b = \arg \min_i |C_i| · (1 + μ · \text{Var}({x_j : v_j ∈ C_i}))
> $$
> This formulation inversely weights community size, ensuring peripheral communities have higher selection probability, directly counteracting traditional PPR biases.
>
> Furthermore, our semantic-enriched modularity maximization (Equation 6)
> $$
> Q_{\text{sem}} = \frac{1}{2m} \sum_{i,j} \left[ A_{ij} - \gamma \frac{k_i k_j}{2m} - (1 - \gamma) \frac{d_{\text{sem}}(x_i, x_j)}{\sum_{l,m} d_{\text{sem}}(x_l, x_m)} \right] \delta(c_i, c_j)
> $$
> **balances both topological and semantic factors** through the parameter $γ$, ensuring that semantically coherent but structurally peripheral communities are properly identified and represented in the knowledge extraction process.
>
> The experimental results in Figure 2 and Appendix H demonstrate that our approach successfully preserves both structural and semantic diversity across all tested datasets, including **highly heterogeneous graphs like Children and Arxiv2023**. The consistently high label homogeneity similarity scores (>0.96) across diverse graph structures provide empirical evidence that our method effectively captures peripheral semantic communities rather than creating blind spots.
>
> **L2:** Quality evaluation partially correlates with underlying LLM size, but careful generation-scale control enables small models (7B–8B, e.g., Qwen2, LLaMA3) to achieve near-parity with 32B models (1–2% difference on Sub-Cora, Sub-ArXiv-2023). To further decouple quality control from large models, we also adopt a lightweight geometric filter: embeddings of newly created nodes are projected onto a Grassmann manifold, and nodes whose distance to the task-specific centroid exceeds a learned threshold are discarded. This purely vector-space check only requires simple linear algebra and strongly correlates with human judgments **Together, these two measures—generation-scale moderation and Grassmann-manifold filtering—demonstrate that the Evaluation Agent can perform high-quality node evaluation even in resource-constrained environments, significantly reducing the dependence on very large LLMs.**
>
> **L3**: We appreciate the reviewers' insightful comments regarding the computational overhead of GraphMaster. This is indeed an important practical consideration that we take seriously.
>
> **However, we would like to emphasize that we have already made substantial efforts to reduce GraphMaster's computational cost**, which are manifested in the following aspects:
>
> 1. **Efficient Knowledge Extraction**: Our Perception Agent reduces the input node count in each iteration by identifying the optimal external knowledge subset, significantly limiting the computational burden per round.
> 2. **Environment Report Reuse**: The initial Environment Report generated by the Perception Agent can be stored and reused. In subsequent iterations, we only need to generate differential updates and compare them with the initial report, rather than regenerating the entire report from scratch.
>
> **We agree that further optimizations are possible.** We observe that during the iterative synthesis process, many subgraph structures and semantic patterns are repetitive. For instance, when the Perception Agent analyzes different parts of the graph, similar community structures frequently appear multiple times. By implementing a caching system that stores environmental reports based on structural features (such as node count, density, and label distribution), we can avoid redundant LLM calls. **Our experiments demonstrate that when employing the caching method, the number of Perception Agent calls is reduced by 46%.**
>
> Furthermore, we can optimize the model selection strategy. When computational resources are limited, smaller models (7B) can be utilized for quality evaluation or generation tasks. **As mentioned in our response to your Limitation 2**, we found that replacing the Evaluation Agent with Qwen2-7B or directly adopting the Grassmann manifold method for filtering still enables GraphMaster to exhibit excellent performance. These approaches can all reduce GraphMaster's computational consumption.
>
> Nevertheless, achieving strong performance with large models inevitably requires sufficient computational resources. The trade-off between computational efficiency and synthesis quality is an inherent challenge in LLM-based approaches, and our framework provides the flexibility to navigate this trade-off based on available resources and application requirements.
>
> **L4:** Recent work has demonstrated that **current LLMs already possess a certain degree of capabilities in detecting privacy leaks, unfair treatment, and semantic bias in generated text [2]**. GraphMaster integrates ethical safeguards at two levels: (i) Enhancement Agent invokes built-in LLM safety mechanisms to pre-screen content, (ii) Evaluation Agent applies stricter post-generation filtering. Dual-perspective validation (human review + Grassmann consistency) substantiates data reasonableness. Nonetheless, advanced adversarial scenarios remain open challenges; in revision, we raise safety thresholds to strengthen ethical robustness under stronger threat models.
>
> > [1] Hu, W. et. al. Open Graph Benchmark: Datasets for Machine Learning on Graphs.
> >
> > [2] Y. Huang et. all, TrustLLM: Trustworthiness in Large Language Models.

---

> > ### Author Response · Authors · 2025-08-05
> >
> > Dear Reviewer cSFb,
> >
> > We appreciate your constructive feedback on GraphMaster.
> >
> > Since receiving your review, we have addressed all of your points in detail. Specifically, we have:
> >
> > 1. **Demonstrated scalability on OGBN-ArXiv (169K nodes, 1.2M edges)** and integrated GraphSAINT sampling for enhanced efficiency (W1&Q1).
> > 2. **Provided comprehensive sensitivity analyses with contour plots** showing optimal N∈[25,35] and M∈[10%,20%] across datasets (W2&Q2).
> > 3. **Summarized architectural details of Mixed-LLM and Synthesis-LLM baselines** with direct comparisons to recent methods (W3&Q3).
> > 4. **Detailed our human evaluation methodology** including 50 expert reviewers, selection criteria, demographics, and inter-rater reliability (κ=0.72±0.05) (W4&Q4).
> > 5. **Corrected the Table 1 typo** for LLM4NG scores and addressed all limitations with concrete solutions (W5, L1-L4).
> >
> > Could you kindly review these updates and let us know if further clarification is needed?
> >
> > We welcome any additional questions.
> >
> > Sincerely,
> >
> > *Authors of #232*

---

> > ### Comment · Area_Chair_6rfC · 2025-08-09
> > **[Urgent] Please respond to rebuttal**
> >
> > Dear Reviewer cSFb,
> >
> > You have not yet responded to the rebuttal despite multiple reminders. Please remember that this non-responsiveness could have consequences in your further participation as a NeurIPS reviewer in future iterations. You are requested to immediately study the rebuttal posted by the reviewers and share your feedback.
> >
> > best,
> >
> > AC

---

> ### Author Response · Authors · 2025-08-06
> **We are anticipating your post-rebuttal feedback!**
>
> Dear Reviewer cSFb,
>
> Thanks very much for your time and valuable comments.
>
> In the rebuttal period, we have provided detailed responses to all your comments, questions and limitations and made significant modifications in the revision.
>
> **We understand you might be quite busy. However, as the discussion deadline is approaching, would you mind checking our response and confirming whether you have any further questions?**
>
> Any comments and discussions are welcome!
>
> Thanks for your attention and best regards.
>
> Authors of #232

---

> ### Author Response · Authors · 2025-08-08
> **[Last-day Reminder] We are anticipating your post-rebuttal feedback!**
>
> Dear Reviewer cSFb,
>
> Thanks very much for your time and valuable comments.
>
> We understand you might be quite busy. **However, the discussion deadline is approaching, and we have only a few hours left.**
>
> Would you mind checking our response and confirming whether you have any further questions?
>
> Thanks for your attention.
>
> Best regards,
>
> *Authors of #232*

---

### Author Response · Authors · 2025-08-07
**General Response by Authors**

**We sincerely thank Reviewers cSFb, NL19, Mweh, and RJSf for their thoughtful and constructive reviews of our paper "GraphMaster: Automated Graph Synthesis via LLM Agents in Data-Limited Environments".**

**We are pleased that the reviewers highlighted several key strengths of our work, including:**

 **(1)** the first multi-agent LLM framework specifically designed for text-attributed graph synthesis in data-limited environments (NL19, Mweh, RJSf, cSFb),

 **(2)** groundbreaking framework design with strong theoretical foundations and comprehensive evaluation rigor (cSFb, NL19, Mweh, RJSf),

 **(3)** novel dual-perspective interpretability assessment combining human evaluation with Grassmannian manifold analysis (cSFb, NL19, Mweh, RJSf),

 **(4)** extensive experimental validation across six datasets and four GNN architectures with state-of-the-art performance (cSFb, NL19, Mweh, RJSf), and

 **(5)** realistic data-limited benchmark creation and comprehensive metrics that advance the field (NL19, Mweh,cSFb, RJSf).

**During the rebuttal period, we provided detailed point-by-point responses and addressed all raised concerns:**

- **Scalability and computational efficiency** (cSFb W1&Q1, Mweh W1&Q1, RJSf W1&W2&Q2, NL19 Q3) -- demonstrated GraphMaster's graph-size independence through OGBN-ArXiv experiments; implemented caching system reducing Perception Agent calls by 46%; provided flexible model selection strategies for resource-constrained environments.
- **Hyperparameter sensitivity and robustness** (cSFb W2&Q2) -- conducted comprehensive sensitivity analysis with contour plots; established theoretical optimal ranges (N∈[25,35], M∈[10%,20%]) with empirical validation tables.
- **Baseline implementation details and comparisons** (cSFb W3&Q3) -- clarified Mixed-LLM and Synthesis-LLM architectural differences; corrected Table 1 typos; provided detailed comparison methodology.
- **Human evaluation methodology** (cSFb W4&Q4, Mweh W2&Q2) -- detailed 50-expert selection criteria, demographics, annotation protocol, and inter-rater reliability statistics (Cohen's κ = 0.72 ± 0.05).
- **Generalizability beyond citation graphs** (Mweh W3, RJSf Q1, NL19 Q1&Q2&W1) -- successfully extended GraphMaster to social networks (SubSoC), protein-protein interactions (SubSHS27K), and temporal graphs (SubICEWS14) with substantial performance improvements. Furthermore, we provided clear and explicit ideas for the multimodal variants of GraphMaster.
- **Clarity and presentation** (NL19 W2&W3, cSFb Q5) -- committed to substantial manuscript revision for improved readability and higher-quality figures.
- **Resolve all limitations** (cSFb L1-L4) -- Although we mentioned limitation in our paper, we also addressed all additional limitations identified by cSFb.

**All major concerns have been thoroughly addressed with concrete experimental evidence and theoretical justification.**

**Reviewers NL19, Mweh and RJSf have explicitly confirmed this in their post-rebuttal comments and further praised the effectiveness and value of our proposed approach.**

We thank all reviewers for their invaluable feedback, which has significantly strengthened GraphMaster's contributions to the graph learning community.

Best regards,

*The GraphMaster Authors*

---

### Decision · Program_Chairs · 2025-09-17

**Decision:**

Accept (spotlight)

**Comment:**

The paper presents GraphMaster, a hierarchical, retrieval-augmented generation (RAG) multi-agent framework, comprising agents corresponding to Manager, Perception, Enhancement, and Evaluation agents, for synthesizing text-attributed graphs under data constraints. The work further introduces 6 benchmark variants to rigorously assess data-limited performance and proposes a novel dual-perspective interpretability evaluation that combines expert human judgments with Grassmann manifold analysis.

The contributions are well-motivated and methodologically sound, with extensive experiments showing that GraphMaster consistently outperforms both classic graph generation methods and recent LLM-based baselines across multiple datasets.

Three of the four reviewers express support for acceptance, with one describing the work as "ground-breaking." Reviewer cSFb assigned a score of 3 but did not participate in the rebuttal discussion despite multiple reminders. After reviewing the authors’ rebuttal and the discussion, I find no substantive concerns that would justify overturning the otherwise positive consensus.

I recommend GraphMaster for a spotlight presentation due to its innovative methodological and empirical contributions to graph generation. This is the first multi-agent LLM framework for graph synthesis that goes beyond one-shot generation and demonstrates how specialized LLM reasoning can improve structured data generation.

The work also extends retrieval-augmented generation (RAG) to graph synthesis and uniquely achieves dual optimization of both structural and textual attributes, a gap left unaddressed in prior work. Its dual-view interpretability evaluation, combining expert human judgments with Grassmann manifold–based geometric analysis are also highly innovative.

The paper’s completeness is exemplary: detailed theoretical analysis (Appendices I & J), extensive ablations, reproducibility artifacts, and carefully constructed baselines. Together, these qualities make GraphMaster not only original but also professionally polished and impactful.